# Compressing Large Language Models using Low Rank and Low Precision Decomposition

**Rajarshi Saha**
Stanford University

**Naomi Sagan**
Stanford University

**Varun Srivastava**
Stanford University

**Andrea J. Goldsmith**
Princeton University

**Mert Pilanci**
Stanford University

## Abstract

The prohibitive sizes of Large Language Models (LLMs) today make it difficult to deploy them on memory-constrained edge devices. This work introduces CALDERA – a new post-training LLM compression algorithm that harnesses the inherent low-rank structure of a weight matrix $\mathbf{W}$ by approximating it via a low-rank, low-precision decomposition as $\mathbf{W} \approx \mathbf{Q} + \mathbf{LR}$. Here, $\mathbf{L}$ and $\mathbf{R}$ are low rank factors, and the entries of $\mathbf{Q}$, $\mathbf{L}$ and $\mathbf{R}$ are quantized. The model is compressed by substituting each layer with its $\mathbf{Q} + \mathbf{LR}$ decomposition, and the zero-shot performance of the compressed model is evaluated. Additionally, $\mathbf{L}$ and $\mathbf{R}$ are readily amenable to low-rank adaptation, consequently enhancing the zero-shot performance. CALDERA obtains this decomposition by formulating it as an optimization problem $\min_{\mathbf{Q},\mathbf{L},\mathbf{R}}\|(\mathbf{Q} + \mathbf{LR} - \mathbf{W})\mathbf{X}^{\top}\|_{\mathrm{F}}^{2}$, where $\mathbf{X}$ is the calibration data, and $\mathbf{Q}, \mathbf{L}, \mathbf{R}$ are constrained to be representable using low-precision formats. Theoretical upper bounds on the approximation error of CALDERA are established using a rank-constrained regression framework, and the tradeoff between compression ratio and model performance is studied by analyzing the impact of target rank and quantization bit budget. Results illustrate that compressing LlaMa-2 7B/13B/70B and LlaMa-3 8B models using CALDERA outperforms existing post-training LLM compression techniques in the regime of less than $2.5$ bits per parameter. The implementation is available at: https://github.com/pilancilab/caldera.

## 1 Introduction

Large Language Models (LLMs) stand out due to their remarkable ability to generate human-like text, thereby supporting a diverse range of applications ranging from writing assistance to code generation. These models leverage vast datasets and significant computational resources to achieve their impressive functionality. The architecture of LLMs typically includes multiple layers, each with weight matrices essential for encoding various aspects of the training data – from simple syntactic patterns to complex semantic relationships. However, the substantial size of these trained models leads to high computational costs and considerable energy consumption during inference, which can be challenging for deployment in resource-constrained environments. As LLMs continue to expand in scale, compression techniques to reduce the memory and computational requirements of the models are becoming crucial to ensure their broad accessibility.

Due to the correlated nature of language syntax and semantics learned during training, often, the weight matrices of LLMs exhibit redundancy, which manifests as a low-rank structure. This redundancy suggests the potential for compression without substantial loss in performance. This work introduces CALDERA: **C**alibration **A**ware **L**ow-Precision **DE**composition with Low-**R**ank **A**daptation, which compresses LLMs by leveraging the approximate low rank structure inherent in

38th Conference on Neural Information Processing Systems (NeurIPS 2024).

these weight matrices. Given a matrix $\mathbf{W} \in \mathbb{R}^{n \times d}$, CALDERA approximates it as $\mathbf{W} \approx \mathbf{Q} + \mathbf{LR}$, where $\mathbf{Q} \in \mathbb{R}^{n \times d}$, $\mathbf{L} \in \mathbb{R}^{n \times k}$ and $\mathbf{R} \in \mathbb{R}^{k \times d}$. Here, the *left* and *right* low rank factors, respectively $\mathbf{L}$ and $\mathbf{R}$, are tall and wide matrices, and $k$ is the target rank. Furthermore, the entries of $\mathbf{Q}$, $\mathbf{L}$ and $\mathbf{R}$ are represented using low-precision formats with $\mathrm{B_Q}$, $\mathrm{B_L}$ and $\mathrm{B_R}$ bits per entry, respectively.

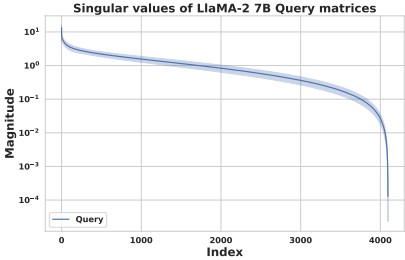

Figure 1: Decaying spectrum of weight matrices (aka, *"approximate low-rank"*)

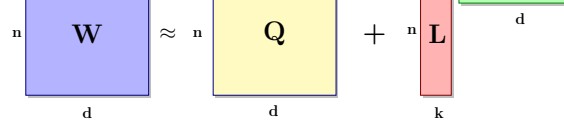

Figure 2: CALDERA decomposes a full-precision weight matrix into a low-rank component ($\mathbf{LR}$), which captures the contribution of the top singular values using $\mathrm{B_L}, \mathrm{B_R}$ bits, and $\mathbf{Q}$ for the trailing singular values with $\mathrm{B_Q}$ bits, enabling flexible precision settings for each component. Typically, $\mathrm{B_Q} < \mathrm{B_L}, \mathrm{B_R}$.

Since the singular value profile (aka spectrum) of the weight matrices of an LLM follow a decaying profile as shown in Fig. 1, the low-rank factors $\mathbf{L}$ and $\mathbf{R}$ capture the effect of the large singular components of $\mathbf{W}$ with high fidelity. Moreover, the backbone $\mathbf{Q}$, which is quantized aggressively – for instance, using $\mathrm{B_Q} = 2$ bits, coarsely captures the essence of the moderately decaying and low singular components of $\mathbf{W}$. CALDERA substitutes each weight matrix $\mathbf{W}$ in an LLM, with its approximate low-precision and low-rank decomposition $\mathbf{Q} + \mathbf{LR}$, resulting in a post-training quantization strategy that delivers state-of-the-art zero-shot performance. In addition, since usually $k \ll \min\{n, d\}$, implying that the total number of parameters in $\mathbf{LR}$ is much smaller compared to the number of entries in $\mathbf{W}$ (i.e., $k(n + d) \ll nd$), CALDERA can readily fine-tune (or *"adapt"*) the low rank factors $\mathbf{L}$ and $\mathbf{R}$ in order to boost the zero-shot results.

## 1.1 Significance and Related Works

Recent efforts have explored various avenues for compression, including but not limited to weight pruning, quantization, and the use of parameter-efficient training methods – each approach offering distinct advantages and tradeoffs. This section briefly reviews the current methodologies, highlighting the contributions and limitations of some studies closely related to this work.

**LLM Compression and Outlier Mitigation**: Recent studies like SmoothQuant [42], OPTQ [8], QuIP [3], AQLM [7], and QuIP# [36] consider the challenging regime of sub-4 bit post-training LLM quantization. These works collectively emphasize the need to manage the impact of outliers, i.e., weights with unusually high magnitudes. Accommodating outliers necessitates choosing the dynamic range (or scale) of a quantizer to be high, consequently increasing the quantization error. QuIP equalizes (and reduces) the weight matrices by using a randomized matrix transform, and subsequently, QuIP# employs E8 lattice to make the weights more amenable to vector quantization. Both QuIP and QuIP# use a refined variant of the column-wise quantization method proposed in OPTQ, wherein error feedback from previously quantized columns of a matrix is used to compensate for the error incurred while quantizing subsequent columns. CALDERA utilizes this diverse arsenal of strategies and builds on top of QuIP#, while capitalizing on the approximate low-rank structure of LLM weight matrices. While it is possible to obtain even more aggressively compressed LLMs [24], this approach requires training from scratch, which is computationally demanding.

**Parameter Efficient Fine-Tuning (PEFT)**: In a related yet distinct vein of work, PEFT methods have gained significant momentum, aiming to adapt LLMs to specific tasks without extensive computational overhead. Recent studies such as QLoRA [5], LoftQ [22], and LQ-LoRA [13] have explored the intersection of PEFT and quantization, demonstrating that fine-tuning through low-rank updates, as originally proposed in LoRA [16], can mitigate the performance losses due to quantization. Given that CALDERA yields a decomposition $\mathbf{Q} + \mathbf{LR}$, the low-rank components are particularly suitable for fine-tuning with any existing PEFT methods, thereby enhancing the zero-shot capabilities.

**Low Rank Approximation**: The rank-$k$ approximation of a matrix $\mathbf{A} \in \mathbb{R}^{n \times d}$ can be represented by the factorization $\mathbf{A} \approx \mathbf{LR}$, with $\mathbf{L} \in \mathbb{R}^{n \times k}$ and $\mathbf{R} \in \mathbb{R}^{k \times d}$, where $k \leq \min\{n, d\}$. Known as the Burer-Monteiro factorization, this method substantially decreases the number of parameters,

thus reducing computational demands. Recent studies such as LoRD [18], ASVD [44], FWSVD [15], LASER [33], LQER [45], and ZeroQuant-V2 [43] have explored the efficacy of low-rank structures in LLM weights, treating low-rank factorization and quantization independently. In contrast, LPLR [31] approaches this by uniquely formulating a joint optimization problem for generic matrices, while simultaneously leveraging the equalization property of randomized transforms, as in [3]. CALDERA formally leverages this inherent low-rank structure for LLM compression alongside existing frameworks such as QuIP# [36] and LoftQ [22], providing additional flexibility for compression. Furthermore, rigorous theoretical guarantees are derived using a rank-constrained regression framework for obtaining a low precision and low-rank decomposition, thereby also analytically demonstrating its superiority over rank-agnostic strategies.

## 2 Problem Formulation

In a neural network layer, a weight matrix $\mathbf{W}$ transforms an input activation $\mathbf{x}$ into an output activation given by $\mathbf{W}\mathbf{x}$. This transformation can be succinctly described using the matrix's singular value decomposition (SVD). For any matrix $\mathbf{A} \in \mathbb{R}^{n \times d}$, the SVD is $\mathbf{A} = \sum_i \sigma_i \mathbf{u}_i \mathbf{v}_i$, where $\sigma_i, \mathbf{u}_i, \mathbf{v}_i$ are the $i^{\text{th}}$ singular value and the corresponding left and right singular vectors, respectively. The impact of each singular component $\mathbf{u}_i \mathbf{v}_i$ on the matrix's transformation is determined by the magnitude of $\sigma_i$. Given that weight matrices exhibit a decaying singular value profile (Fig. 1), indicating an approximate low-rank structure, lesser contributing singular components can be pruned with minimal impact on the functionality of the matrix, ensuring minimal distortion in the output activations.

CALDERA approximates the weight matrix of a neural network, $\mathbf{W}$, as a low-precision, low-rank decomposition, $\mathbf{W} \approx \mathbf{Q} + \mathbf{LR}$, with all components $\mathbf{Q}, \mathbf{L}, \mathbf{R}$ in low-precision format. Unlike previous works such as [13, 22, 44, 45], which represent the low-rank factors $\mathbf{L}$ and $\mathbf{R}$ in high-precision (16 or 32-bit floating point), this work extends their representation to low-precision. This further reduces the memory footprint while preserving performance. Alternatively, for the same memory footprint, it allows the target rank $k$ to be higher, thereby capturing the low rank structure with higher fidelity by including more of the higher singular value components. The following paragraph formalizes this as a constrained optimization problem.

For a given quantizer, let $\mathbb{Q}$ denote the set of discrete quantization points in $\mathbb{R}$. For B-bit quantization, the cardinality of $\mathbb{Q}$ satisfies $\log_2 |\mathbb{Q}| \leq \text{B}$. Consider a matrix $\mathbf{W} \in \mathbb{R}^{n \times d}$. The goal of this work is to obtain an decomposition $\mathbf{W} \approx \mathbf{Q} + \mathbf{LR}$ by approximately solving the minimization problem

$$\min_{\mathbf{Q}, \mathbf{L}, \mathbf{R}} \left\| (\mathbf{Q} + \mathbf{LR} - \mathbf{W}) \mathbf{X}^\top \right\|_{\text{F}}^2 \quad \text{subject to} \quad \mathbf{Q} \in \mathbb{Q}_{\text{Q}}^{n \times d}, \ \mathbf{L} \in \mathbb{Q}_{\text{L}}^{n \times k}, \ \text{and} \ \mathbf{R} \in \mathbb{Q}_{\text{R}}^{k \times d}. \quad (1)$$

Here, $\mathbb{Q}_{\text{Q}}, \mathbb{Q}_{\text{L}}$ and $\mathbb{Q}_{\text{R}}$ denote the lattice codebooks used to quantize $\mathbf{Q}, \mathbf{L}$ and $\mathbf{R}$, using $\text{B}_{\text{Q}}, \text{B}_{\text{L}}$ and $\text{B}_{\text{R}}$ bits, respectively. Furthermore, $\mathbf{X} \in \mathbb{R}^{m \times d}$ is a calibration matrix that aims to preserve the Frobenius norm error of the compressed layer output activations. If $\mathbf{W}$ is the first layer's weight matrix, $\mathbf{X}$ includes input embeddings from a calibration dataset, such as a subset of RedPajama [34], with the $i^{\text{th}}$ row representing the $i^{\text{th}}$ datapoint. For intermediate layers, $\mathbf{X}$ contains the input activations, which are the output activations of the preceding layer.

Using the Frobenius norm of the output of a layer as a proxy objective for quantizing the weight matrices of an LLM is a popular strategy, and was used in prior work of Nagel et al. [27] . This proxy objective function is particularly useful for post-training quantization of LLMs because their large size makes it difficult to apply sophisticated compression methods.

## 3 Proposed Algorithm: Calibration-Aware Low-Precision Decomposition with Low Rank Adaptation

This section introduces CALDERA to approximately solve (1) and get a $\mathbf{Q} + \mathbf{LR}$ decomposition of a weight matrix $\mathbf{W}$ using the calibration matrix $\mathbf{X}$. The pseudocode is provided in Alg. 1. It consists of a nested loop for alternately optimizing the variables $\mathbf{Q}, \mathbf{L}$ and $\mathbf{R}$. Suppose $Q_{\text{Q}}, Q_{\text{L}}$ and $Q_{\text{R}}$, respectively, denote quantizers used for quantizing $\mathbf{Q}, \mathbf{L}$ and $\mathbf{R}$. For instance, they can refer to uniformly dithered scalar quantizers, as described in App. G.2. Initially, the low-rank factors are set to $\mathbf{0}$, and $\mathbf{W}$ is quantized using the LDLQ quantizer proposed in [3, §3.1]. LDLQ is an adaptive

quantization method that iteratively quantizes [8] each column of $\mathbf{W}$ using $Q_Q$ to get $\mathbf{Q}$ as

$$\mathbf{Q}^{(k)} = Q_Q(\mathbf{W}^{(k)} + (\mathbf{W}^{(1:k-1)} - \mathbf{Q}^{(1:k-1)})\mathbf{a}_k), \tag{2}$$

where $\mathbf{Q}^{(k)}, \mathbf{W}^{(k)}$ denote the $k^{\text{th}}$ column, $\mathbf{W}^{(1:k-1)}$ denotes the first $k$ columns, $Q_Q$ has a bit-budget $B_Q$, and $\mathbf{a}_k \in \mathbb{R}^{k-1}$ is a learnable sequence of vectors. Update Eq. (2) incorporates linear feedback from already quantized columns, it can be seen that $\mathbf{Q}$ satisfies $\mathbf{Q} = Q_Q(\mathbf{W} + (\mathbf{W} - \mathbf{Q})\mathbf{M})$, where the feedback matrix $\mathbf{M}$ is a strictly upper triangular matrix with columns $\mathbf{a}_k$. Defining $\mathbf{H} \triangleq \frac{1}{m}\mathbf{X}^\top\mathbf{X}$ to be the (scaled) Hessian of the least squares objective in (1), [3] show that the optimal feedback matrix is the $\mathbf{M}$ obtained from the LDL decomposition of $m\mathbf{H}$, given by $m\mathbf{H} = (\mathbf{M}+\mathbf{I})\mathbf{D}(\mathbf{M}+\mathbf{I})^\top$.

Subsequently, $\mathbf{Q}$ is fixed and the Low-Precision Low-Rank (LPLR) factorization of the residual, $(\mathbf{W} - \mathbf{Q})$, is computed. This is done by the LPLRFACTORIZE submodule (Alg. 2), which is a refined version of the LPLR algorithm proposed in [31]. For a given matrix $\mathbf{A}$, Alg. 2 minimizes

$$\min_{\mathbf{L},\mathbf{R}} \left\|(\mathbf{LR} - \mathbf{A})\mathbf{X}^\top\right\|_{\text{F}}^2 \quad \text{subject to} \quad \mathbf{L} \in \mathbb{Q}_{\text{L}}^{n \times k}, \text{ and } \mathbf{R} \in \mathbb{Q}_{\text{R}}^{k \times d}, \tag{3}$$

where $Q_L$ and $Q_R$ use $B_L$ and $B_R$ bits, respectively. In contrast to [31], the objective in (3) is calibration-data aware. Therefore, the update equations are derived using a rank-constrained regression framework, as described in App. B. Moreover, lines 7 to 14 in LPLRFACTORIZE iteratively refine the estimates of $\mathbf{L}$ and $\mathbf{R}$, and can only yield a smaller Frobenius norm error. The left and right low-rank factor update equations are described as follows.

**Initialization**: In the absence of quantization constraints, a globally optimal solution to the optimization problem (3) can be found as described later in lemma 4.2. Consequently, the low-rank factors are initialized using rank-constrained regression in lines $2 - 4$. Since subsequent quantization disrupts optimality of the solution, the factors are iteratively updated to minimize this distortion.

**Updating L**: To update the left factor $\mathbf{L}$, lines 5 and 9 of Alg. 2 solves $\min_\mathbf{Z}\|(\mathbf{ZR} - \mathbf{A})\mathbf{X}^\top\|_{\text{F}}^2$. For a fixed $\mathbf{R}$, this is a least squares minimization, whose solution is available is closed form as $\grave{\mathbf{L}} = (\mathbf{AX}^\top)(\mathbf{RX}^\top)^\dagger = \mathbf{AHR}^\top(\mathbf{RHR}^\top)^{-1}$, as derived in App. C.1.

**Updating R**: Line 8 of Alg. 2, updates the right factor $\mathbf{R}$ by keeping $\mathbf{L}$ fixed and solving $\min_\mathbf{Z}\|(\mathbf{LZ} - \mathbf{A})\mathbf{X}^\top\|_{\text{F}}^2$. As this is an under-determined linear system, there exist multiple solutions for $\mathbf{Z}$, all attaining the same objective function value. It is shown in App. C.1 that $\grave{\mathbf{R}} = \mathbf{L}^\dagger\mathbf{AHH}^\dagger$ is a solution. The corresponding error is also obtained, which is used in the derivation of Thm. 4.1.

**Computational Complexity**: A high-level calculation is provided here, and detailed discussions can be found in App. D. It is worthwhile to note that the closed form expressions of $\grave{\mathbf{L}}$ and $\grave{\mathbf{R}}$, which are iteratively quantized, are functions of the Hessian $\mathbf{H} = \frac{1}{m}\mathbf{X}^\top\mathbf{X}$. Therefore, $\mathbf{H}$ can be computed offline initially, per LLM, by doing a single forward pass, and subsequently used for all model quantization experiments. For each layer, this pre-processing includes computing $\mathbf{H}$ and its LDL decomposition, along with computing $\mathbf{HH}^\dagger$, requiring a total of $O(md^2 + 2d^3)$ multiplications. Each outer iteration involves an LDLQ quantization. Quantizing the $k^{\text{th}}$ column has complexity $O(nk)$, since feedback from $k$ already quantized columns need to be incorporated. Hence, quantizing a matrix in $\mathbb{R}^{n \times d}$ entails $O(n^2 + 3n)$ complexity. Moreover, LPLRFACTORIZE requires $O(m^2(n + d))$ to initialize, and subsequently, each inner iteration entails $O(ndk)$. Assuming $n, d \geq m \gg k$, and keeping only the dominant terms, the total complexity of CALDERA, not including the complexity of the pre-processing discussed earlier, is $O\left(T_{\text{out}}\left(n^2 + m^2(n + d) + ndk\,T_{\text{in}}\right)\right)$.

**Fine tuning via Low-Rank Adaptation**: Once the weight matrices of each layer are replaced by its $\mathbf{Q} + \mathbf{LR}$ approximation, the zero-shot performance of (post-training) quantized model can be evaluated. §5 shows that CALDERA quantized models outperform existing strategies. Additionally, if desired, the low-rank factors $\mathbf{L}$ and $\mathbf{R}$ can be further fine-tuned using low-rank adaptation [13, 16, 22] on a small task-specific dataset. While the initialization of the fine-tuning step has quantized $\mathbf{Q}, \mathbf{L}$ and $\mathbf{R}$, the fine-tuned factors are represented using 16-bits (BF16 format). Although this leads to a slight increase in the memory footprint, the performance gains from fine-tuning are substantial.

## 4 Approximation Error Analysis

The approximation error upper bounds are derived via a rank-constrained regression framework. Thm. 4.1 below (formally stated and proved in App. C.4) is an informal version of the main theoretical result of this paper, and provides an upper bound on the Frobenius norm error when CALDERA

approximates a weight matrix $\mathbf{W}$ is as $\mathbf{W} \approx \mathbf{Q} + \mathbf{LR}$ by solving the optimization problem (1) using Alg. 1. For convenience of analysis, it is assumed that the dynamic range of $Q_Q$, denoted as R, is chosen to be high enough, ensuring it remains unsaturated. Consequently, for a scalar input the quantization error from $Q_Q$ has zero mean and bounded variance, given by $\frac{\Delta^2}{4} = \frac{R^2}{(2^{B_Q}-1)^2}$.

**Theorem 4.1. Approximation error of CALDERA (Informal)** *Given $\mathbf{W} \in \mathbb{R}^{n \times d}$ and $\mathbf{X} \in \mathbb{R}^{m \times d}$ with $m \leq d$, let $\mathbf{D}$ be obtained from the LDL decomposition $\mathbf{X}^\top \mathbf{X} = m\mathbf{H} = (\mathbf{M} + \mathbf{I})\mathbf{D}(\mathbf{M} + \mathbf{I})^\top$, and $\lambda_{\max}, \lambda_{\min}$ denote the max and min eigenvalues of $\mathbf{H}$. Additionally, let $\mathbf{Q} \triangleq \text{LDLQ}(\mathbf{W}, Q_Q)$, where $Q_Q$ has dynamic range $R$ and bit-budget $B_Q$, the quantization error be $\boldsymbol{\eta} \triangleq Q_Q(\mathbf{Q} + (\mathbf{W} - \mathbf{Q})\mathbf{M}) - (\mathbf{Q} + (\mathbf{W} - \mathbf{Q})\mathbf{M})$, and $\sigma_1 \geq \ldots \geq \sigma_k \ldots$ be the singular values of $\mathbf{X}(\mathbf{W} - \mathbf{Q})^\top$. If the target rank $k$ satisfies $0.25 \lambda_{\min}^{1/2}(m\sigma_1)^{-1}\lambda_{\max}^{-3/2}\sum_{i>k}\sigma_i^2 \leq k \leq m$, and the dynamic ranges of $Q_L$ and $Q_R$ are set as $R_L = \frac{2\sigma_1}{\sigma_k \sqrt{m\lambda_{\min}}}$ and $R_R = \sigma_1$, then $\mathbf{Q}, \mathbf{L}$ and $\mathbf{R}$ returned by Alg. 1 satisfy*

$$\frac{1}{nm} \mathbb{E} \left\| (\mathbf{Q} + \mathbf{LR} - \mathbf{W})\mathbf{X}^\top \right\|_F^2 \leq \frac{1}{n}\sum_{i>k}\mathbb{E}\lambda_i(\boldsymbol{\eta}\mathbf{D}\boldsymbol{\eta}^\top) + \epsilon \lesssim \frac{4d\lambda_{\max}R^2}{\pi\left(2^{B_Q}-1\right)^2}\left(1 - \frac{k}{2n}\right)^2 + \epsilon,$$

*while utilizing an average budget of $\frac{1}{2}\log_2\left(\frac{k\sigma_1^3}{m\epsilon\sigma_k}\frac{\lambda_{\max}}{\lambda_{\min}}\sqrt{d/n}\right)$ bits per parameter for the low-rank factors $\mathbf{L}$ and $\mathbf{R}$, when $n \approx d$. Here, the expectation is over the stochasticity of the quantizers.*

An informal version of the main result is provided here, and the formal version including specific constant values, along with the derivation, can be found in App. C.4. The requirement $m \leq d$ is not restrictive, because when $\mathbf{H}$ is positive definite, (1) can be rewritten as $\|(\mathbf{Q} + \mathbf{LR} - \mathbf{W})\mathbf{X}^\top\|_F^2 = \|(\mathbf{Q} + \mathbf{LR} - \mathbf{W})\mathbf{H}^{1/2}\|_F^2$, ensuring $m = d$. This is detailed further in App. C.5. The approximation error upper bound given by Thm. 4.1 can be directly compared with the result of Chee et al. [3, Thm. 1], which states that for vanilla LDLQ without LPLRFACTORIZE,

$$\mathbb{E}\left\|(\mathbf{Q} - \mathbf{W})\mathbf{X}^\top\right\|_F^2 \leq \mathbb{E}\left[\text{Tr}\left(\boldsymbol{\eta}\mathbf{D}\boldsymbol{\eta}^\top\right)\right] = \sum_{i=1}^{n}\mathbb{E}\lambda_i(\boldsymbol{\eta}\mathbf{D}\boldsymbol{\eta}^\top). \tag{4}$$

Evidently, Alg. 1 yields a smaller error provided, $\sum_{i>k}\mathbb{E}\lambda_i(\boldsymbol{\eta}\mathbf{D}\boldsymbol{\eta}^\top) < \sum_{i=1}^{k}\mathbb{E}\lambda_i(\boldsymbol{\eta}\mathbf{D}\boldsymbol{\eta}^\top) - \epsilon$, where $\epsilon$ can be chosen to be arbitrarily small. Furthermore, since the expression in Thm. 4.1 consists of two terms, namely, the rank-constrained regression error, which depends on the target rank $k$, and the additive quantization error of $\epsilon$, which is dictated by the bit-budgets used for $\mathbf{L}$ and $\mathbf{R}$, this upper

---

**Algorithm 1: CALDERA: Calibration Aware Low-Precision DEcomposition with Low-Rank Adaptation**

**Input:** Matrix: $\mathbf{W} \in \mathbb{R}^{n \times d}$, Target rank: $k$, Calibration matrix: $\mathbf{X} \in \mathbb{R}^{m \times d}$, Outer and inner iterations: $T_{\text{out}}, T_{\text{in}}$, Quantizers: $Q_Q, Q_L, Q_R$, Flag: EnableLoRA, Fine-tune rank: $r$

**Output:** LPLR decomposition: $\mathbf{Q} \in \mathbb{Q}_Q^{n \times d}, \mathbf{L} \in \mathbb{Q}_L^{n \times k}, \mathbf{R} \in \mathbb{Q}_R^{k \times d}$ s.t. $\mathbf{W}\mathbf{X}^\top \approx (\mathbf{Q} + \mathbf{LR})\mathbf{X}^\top$

1 **Initialize:** $t \leftarrow 0$, $\mathbf{L}_0 \leftarrow \mathbf{0}$, $\mathbf{R}_0 \leftarrow \mathbf{0}$, MinError $\leftarrow \infty$
2 **while** $t < T_{\text{out}}$ **do**
3     Update $\mathbf{Q}$: $\mathbf{Q}_{t+1} \leftarrow \text{LDLQ}(\mathbf{W} - \mathbf{L}_t\mathbf{R}_t, Q_Q)$
4     Update low-rank factors:
      $\mathbf{L}_{t+1}, \mathbf{R}_{t+1} \leftarrow \text{LPLRFACTORIZE}(\mathbf{W} - \mathbf{Q}_{t+1}, k, \mathbf{X}, Q_L, Q_R, T_{\text{in}})$
5     **if** $\|(\mathbf{Q}_{t+1} + \mathbf{L}_{t+1}\mathbf{R}_{t+1} - \mathbf{W})\mathbf{X}^\top\|_F^2 <$ MinError **then**
6         $\mathbf{Q}_{\text{best}} \leftarrow \mathbf{Q}_{t+1}$, $\mathbf{L}_{\text{best}} \leftarrow \mathbf{L}_{t+1}$, $\mathbf{R}_{\text{best}} \leftarrow \mathbf{R}_{t+1}$,
        MinError $\leftarrow \|(\mathbf{Q}_{t+1} + \mathbf{L}_{t+1}\mathbf{R}_{t+1} - \mathbf{W})\mathbf{X}^\top\|_F^2$
7     **end**
8     $t \leftarrow t + 1$
9 **end**
10 **if** EnableLoRA is TRUE **then**
11     Further Fine-tune top-$r$ singular components of $\mathbf{L}_{\text{best}}$ and $\mathbf{R}_{\text{best}}$ to 16-bit precision using Low-Rank Adaptation (LoRA) (as in [13, 16, 22])
12 **end**
13 **return** $\mathbf{Q}_{\text{best}}, \mathbf{L}_{\text{best}}, \mathbf{R}_{\text{best}}$

**Algorithm 2:** LPLRFACTORIZE($\mathbf{A}, k, \mathbf{X}, Q_L, Q_R, T_{in}$): LPLR factorization submodule

---

**Input:** Matrix: $\mathbf{A} \in \mathbb{R}^{n \times d}$, Target rank: $k$, Calibration matrix: $\mathbf{X} \in \mathbb{R}^{m \times d}$, Iterations: $T_{in}$,
    Quantizers: $Q_L, Q_R$

**Output:** Low precision Low Rank factors: $\mathbf{L} \in \mathbb{Q}^{n \times k}, \mathbf{R} \in \mathbb{Q}^{k \times d}$ s.t. $\mathbf{A} \mathbf{X}^\top \approx \mathbf{L} \mathbf{R} \mathbf{X}^\top$

---

**1 Initialize:** Iteration counter: $i \leftarrow 0$
**2** Compute SVD of $\mathbf{X}$ as $\mathbf{U}\widetilde{\boldsymbol{\Sigma}}\mathbf{V}^\top$.
**3** Compute SVD of $\mathbf{U}^\top \mathbf{X} \mathbf{A}^\top$ as $\grave{\mathbf{U}}\grave{\boldsymbol{\Sigma}}\grave{\mathbf{V}}^\top$
**4** Get right low-rank factor: $\mathbf{R}_0 \leftarrow Q_R(\mathbf{I}_k^\top \grave{\boldsymbol{\Sigma}}\grave{\mathbf{V}}^\top)$
**5** Get left low-rank factor: $\mathbf{L}_0 \triangleq Q_L(\grave{\mathbf{L}}_0)$, where $\grave{\mathbf{L}}_0 = \arg\min_{\mathbf{Z} \in \mathbb{R}^{k \times d}} \left\| (\mathbf{Z}\mathbf{R}_0 - \mathbf{A})\mathbf{X}^\top \right\|_F^2$
**6** $\mathbf{L}_{best} \leftarrow \mathbf{L}_0, \mathbf{R}_{best} \leftarrow \mathbf{R}_0, \text{MinError} \leftarrow \left\| (\mathbf{L}_0\mathbf{R}_0 - \mathbf{A})\mathbf{X}^\top \right\|_F^2$.
**7 while** $i < T_{in}$ **do**
**8**   Update right: $\mathbf{R}_{i+1} \leftarrow Q_R(\grave{\mathbf{R}}_{i+1})$, where $\grave{\mathbf{R}}_{i+1} = \arg\min_{\mathbf{Z} \in \mathbb{R}^{k \times d}} \left\| (\mathbf{L}_i\mathbf{Z} - \mathbf{A})\mathbf{X}^\top \right\|_F^2$
**9**   Update left: $\mathbf{L}_{i+1} \leftarrow Q_L(\grave{\mathbf{L}}_{i+1})$, where $\grave{\mathbf{L}}_{i+1} = \arg\min_{\mathbf{Z} \in \mathbb{R}^{n \times k}} \left\| (\mathbf{Z}\mathbf{R}_i - \mathbf{A})\mathbf{X}^\top \right\|_F^2$
**10**   **if** $\left\| (\mathbf{L}_{i+1}\mathbf{R}_{i+1} - \mathbf{A})\mathbf{X}^\top \right\|_F^2 < \text{MinError}$ **then**
**11**    $\mathbf{L}_{best} \leftarrow \mathbf{L}_{i+1}, \mathbf{R}_{best} \leftarrow \mathbf{R}_{i+1}, \text{MinError} \leftarrow \left\| (\mathbf{L}_{i+1}\mathbf{R}_{i+1} - \mathbf{A})\mathbf{X}^\top \right\|_F^2$
**12**   **end**
**13**   $i \leftarrow i + 1$
**14 end**
**15 return** $\mathbf{L}_{best}, \mathbf{R}_{best}$

---

bound can be made arbitrarily small by ensuring that the two terms are approximately equal, i.e., $\mathbb{E}\|(\mathbf{Q} + \mathbf{L}\mathbf{R} - \mathbf{W})\mathbf{X}^\top\|_F^2$ is upper bounded by $2\epsilon$. This is apparent in the following regimes:

(i) $k \ll n$: In this regime, $k$ is treated as a constant as $n$ grows. Then, if the bit-budget $B_Q$ satisfies

$$B_Q \geq \log_2\left(2R(\pi\epsilon)^{-1/2}\sqrt{nmd\,\lambda_{max}} + 1\right), \quad \text{then} \quad \mathbb{E}\left\|(\mathbf{Q} + \mathbf{L}\mathbf{R} - \mathbf{W})\mathbf{X}^\top\right\|_F^2 \leq 2\epsilon.$$

(ii) $k = O(n)$: For a fixed $B_Q$, if $k$ is allowed to grow with dimension $n$, then choosing $k$ to satisfy

$$k \geq 2n - (2^{B_Q} - 1)R^{-1}(\pi\epsilon)^{1/2}(md\lambda_{max}^{-1/2})\sqrt{n} \quad \text{ensures} \quad \mathbb{E}\left\|(\mathbf{Q} + \mathbf{L}\mathbf{R} - \mathbf{W})\mathbf{X}^\top\right\|_F^2 \leq 2\epsilon.$$

This implies that the upper bound can be made arbitrarily small by either (i) increasing the bit-budget of the backbone, i.e., $B_Q$, for a fixed rank $k$, or (ii) increasing the rank $k$ for a fixed $B_Q$, for example, $B_Q = 2$. Alternatively stated, this provides a tunable knob for controlling the error by trading off the allocated bit-budget between the backbone $\mathbf{Q}$ and the low-rank factors $\mathbf{L}, \mathbf{R}$.

## 4.1 Analysis Outline

In this section, a brief proof sketch is presented, highlighting the major challenges in the proof and how they are addressed. For analysis, $\mathbf{Q}$ is assumed to be updated prior to $\mathbf{L}, \mathbf{R}$ in Alg. 1. However, in practice, the update order is inconsequential, and can be swapped, depending on whichever yields a smaller error. The complete derivation of the approximation error is provided in App. C. A key ingredient of the proof is the solution of the rank-constrained regression problem, which is defined as,

$$\min_{\text{rank}(\mathbf{Z}) \leq k} \|\mathbf{X}\mathbf{Z} - \mathbf{Y}\|_F^2. \tag{5}$$

Although this problem is non-convex, it can be solved to global optimality via two SVDs [41]. The following lemma characterizes the solution to the optimization problem in (5).

**Lemma 4.2.** *Given* $\mathbf{Y} \in \mathbb{R}^{m \times n}$, *and full rank* $\mathbf{X} \in \mathbb{R}^{m \times d}$, *where* $m \leq d$. *Let* $\mathbf{X} = \mathbf{U}\widetilde{\boldsymbol{\Sigma}}\mathbf{V}^\top$ *and* $\mathbf{U}^\top\mathbf{Y} = \grave{\mathbf{U}}\grave{\boldsymbol{\Sigma}}\grave{\mathbf{V}}^\top$ *denote full SVDs of* $\mathbf{X}$ *and* $\mathbf{U}^\top\mathbf{Y}$. *Then, for* $k \leq m$, *the solution of* (5) *is given by*

$$\mathbf{Z}_* \triangleq \arg\min_{\text{rank}(\mathbf{Z}) \leq k} \|\mathbf{X}\mathbf{Z} - \mathbf{Y}\|_F^2 = \left(\mathbf{V}\mathbf{I}_m\boldsymbol{\Sigma}^{-1}\grave{\mathbf{U}}\mathbf{I}_k\right)\left(\mathbf{I}_k^\top\grave{\boldsymbol{\Sigma}}\grave{\mathbf{V}}^\top\right),$$

where $\mathbf{\Sigma} := \widetilde{\mathbf{\Sigma}} \mathbf{I}_m \in \mathbb{R}^{m \times m}$ *is a diagonal matrix consisting of the non-zero singular values of* $\mathbf{X}$. *Moreover, denoting the non-zero singular values of* $\mathbf{Y}$ *as* $\{\sigma_i(\mathbf{Y})\}_{i=1}^m$, *the optimal value of* (7) *is*

$$\min_{\text{rank}(\mathbf{Z}) \leq k} \|\mathbf{XZ} - \mathbf{Y}\|_{\text{F}}^2 = \|\mathbf{XZ}_* - \mathbf{Y}\|_{\text{F}}^2 = \sum_{i=k+1}^{m} \sigma_i^2(\mathbf{Y}). \qquad (6)$$

The complete lemma (with the case $m > d$), and the derivation, are provided in App. B. Using lemma 4.2, the approximation error of LPLRFACTORIZE is analyzed in App. C.3. Specifically, lemma C.3 shows that for any input matrix $\mathbf{A}$, Alg. 2 with suitably chosen $\mathrm{B_L}$ and $\mathrm{B_R}$, ensures that $\mathbb{E} \left\| (\mathbf{LR} - \mathbf{A})\mathbf{X}^\top \right\|_{\text{F}}^2$, as in (3), can be upper bounded by twice the sum of squared trailing singular values, (ref. (6)). While proving lemma C.3, it is assumed that if $\mathrm{Q_L}$ or $\mathrm{Q_R}$ gets saturated, a trivial output of $\mathbf{L} = \mathbf{0}, \mathbf{R} = \mathbf{0}$ is returned. Therefore, lemmas C.1 and C.2 specify choosing the dynamic ranges $\mathrm{R_R}$ and $\mathrm{R_L}$ to be sufficiently high so that saturation happens with a very low probability. The proof of Thm. 4.1 is completed by using the LDL decomposition of $m\mathbf{H}$ as proposed in [3], along with an application of Marchenko-Pastur approximation to bound the expected eigenvalues of the quantization error, i.e., $\mathbb{E}\lambda_i\left(\boldsymbol{\eta}\boldsymbol{\eta}^\top\right)$, yielding the final inequality.

## 5   Numerical Simulations

The efficacy of CALDERA is assessed by using it to compress four popular open source LLMs from Meta AI, namely, LLaMa-2 7B, LLaMa-2 13B, LLaMa-2 70B [35] and LLaMa-3 8B [26]. The framework is built in PyTorch on top of the QuIP# [36] and LoftQ [22], and is available at https://github.com/pilancilab/caldera.

**Baselines**. The full-rank matrix $\mathbf{Q}$, also referred to as the backbone, is quantized to 2-bits using the LDLQ procedure from QuIP [3, 36], employing an E8 lattice quantizer [39]. For CALDERA, which allows even the low-rank factors, $\mathbf{L}$ and $\mathbf{R}$, to be represented in low-precision, the quantization is also performed with an E8 lattice. Prior to running Alg. 1, a randomized Hadamard transform (RHT) is applied to the left and the right of the input weight matrix, as the incoherence pre-processing step, to equalize the magnitude of the entries making them more robust to quantization. In other words, CALDERA decomposition is performed on $\widetilde{\mathbf{W}} \triangleq \mathbf{H}_{\text{L}}^\top \mathbf{W} \mathbf{H}_{\text{R}}$, where $\mathbf{H}_{\text{L}}$ and $\mathbf{H}_{\text{R}}$ are Hadamard matrices, right-multiplied by a diagonal matrix with i.i.d. $\{\pm 1\}$ entries. In addition, the Hessian matrix obtained from the calibration data is substituted by $\widetilde{\mathbf{H}} \triangleq \mathbf{H}_{\text{R}}^\top \mathbf{H} \mathbf{H}_{\text{R}}$. As described in [3], this improves the quantization error incurred by LDLQ. Further details are provided in App. E.2.

**Metrics**. The performance of CALDERA is evaluated using perplexity on the test splits of the Wikitext2 [25] and C4 [6] datasets, as well as task-specific goodness-of-fit metrics such as zero-shot accuracy for sequence classification. Specifically, zero-shot accuracy was measured on the Winogrande [19], RTE [1, 40], PiQA [2], ARC-Easy, and ARC-Challenge [4] tasks. App. E.3 provides more details regarding these benchmarks. Perplexity was measured using a sequence length equal to the model's maximum context length, i.e., 4096 for LLaMa-2, and 8192 for LLaMa-3. Zero-shot experiments were performed using EleutherAI's Language Model Evaluation Harness [9].

### 5.1   Zero-shot Results

Tables 1 and 2 report the perplexities and accuracies for CALDERA with varying target rank $(k)$ of $\mathbf{L}$ and $\mathbf{R}$. A smaller value is better for perplexity, which is defined as the $\exp(\cdot)$ of the training objective, while zero-shot accuracies are reported as percentages. Per-parameter bit budgets range from $2.1$ (e.g., rank-64 factors in 4-bit precision) to $2.4$ bits (e.g., rank-64 factors in half precision or rank-256 factors in 4-bit precision). For comparison, the $\mathbf{Q} + \mathbf{LR}$ decomposition of weight matrices found in the QuIP# codebase was performed on each model. For the sake of direct comparison, fine-tuning of the diagonal matrices in RHT was omitted. As QuIP# does not support quantized factors, $\mathbf{L}$ and $\mathbf{R}$ are rank-64 in order to ensure that the per-parameter bit-budget remains in the $2 - 2.4$ range. As another baseline comparison, each model is quantized using QuIP# without any low-rank factors. Results for the unquantized models are also provided.

For all models, the rank-256 CALDERA decomposition with 4-bit factors had the lowest perplexity and generally had the highest accuracies. As CALDERA supports quantizing low-rank factors with minimal performance loss, more singular components can be captured compared to using half-precision factors while employing the same number of bits. Consequently, the low-rank factors

Table 1: Zero-shot perplexities (denoted by ↓) and accuracies (↑) for LLaMa-2. $B_Q = 2$ bits throughout.

| Method | Rank | $B_L(=B_R)$ | Avg Bits | Wiki2 ↓ | C4 ↓ | Wino ↑ | RTE ↑ | PiQA ↑ | ArcE ↑ | ArcC ↑ |
|---|---|---|---|---|---|---|---|---|---|---|
| CALDERA (7B) | 64 | 16 | 2.4 | 7.36 | 9.47 | 64.6 | 66.4 | 73.7 | 60.8 | 31.7 |
| CALDERA (7B) | 64 | 4 | 2.1 | 7.37 | 9.74 | 63.7 | 62.1 | 72.3 | 60.9 | 31.7 |
| CALDERA (7B) | 128 | 4 | 2.2 | 6.76 | 8.83 | 63.8 | 59.9 | 75.1 | **65.1** | **34.6** |
| CALDERA (7B) | 256 | 4 | 2.4 | **6.19** | **8.14** | **66.0** | 60.6 | **75.6** | 63.6 | 34.0 |
| QuIP# (7B, No FT) | 64 | 16 | 2.4 | 7.73 | 10.0 | 63.1 | **66.8** | 71.7 | 63.2 | 31.7 |
| QuIP# (7B, No FT) | 0 | — | 2 | 8.23 | 10.8 | 61.7 | 57.8 | 69.6 | 61.2 | 29.9 |
| CALDERA (13B) | 64 | 4 | 2.08 | 6.04 | 7.98 | 66.9 | 61.0 | 76.0 | 69.5 | 37.2 |
| CALDERA (13B) | 128 | 4 | 2.16 | 5.72 | 7.66 | **67.9** | 58.5 | 76.0 | 68.5 | 38.7 |
| CALDERA (13B) | 256 | 4 | 2.32 | **5.41** | **7.21** | 66.9 | **62.1** | **76.2** | **70.3** | **40.4** |
| QuIP# (13B, No FT) | 0 | — | 2 | 6.06 | 8.07 | 63.6 | 54.5 | 74.2 | 68.7 | 36.2 |
| CALDERA (70B) | 128 | 4 | 2.1 | 4.11 | 5.95 | 75.5 | 69.3 | **79.8** | 76.9 | 47.7 |
| CALDERA (70B) | 256 | 4 | 2.2 | **3.98** | **5.76** | **77.6** | **71.5** | **79.8** | **79.5** | 47.4 |
| QuIP# (70B, No FT) | 0 | — | 2 | 4.16 | 6.01 | 74.2 | 70.0 | 78.8 | 77.9 | **48.6** |
| Unquantized (7B) | 0 | — | 16 | 5.12 | 6.63 | 67.3 | 63.2 | 78.5 | 69.3 | 40.0 |
| Unquantized (13B) | 0 | — | 16 | 4.57 | 6.05 | 69.5 | 61.7 | 78.8 | 73.2 | 45.6 |
| Unquantized (70B) | 0 | — | 16 | 3.12 | 4.97 | 77.0 | 67.9 | 81.1 | 77.7 | 51.1 |

Table 2: Zero-shot perplexities (denoted by ↓) and accuracies (↑) for LLaMa-3 8B. $B_Q = 2$ bits throughout.

| Method | Rank | $B_L(=B_R)$ | Avg Bits | Wiki2 ↓ | C4 ↓ | Wino ↑ | RTE ↑ | PiQA ↑ | ArcE ↑ | ArcC ↑ |
|---|---|---|---|---|---|---|---|---|---|---|
| CALDERA | 64 | 16 | 2.4 | 9.22 | 10.5 | 68.9 | 63.9 | 72.9 | 69.9 | 36.5 |
| CALDERA | 64 | 4 | 2.1 | 10.6 | 11.8 | 66.9 | 58.5 | 71.8 | 68.2 | 34.3 |
| CALDERA | 128 | 4 | 2.2 | 9.21 | 10.5 | 67.6 | **69.7** | 74.4 | 71.8 | 36.3 |
| CALDERA | 256 | 4 | 2.4 | **8.22** | **9.56** | **69.7** | 65.0 | **75.1** | **73.2** | **40.0** |
| QuIP# (No FT) | 64 | 16 | 2.4 | 10.9 | 11.8 | 66.5 | 57.0 | 69.6 | 63.8 | 31.0 |
| QuIP# (No FT) | 0 | — | 2 | 13.8 | 15.6 | 63.2 | 52.7 | 67.6 | 57.6 | 28.2 |
| Unquantized | 0 | — | 16 | 5.54 | 7.01 | 73.5 | 68.6 | 79.7 | 80.1 | 50.2 |

can regain the performance that was compromised when the backbone **Q** was quantized to 2 bits. Since zero-shot experiments have some inherent randomness and low-rank regularization effects [33], the zero-shot accuracies reported here are not as directly indicative of quantization performance as the perplexity results. In addition, §5.3, demonstrates that degradation in zero-shot accuracy can be recovered via LoRA fine-tuning. It is worthwhile to note these results substantiate the claims of [17], which report that low-bit quantization of LLaMa-3 8B, significantly deteriorates model performance across various post-training quantization techniques, more so than with the LLaMa-2 series.

## 5.2 Fine-tuning of Randomized Hadamard Transform (RHT) Parameters

As CALDERA presents a general optimization framework for matrix decompositions of the form **Q** + **LR**, it can easily be extended with additional heuristics to improve performance. This section serves as a proof of concept, by examining one such heuristic: Fine-tuning of randomized Hadamard transform parameters. This technique, proposed in QuIP# [36], involves fine-tuning the diagonal Rademacher matrices with ±1 entries in the RHT to minimize the cross-entropy loss between the output of the original and quantized models on the calibration dataset. Subsequently, RHT fine-tuning is performed on the models quantized using CALDERA in §5.1.[1] Details on specific fine-tuning hyperparameters can be found in App. E.4.

Perplexity and zero-shot results in Tables 3 and 4 match the trends in §5.1, i.e., CALDERA with rank-256 factors typically performs best, with the exception of RTE. In addition, perplexities are substantially lower than without the fine-tuning of randomized Hadamard transform parameters.

---

[1] Since this is primarily a proof of concept, the fine-tuning is not as extensive as in [36] due to computational limits. While [36] performs fine-tuning layer-by-layer prior to doing so on the end-to-end objective, experiments in this section only include the end-to-end fine-tuning. Furthermore, the fine-tuning is performed with a sequence length of 512, as opposed to 4096 as in [36]. As such, the QuIP# numbers reported here are different from [36].

Table 3: Zero-shot perplexities and accuracies for LLaMa-2 7B, with end-to-end fine-tuning of randomized Hadamard transform parameters. $B_Q = 2$ bits throughout. *See Footnote 1.

| Method | Rank | $B_L(= B_R)$ | Avg Bits | Wiki2 ↓ | C4 ↓ | Wino ↑ | RTE ↑ | PiQA ↑ | ArcE ↑ | ArcC ↑ |
|---|---|---|---|---|---|---|---|---|---|---|
| CALDERA | 64 | 16 | 2.4 | 6.22 | 8.23 | 64.2 | 63.2 | 76.1 | 63.4 | 34.7 |
| CALDERA | 64 | 4 | 2.1 | 6.30 | 8.32 | 64.6 | 65.7 | 75.4 | 63.3 | 35.4 |
| CALDERA | 128 | 4 | 2.2 | 6.09 | 8.06 | 65.1 | 61.0 | **76.5** | **65.1** | 35.6 |
| CALDERA | 256 | 4 | 2.4 | **5.84** | **7.75** | **65.7** | 60.6 | **76.5** | 64.6 | **35.9** |
| QuIP#* | 64 | 16 | 2.4 | 6.32 | 8.31 | 64.9 | **66.4** | 75.0 | **65.2** | 34.5 |
| QuIP#* | 0 | — | 2 | 6.58 | 8.62 | 64.4 | 53.4 | 75.0 | 64.8 | 34.0 |

Table 4: Zero-shot perplexities and accuracies for LLaMa-3 8B, with end-to-end fine-tuning of randomized Hadamard transform parameters. $B_Q = 2$ bits throughout. *See Footnote 1.

| Method | Rank | $B_L(= B_R)$ | Avg Bits | Wiki2 ↓ | C4 ↓ | Wino ↑ | RTE ↑ | PiQA ↑ | ArcE ↑ | ArcC ↑ |
|---|---|---|---|---|---|---|---|---|---|---|
| CALDERA | 64 | 16 | 2.4 | 7.63 | 8.9 | **70.3** | **70.8** | 75.4 | 72.4 | 39.0 |
| CALDERA | 64 | 4 | 2.1 | 8.06 | 9.34 | 69.5 | 64.3 | 76.0 | 71.5 | 40.0 |
| CALDERA | 128 | 4 | 2.2 | 7.76 | 9.02 | 69.4 | 63.9 | 76.0 | **73.7** | 41.8 |
| CALDERA | 256 | 4 | 2.4 | **7.34** | **8.68** | **70.3** | 70.4 | **76.5** | 73.6 | **42.3** |
| QuIP#* | 64 | 16 | 2.4 | 7.92 | 9.15 | 68.4 | 58.1 | 74.9 | 72.3 | 40.4 |
| QuIP#* | 0 | — | 2 | 8.44 | 9.75 | 67.5 | 57.8 | 72.9 | 67.6 | 37.3 |

## 5.3 Low Rank Adaptation (LoRA) Fine-tuning Results

In addition to RHT FT as described above, once the $\mathbf{Q} + \mathbf{LR}$ decomposition with target rank $k$ is obtained, and $k$ takes values $64, 128$ and $256$, fine-tuning the top $r$ ($\leq k$) singular components on a specific downstream datasets can recover the performance lost due to quantization. We consider three such tasks – (i) language modeling on Wikitext (Wiki2), (ii) recognizing textual entailment (RTE), and (iii) commonsense reasoning (WinoGrande). Throughout all experiments in Table 5, $r = 64$ is chosen and those singular components are fine-tuned to 16-bit precision, i.e., BF16 format. The tasks (ii) and (iii) are sequence classification tasks, and the pre-trained LLaMa model is augmented with a linear classification head, which is fine-tuned along with the low-rank factors [29]. In other words, the approximation is written as $\mathbf{W} \approx \mathbf{Q} + \mathbf{L}_1\mathbf{R}_1 + \mathbf{L}_2\mathbf{R}_2$, where $\mathbf{L}_1 \in \mathbb{R}^{n \times r}$, $\mathbf{L}_2 \in \mathbb{R}^{n \times (k-r)}$, $\mathbf{R}_1 \in \mathbb{R}^{r \times d}$, $\mathbf{R}_2 \in \mathbb{R}^{(k-r) \times d}$, $\mathbf{L} = [\mathbf{L}_1 \mid \mathbf{L}_2]$, $\mathbf{R}^\top = [\mathbf{R}_1^\top \mid \mathbf{R}_2^\top]$. The value of $r$ is set to 64 and $\mathbf{L}_2, \mathbf{R}_2$ are fined-tuned to $\mathbf{L}_{\text{bf16}}, \mathbf{R}_{\text{bf16}}$ using low-rank adaptation similar to [13, 16, 22]. Doing this significantly on a small task-specific dataset like WikiText2, RTE, or Winogrande, can noticeably boost the zero-shot accuracy, as can be seen from Table 5.[2]

Experimental details can be found in App. E.4. For each dataset, ten checkpoints are saved during the course of fine-tuning, and the best test performance is reported in Table 5. For datasets where test labels are not available, evaluation performance is reported instead.

For comparison, results from the LoftQ [22] and LQ-LoRA [13] papers are also reported, where available. As these papers were published before the release of LLaMa-3, only LLaMa-2 results are available.[3] In each case, CALDERA achieves better performance at a lower bit budget.

## 5.4 Autoregressive Generation Throughput

The low-rank ($\mathbf{LR}$) component in CALDERA can recover some of the accuracy lost due to the aggressive 2-bit quantization of $\mathbf{Q}$. However, CALDERA also needs to dequantize and multiply the low-rank factors, which results in a slight (albeit, acceptable) throughput degradation compared to QuIP# (shown in Table 6). Nevertheless, CALDERA's throughput is significantly higher than that of the unquantized model. This is because compressing weight matrices results in a smaller volume of data transfer from and to the GPU's SRAM, speeding up forward passes. It is worthwhile to note that

---

[2]There is an element of stochasticity in fine-tuning, so the final accuracies listed in Table 5 for different CALDERA parameters are not directly indicative of the smaller error (i.e., $\|(\mathbf{Q} + \mathbf{LR} - \mathbf{W})\mathbf{X}^\top\|_F^2$) of the initialization. Rather, they show that low-rank adaptation can recover accuracy degradation from quantization.

[3]Some LLaMa-3 results for LoftQ are present on its GitHub repository, but none for the tasks evaluated here.

Table 5: CALDERA fine-tuning results for LLaMa-2 7B and LLaMa-3 8B. $B_L$, $B_R$ are the bit-budgets of $\mathbf{L}$ and $\mathbf{R}$ for the low-rank initialization. Rank-64 fine-tuned factors are represented in BF16 precision.

| | | | | | | LLaMa-2 7B | | | LLaMa-3 8B | | |
|---|---|---|---|---|---|---|---|---|---|---|---|
| Method | Rank | $B_Q$ | $B_L(= B_R)$ | RHT FT | Avg Bits | Wiki2 ↓ | RTE ↑ | Wino ↑ | Wiki2 ↓ | RTE ↑ | Wino ↑ |
| CALDERA | 64 | 2 | 16 | No | 2.4 | 6.06 | 82.31 | 84.06 | 7.91 | 84.48 | 85.56 |
| CALDERA | 64 | 2 | 16 | Yes | 2.4 | 5.89 | 85.19 | 85.32 | 7.88 | 86.28 | 88.16 |
| CALDERA | 64 | 2 | 4 | No | 2.4 | 6.01 | 81.23 | 84.06 | 8.33 | 85.56 | 88.40 |
| CALDERA | 64 | 2 | 4 | Yes | 2.4 | 5.91 | 85.56 | 83.42 | 7.96 | **87.00** | 88.40 |
| CALDERA | 128 | 2 | 4 | No | 2.5 | 5.84 | 83.75 | 85.32 | 7.84 | 84.84 | 88.63 |
| CALDERA | 128 | 2 | 4 | Yes | 2.5 | 5.77 | 84.12 | 85.00 | 7.68 | 86.64 | 88.00 |
| CALDERA | 256 | 2 | 4 | No | 2.7 | 5.61 | 83.75 | **85.4** | **7.44** | 86.28 | 88.08 |
| CALDERA | 256 | 2 | 4 | Yes | 2.7 | **5.55** | **86.28** | 84.93 | **7.44** | 85.20 | **89.19** |
| LoftQ | 64 | 2 | 16 | — | 2.4 | 7.85 | — | — | — | — | — |
| LoftQ | 64 | 2.5 | 16 | — | 2.9 | 5.78 | — | — | — | — | — |
| LQ-LoRA | 64 | 2.75 | 8 | — | 2.95 | 5.67 | — | 72.4 | — | — | — |

Llama-2 70B runs into out-of-memory (OOM) as an A10G GPU only has 24 GiB of VRAM, which is not enough for 70B parameters in FP16 format (which approximately requires 140 GiB).

Notably, the throughput is higher when the $\mathbf{LR}$ factors are in 16-bit compared to when they are in 4-bit. This is because CALDERA used QuIP#'s lattice dequantizers for the low-rank factors as well, adding to the compute overhead. Moreover, QuIP# also used fused kernels, and CALDERA's throughput can be improved by leveraging such optimizations. Since this work is primarily motivated with the goal of closing the gap with respect to uncompressed models in the 2 to

Table 6: Throughputs for meta-llama/Llama-2-{7,70}b-hf on an NVIDIA A10G GPU for a batch size and sequence length of 1 ($B_Q = 2$ for all rows)

| Method | Rank | $B_L(= B_R)$ | Throughput (tok/s) |
|---|---|---|---|
| Uncompressed (7B, FP16) | — | — | 31.75 |
| CALDERA (7B) | 64 | 16 | 61.68 |
| CALDERA (7B) | 64 | 4 | 46.29 |
| CALDERA (7B) | 128 | 4 | 46.19 |
| CALDERA (7B) | 256 | 4 | 45.89 |
| QuIP# (7B) | 0 | — | 87.74 |
| Uncompressed (70B, FP16) | – | – | OOM |
| CALDERA (70B) | 128 | 4 | 5.33 |
| CALDERA (70B) | 256 | 4 | 4.66 |
| QuIP# (70B) | – | – | 18.18 |

2.5 bits per parameter regime, throughput improvement using custom kernels, paged attention, etc., is left for future work. We discuss the broader impacts of our work along with limitations in App. H.

## 6 Conclusions

In this work, the problem of obtaining a low-precision and low-rank decomposition of an LLM weight matrix was considered. A $\mathbf{Q} + \mathbf{LR}$ decomposition efficiently captures the high singular components of the weight matrix with sufficient fidelity, while coarsely compressing the less significant moderate-to-low singular components. An optimization-theoretically motivated algorithm was proposed to obtain this decomposition, which iteratively optimized the quantized backbone $\mathbf{Q}$ and the low-rank factors $\mathbf{L}$, $\mathbf{R}$. Additionally, it was shown that $\mathbf{L}$ and $\mathbf{R}$ can be efficiently fine-tuned using low-rank adaptation to boost the zero-shot performance of the quantized model. By utilizing a rank-constrained regression framework, an upper bound was established on the approximation error of the algorithm, and it was shown that this upper bound can be significantly smaller than prior bounds in the literature. Finally, the proposed method was empirically evaluated by compressing the LlaMA family of LLMs in the challenging sub-2.5 bits per parameter regime. The proposed approach can also be used to complement existing compression strategies; thereby making it efficient to distribute compressed LLMs and deploy them on regular consumer hardware, making them more accessible to researchers.

## Acknowledgements

This work was supported in part by the National Science Foundation (NSF) under Grant DMS-2134248, CCF-1908308; in part by the NSF CAREER Award under Grant CCF-2236829; in part by the U.S. Army Research Office Early Career Award under Grant W911NF-21-1-0242; in part by the Office of Naval Research under Grant N00014-24-1-2164; and in part by the AFOSR award #002484665.

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

# Contents

## A Notations

This section begins by outlining key notations used in both linear algebra and probability theory. Boldface uppercase and lowercase letters, such as $\mathbf{A}$ and $\mathbf{a}$, represent matrices and vectors respectively. $\mathbf{I}$ denotes the identity matrix, and its dimension is assumed to be imminent from the context. The first $k$ columns of the identity matrix $\mathbf{I}$ as $\mathbf{I}_k$, and let $\bar{\mathbf{I}}_k$ be the submatrix formed by the last $k$ columns. The singular values of $\mathbf{A}$ are denoted by $\sigma_{\max}(\mathbf{A}) = \sigma_1 \geq \sigma_2 \geq \ldots \geq \sigma_r = \sigma_{\min}(\mathbf{A})$, where $r = \mathrm{rank}(\mathbf{A})$. Similarly, the eigenvalues are denoted as $\lambda_1(\mathbf{A}), \ldots, \lambda_r(\mathbf{A})$. The max-norm of $\mathbf{A}$ is defined as $\|A\|_{\max} = \max_{i,j}|A_{ij}|$, the spectral norm of $\mathbf{A}$ is defined as $\|\mathbf{A}\|_2 = \sup_{\|\mathbf{x}\|=1}\|\mathbf{Ax}\| = \sigma_{\max}(\mathbf{A})$, and the Frobenius norm is $\|\mathbf{A}\|_{\mathrm{F}} = \left(\sum_{i,j} A_{ij}^2\right)^{1/2} = \mathrm{Tr}\left(\mathbf{A}^\top \mathbf{A}\right) = \left(\sum_{k\in[r]} \sigma_k^2\right)^{1/2}$.

For any matrix $\mathbf{X}$, its Moore-Penrose pseudo-inverse is denoted by $\mathbf{X}^\dagger$. The notations $\approx$, $\lesssim$ and $\gtrsim$ are used to denote approximate equality and inequalities that hold asymptotically in the limit when dimensions grow to infinity. In other words, for any two functions $A(n)$ and $B(n)$,

$$A(n) \lesssim B(n) \quad \text{iff} \quad \lim_{n\to\infty} A(n) \leq \lim_{n\to\infty} B(n).$$

Notations $\approx$ and $\gtrsim$ are defined analogously. Wherever relevant, dimension-dependent terms are highlighted blue.

Table 7: Notations used in this paper

| Notation | Description | Remarks |
|---|---|---|
| $\mathbf{W}$ | LLM weight matrix of any layer | $\mathbf{W} \in \mathbb{R}^{n\times d}$ |
| $\mathbf{Q}, \mathbf{L}, \mathbf{R}$ | Backbone, left and right low rank factors | $\mathbf{Q} \in \mathbb{R}^{n\times d}$, $\mathbf{L} \in \mathbb{R}^{n\times k}$, and $\mathbf{R} \in \mathbb{R}^{k\times d}$ are represented using $\mathrm{B_Q}$, $\mathrm{B_L}$, and $\mathrm{B_R}$ bits, respectively. Approximation: $\mathbf{W} \approx \mathbf{Q} + \mathbf{LR}$ |
| $\mathbf{X}$ | Calibration data matrix | $\mathbf{X} \in \mathbb{R}^{m\times d}$. Input activation for each layer. Computed once offline for each LLM. |
| $\mathbf{H}$ | $\mathbf{H} \triangleq \frac{1}{m}\mathbf{X}^\top \mathbf{X}$ | (Scaled) Hessian of least-squares objectives (1) and (3). Computed offline once. |
| $k$ | Target rank for low-rank factors | $k = 64, 128, 256$ in our expts. |
| $\mathrm{Q_Q}, \mathrm{B_Q}, \mathrm{R}$ | Quantizer, bit-budget and dynamic range for the backbone | Operates on matrices in $\mathbb{R}^{n\times d}$. |
| $\mathrm{Q_L}, \mathrm{B_L}, \mathrm{R_L}$ | Quantizer, bit-budget and dynamic range for the left low-rank factor | Operates on matrices in $\mathbb{R}^{n\times k}$. |
| $\mathrm{Q_R}, \mathrm{B_R}, \mathrm{R_R}$ | Quantizer, bit-budget and dynamic range for the right low-rank factor | Operates on matrices in $\mathbb{R}^{k\times d}$. |
| $\mathbf{M}$ | Strictly upper triangular matrix from the LDL decomposition of $m\mathbf{H}$ | $m\mathbf{H} = (\mathbf{M} + \mathbf{I})\mathbf{D}(\mathbf{M} + \mathbf{I})^\top$ |
| $\mathbf{D}$ | Diagonal matrix obtained from the LDL decomposition of $m\mathbf{H}$ | $m\mathbf{H} = (\mathbf{M} + \mathbf{I})\mathbf{D}(\mathbf{M} + \mathbf{I})^\top$ |
| $\boldsymbol{\eta}$ | Quantization error of LDLQ, given by $\boldsymbol{\eta} \triangleq \mathrm{Q_Q}(\mathbf{Q} + (\mathbf{W} - \mathbf{Q})\mathbf{M}) - (\mathbf{Q} + (\mathbf{W} - \mathbf{Q})\mathbf{M})$ | $\boldsymbol{\eta} \in \mathbb{R}^{n\times d}$ is assumed to consist of i.i.d. entries for analytical tractability. |
| $\mathbf{E_L}, \mathbf{E_R}$ | Quantization error matrices from quantizing left and right factors | $\mathbf{E_L} \in \mathbb{R}^{n\times k}$ and $\mathbf{E_R} \in \mathbb{R}^{k\times d}$ consists of zero-mean random variables with bounded variance. |

## B Rank-constrained Regression

Recall that the submatrix formed by the first $m$ columns of the identity matrix $\mathbf{I}$ is denoted as $\mathbf{I}_m$, and let $\bar{\mathbf{I}}_m$ be the submatrix formed by the last $m$ columns. The dimension of $\mathbf{I}$ is inferred depending on context. Consider the following:

$$\min_{\mathrm{rank}(\mathbf{Z})\leq k} \|\mathbf{XZ} - \mathbf{Y}\|_{\mathrm{F}}^2. \tag{7}$$

Although this problem is non-convex, it can be solved to global optimality via two SVDs. The following lemma characterizes the solution the rank-constrained regression in (5).

**Lemma B.1. (Global optimality of rank-constrained regression)** *Suppose* $\mathbf{Y} \in \mathbb{R}^{m \times n}$ *is given, and suppose* $\mathbf{X} \in \mathbb{R}^{m \times d}$ *is full rank, i.e.,* $\mathrm{rank}(\mathbf{X}) = \min\{m, d\}$, *with SVD,* $\mathbf{X} = \mathbf{U}\widetilde{\boldsymbol{\Sigma}}\mathbf{V}^{\top}$. *Furthermore, let* $\grave{\mathbf{U}}\grave{\boldsymbol{\Sigma}}\grave{\mathbf{V}}^{\top}$ *denote the full SVD of* $\mathbf{U}^{\top}\mathbf{Y}$ *if* $m \leq d$, *or the full SVD of* $(\mathbf{U}\mathbf{I}_d)^{\top}\mathbf{Y}$ *if* $m > d$. *Then, for* $k \leq m$, *the solution of* (5) *is given by*

$$\mathbf{Z}_* := \underset{\mathrm{rank}(\mathbf{Z}) \leq k}{\arg\min} \|\mathbf{X}\mathbf{Z} - \mathbf{Y}\|_{\mathrm{F}}^2 = \begin{cases} \left(\mathbf{V}\mathbf{I}_m\boldsymbol{\Sigma}^{-1}\grave{\mathbf{U}}\mathbf{I}_k\right)\left(\mathbf{I}_k^{\top}\grave{\boldsymbol{\Sigma}}\grave{\mathbf{V}}^{\top}\right) & \text{if} \quad m \leq d, \\ \left(\mathbf{V}\boldsymbol{\Sigma}^{-1}\grave{\mathbf{U}}\mathbf{I}_k\right)\left(\mathbf{I}_k^{\top}\grave{\boldsymbol{\Sigma}}\mathbf{V}^{\top}\right) & \text{otherwise.} \end{cases} \tag{8}$$

*Here,* $\boldsymbol{\Sigma}$ *is a diagonal matrix of the non-zero singular values of* $\mathbf{X}$, *defined as* $\boldsymbol{\Sigma} := \widetilde{\boldsymbol{\Sigma}}\mathbf{I}_m \in \mathbb{R}^{m \times m}$ *when* $m \leq d$, *and* $\boldsymbol{\Sigma} := \mathbf{I}_d^{\top}\widetilde{\boldsymbol{\Sigma}} \in \mathbb{R}^{d \times d}$ *when* $d < m$. *Additionally, the optimal value is*

$$\min_{\mathrm{rank}(\mathbf{Z}) \leq k} \|\mathbf{X}\mathbf{Z} - \mathbf{Y}\|_{\mathrm{F}}^2$$

$$= \|\mathbf{X}\mathbf{Z}_* - \mathbf{Y}\|_{\mathrm{F}}^2 = \begin{cases} \sum_{i=k+1}^m \sigma_i^2(\mathbf{Y}), & \text{if} \quad m \leq d, \\ \sum_{i=k+1}^m \sigma_i^2\left((\mathbf{U}\mathbf{I}_d)^{\top}\mathbf{Y}\right) + \left\|(\mathbf{U}\bar{\mathbf{I}}_{m-d})^{\top}\mathbf{Y}\right\|_{\mathrm{F}}^2, & \text{otherwise.} \end{cases} \tag{9}$$

*Proof.* **Case m $\leq$ d:** Since the full SVD of $\mathbf{X} \in \mathbb{R}^{m \times d}$ is $\mathbf{X} = \mathbf{U}\widetilde{\boldsymbol{\Sigma}}\mathbf{V}^{\top}$, where $\mathbf{U} \in \mathbb{R}^{m \times m}$, $\widetilde{\boldsymbol{\Sigma}} \in \mathbb{R}^{m \times d}$, and $\mathbf{V} \in \mathbb{R}^{d \times d}$, the last $(d - m)$ columns of $\widetilde{\boldsymbol{\Sigma}}$ will be zero, i.e., $\widetilde{\boldsymbol{\Sigma}}\bar{\mathbf{I}}_{m-d} = \mathbf{0}$. Let $\mathbf{Z}' := \mathbf{V}^{\top}\mathbf{Z} \in \mathbb{R}^{d \times n}$ be the transformed optimization variable. Since $\mathbf{V}$ is a unitary matrix, $\mathrm{rank}(\mathbf{Z}) \leq k$ if and only if $\mathrm{rank}(\mathbf{Z}') \leq k$. Splitting $\mathbf{Z}' \in \mathbb{R}^{d \times n}$ into $\mathbf{Z}'' := \mathbf{I}_m^{\top}\mathbf{Z}' \in \mathbb{R}^{m \times n}$ and $\overline{\mathbf{Z}}'' := \bar{\mathbf{I}}_{d-m}^{\top}\mathbf{Z}' \in \mathbb{R}^{(d-m) \times n}$, it can be seen that $\widetilde{\boldsymbol{\Sigma}}\mathbf{Z}' = \widetilde{\boldsymbol{\Sigma}}\mathbf{I}_d\mathbf{Z}'' + \widetilde{\boldsymbol{\Sigma}}\bar{\mathbf{I}}_{m-d}\overline{\mathbf{Z}}'' = \boldsymbol{\Sigma}\mathbf{Z}''$, where $\boldsymbol{\Sigma} := \widetilde{\boldsymbol{\Sigma}}\mathbf{I}_m \in \mathbb{R}^{m \times m}$ is a diagonal matrix comprised of the non-zero singular values of $\mathbf{X}$. Then,

$$\min_{\mathrm{rank}(\mathbf{Z}) \leq k} \|\mathbf{X}\mathbf{Z} - \mathbf{Y}\|_{\mathrm{F}}^2 \equiv \min_{\mathrm{rank}(\mathbf{Z}') \leq k} \left\|\widetilde{\boldsymbol{\Sigma}}\mathbf{Z}' - \mathbf{U}^{\top}\mathbf{Y}\right\|_{\mathrm{F}}^2 \equiv \min_{\mathrm{rank}(\mathbf{Z}'') \leq k} \left\|\boldsymbol{\Sigma}\mathbf{Z}'' - \mathbf{U}^{\top}\mathbf{Y}\right\|_{\mathrm{F}}^2. \tag{10}$$

Note that the objective function value is independent of $\overline{\mathbf{Z}}''$, as $\overline{\mathbf{Z}}''$ lies in the null space of $\mathbf{X}^{\top}$, and $\mathrm{rank}(\mathbf{Z}'') \leq k$ follows from the fact that rank of a submatrix cannot exceed the full matrix. Since $\mathbf{X}$ is full rank, $\boldsymbol{\Sigma}$ is invertible, and the minimization in (10) is equivalent to

$$\min_{\mathrm{rank}(\widetilde{\mathbf{Z}}) \leq k} \left\|\widetilde{\mathbf{Z}} - \mathbf{U}^{\top}\mathbf{Y}\right\|_{\mathrm{F}}^2 = \sum_{i=k+1}^m \sigma_i^2(\mathbf{Y}), \tag{11}$$

where $\widetilde{\mathbf{Z}} := \boldsymbol{\Sigma}\mathbf{Z}'' \in \mathbb{R}^{m \times n}$. The equality in (11) follows as a consequence of Eckart-Young-Mirsky theorem (lemma G.1), which states that $\|\widetilde{\mathbf{Z}} - \mathbf{U}^{\top}\mathbf{Y}\|_{\mathrm{F}}^2$ is minimized by taking the best rank-$k$ approximation of $\mathbf{U}^{\top}\mathbf{Y}$, and the fact that the singular values of $\mathbf{U}^{\top}\mathbf{Y}$ and $\mathbf{Y}$ are the same as $\mathbf{U}$ is unitary. Moreover, as the SVD of $\mathbf{U}^{\top}\mathbf{Y}$ is $\grave{\mathbf{U}}\grave{\boldsymbol{\Sigma}}\grave{\mathbf{V}}^{\top}$,

$$\widetilde{\mathbf{Z}}_* := \underset{\mathrm{rank}(\widetilde{\mathbf{Z}}) \leq k}{\arg\min} \left\|\widetilde{\mathbf{Z}} - \mathbf{U}^{\top}\mathbf{Y}\right\|_{\mathrm{F}}^2 = \left(\grave{\mathbf{U}}\mathbf{I}_k\right)\left(\mathbf{I}_k^{\top}\grave{\boldsymbol{\Sigma}}\grave{\mathbf{V}}^{\top}\right)$$

$$\text{and,} \quad \mathbf{Z}''_* := \boldsymbol{\Sigma}^{-1}\widetilde{\mathbf{Z}}_* = \left(\boldsymbol{\Sigma}^{-1}\grave{\mathbf{U}}\mathbf{I}_k\right)\left(\mathbf{I}_k^{\top}\grave{\boldsymbol{\Sigma}}\grave{\mathbf{V}}^{\top}\right). \tag{12}$$

In other words, if $\mathbf{Z}'_*$ denotes the solution of the middle optimization problem in (10), $\mathbf{I}_m^{\top}\mathbf{Z}_* = \mathbf{Z}''_*$. Furthermore, note that setting $\bar{\mathbf{I}}_{d-m}^{\top}\widetilde{\mathbf{Z}}_* = \mathbf{0}$ ensures $\mathrm{rank}(\mathbf{Z}'_*) = \mathrm{rank}(\mathbf{Z}''_*) \leq k$, while keeping the objective value in (10) unchanged. Hence, an optimal solution is given by

$$\mathbf{Z}'_* = \begin{bmatrix} \left(\boldsymbol{\Sigma}^{-1}\grave{\mathbf{U}}\mathbf{I}_k\right)\left(\mathbf{I}_k^{\top}\grave{\boldsymbol{\Sigma}}\grave{\mathbf{V}}^{\top}\right) \\ \mathbf{0}_{(d-m) \times n} \end{bmatrix} \quad \text{and,} \quad \mathbf{Z}_* = \mathbf{V}\mathbf{Z}'_* = \left(\mathbf{V}\mathbf{I}_m\boldsymbol{\Sigma}^{-1}\grave{\mathbf{U}}\mathbf{I}_k\right)\left(\mathbf{I}_k^{\top}\grave{\boldsymbol{\Sigma}}\grave{\mathbf{V}}^{\top}\right). \tag{13}$$

**Case m > d:** Recalling that the full SVD of $\mathbf{X} \in \mathbb{R}^{m \times d}$ is $\mathbf{X} = \mathbf{U}\widetilde{\boldsymbol{\Sigma}}\mathbf{V}^{\top}$, where $\mathbf{U} \in \mathbb{R}^{m \times m}$, $\widetilde{\boldsymbol{\Sigma}} \in \mathbb{R}^{m \times d}$, and $\mathbf{V} \in \mathbb{R}^{d \times d}$, in this case, the last $m - d$ rows of $\widetilde{\boldsymbol{\Sigma}}$ will be zero, i.e., $\bar{\mathbf{I}}_{m-d}^{\top}\widetilde{\boldsymbol{\Sigma}} = \mathbf{0}$. Denote $\mathbf{Z}' := \mathbf{V}^{\top}\mathbf{Z}$. This time, as $\boldsymbol{\Sigma} := \mathbf{I}_d^{\top}\widetilde{\boldsymbol{\Sigma}}$,

$$\min_{\mathrm{rank}(\mathbf{Z}) \leq k} \|\mathbf{X}\mathbf{Z} - \mathbf{Y}\|_{\mathrm{F}}^2 \equiv \min_{\mathrm{rank}(\mathbf{Z}') \leq k} \left\|\widetilde{\boldsymbol{\Sigma}}\mathbf{Z}' - \mathbf{U}^{\top}\mathbf{Y}\right\|_{\mathrm{F}}^2$$

$$\equiv \min_{\operatorname{rank}(\mathbf{Z}') \le k} \left\| \boldsymbol{\Sigma} \mathbf{Z}' - \mathbf{I}_d^\top \mathbf{U}^\top \mathbf{Y} \right\|_{\mathrm{F}}^2 + \left\| \bar{\mathbf{I}}_{m-d}^\top \mathbf{U}^\top \mathbf{Y} \right\|_{\mathrm{F}}^2$$

$$\equiv \min_{\operatorname{rank}(\mathbf{Z}'') \le k} \left\| \mathbf{Z}'' - \mathbf{I}_d^\top \mathbf{U}^\top \mathbf{Y} \right\|_{\mathrm{F}}^2 + \left\| \bar{\mathbf{I}}_{m-d}^\top \mathbf{U}^\top \mathbf{Y} \right\|_{\mathrm{F}}^2$$

$$\overset{(i)}{=} \sum_{i=k+1}^{m} \sigma_i^2 \left( (\mathbf{U}\mathbf{I}_d)^\top \mathbf{Y} \right) + \left\| (\mathbf{U}\bar{\mathbf{I}}_{m-d})^\top \mathbf{Y} \right\|_{\mathrm{F}}^2 , \tag{14}$$

where $\mathbf{Z}'' := \boldsymbol{\Sigma} \mathbf{Z}' = \boldsymbol{\Sigma} \mathbf{V}^\top \mathbf{Z}$. Note that the term $\left\| (\mathbf{U}\bar{\mathbf{I}}_{m-d})^\top \mathbf{Y} \right\|_{\mathrm{F}}^2$ is the irreducible error, and (i) is, once again, a consequence of the fact that $\left\| \mathbf{Z}' - \mathbf{I}_d^\top \mathbf{U}^\top \mathbf{Y} \right\|_{\mathrm{F}}^2$ is minimized by taking the best rank-$k$ approximation of $\mathbf{I}_d^\top \mathbf{U}^\top \mathbf{Y}$. Moreover, since the SVD of $\mathbf{I}_d^\top \mathbf{U}^\top \mathbf{Y}$ is $\grave{\mathbf{U}}\grave{\boldsymbol{\Sigma}}\grave{\mathbf{V}}^\top$,

$$\mathbf{Z}_*'' := \arg\min_{\operatorname{rank}(\mathbf{Z}'') \le k} \left\| \mathbf{Z}'' - \mathbf{I}_d^\top \mathbf{U}^\top \mathbf{Y} \right\|_{\mathrm{F}}^2 = \left( \grave{\mathbf{U}}\mathbf{I}_k \right) \left( \mathbf{I}_k^\top \grave{\boldsymbol{\Sigma}}\grave{\mathbf{V}}^\top \right) ,$$

$$\text{and} \quad \mathbf{Z}_* = \mathbf{V}\boldsymbol{\Sigma}^{-1}\mathbf{Z}_*'' = \left( \mathbf{V}\boldsymbol{\Sigma}^{-1}\grave{\mathbf{U}}\mathbf{I}_k \right) \left( \mathbf{I}_k^\top \grave{\boldsymbol{\Sigma}}\grave{\mathbf{V}}^\top \right) . \tag{15}$$

$\square$

**Remark**: It is not necessary for $\mathbf{X}$ to be full rank. An equivalent result can be derived with $\boldsymbol{\Sigma}^{-1}$ replaced by $\boldsymbol{\Sigma}^\dagger$.

**Computational complexity of rank-constrained regression**: Arriving at the globally optimal solution in lemma B.1 requires computing two SVDs, namely $\mathbf{X} \in \mathbb{R}^{m \times d}$, which entails a complexity of $\mathrm{O}(dm^2)$, and $\mathbf{U}^\top \mathbf{Y} \in \mathbb{R}^{m \times n}$, with a complexity of $\mathrm{O}(nm^2)$. Hence, the total computational complexity is $\mathrm{O}(m^2(n+d))$.

## C  Derivations for Calibration-Aware Low-Precision and Low-Rank Decomposition: CALDERA

### C.1  Update Equations for Low-Rank Factors in LPLRFACTORIZE submodule

As discussed in §3, the left and right low-rank factors in the LPLRFACTORIZE sub-module (Alg. 2) are found by solving least squares minimization problem. The closed form expressions can be obtained by solving the normal equations directly. In what follows, the same expressions are also derived explicitly via the singular value decomposition, as it gives an expression for the error (for example, refer to (17)) – which consists of an additive and irreducible error term.

**Updating L with fixed R**: For a fixed $\mathbf{R} \in \mathbb{R}^{k \times d}$, the left low rank factor is computed in lines 5 and 9 of Alg. 2 by solving

$$\min_{\mathbf{Z} \in \mathbb{R}^{k \times n}} \left\| \mathbf{X}\mathbf{R}^\top \mathbf{Z} - \mathbf{Y} \right\|_{\mathrm{F}}^2 , \quad \text{where } \mathbf{Y} := \mathbf{X}\mathbf{A}^\top \in \mathbb{R}^{m \times n}. \tag{16}$$

Let $\mathbf{X}\mathbf{R}^\top = \mathbf{U}_{\mathrm{XR}}\widetilde{\boldsymbol{\Sigma}}_{\mathrm{XR}}\mathbf{V}_{\mathrm{XR}}^\top$ be the full SVD of $\mathbf{X}\mathbf{R}^\top$, where $\mathbf{U}_{\mathrm{XR}} \in \mathbb{R}^{m \times m}$, $\widetilde{\boldsymbol{\Sigma}}_{\mathrm{XR}} \in \mathbb{R}^{m \times k}$, and $\mathbf{V}_{\mathrm{XR}} \in \mathbb{R}^{k \times k}$. Recall from App. B that $\mathbf{I}$ denotes the submatrix formed by the first $m$ columns of the identity matrix $\mathbf{I}_m$, and $\bar{\mathbf{I}}_m$ is the submatrix formed by the last $m$ columns. Denoting $\boldsymbol{\Sigma}_{\mathrm{XR}} := \mathbf{I}_k^\top \widetilde{\boldsymbol{\Sigma}}_{\mathrm{XR}}$ and $\mathbf{Z}' := \mathbf{V}_{\mathrm{XR}}^\top \mathbf{Z}$, since $\bar{\mathbf{I}}_{m-k}^\top \widetilde{\boldsymbol{\Sigma}}_{\mathrm{XR}} = \mathbf{0}$, it can be seen that

$$\min_{\mathbf{Z} \in \mathbb{R}^{k \times n}} \left\| \mathbf{U}_{\mathrm{XR}}\widetilde{\boldsymbol{\Sigma}}_{\mathrm{XR}}\mathbf{V}_{\mathrm{XR}}^\top \mathbf{Z} - \mathbf{Y} \right\|_{\mathrm{F}}^2 = \min_{\mathbf{X} \in \mathbb{R}^{k \times n}} \left\| \widetilde{\boldsymbol{\Sigma}}_{\mathrm{XR}}\mathbf{V}_{\mathrm{XR}}^\top \mathbf{Z} - \mathbf{U}_{\mathrm{XR}}^\top \mathbf{Y} \right\|_{\mathrm{F}}^2$$

$$= \left\| (\mathbf{U}_{\mathrm{XR}}\bar{\mathbf{I}}_{m-k})^\top \mathbf{Y} \right\|_{\mathrm{F}}^2 + \min_{\mathbf{Z}' \in \mathbb{R}^{k \times n}} \left\| \boldsymbol{\Sigma}_{\mathrm{XR}}\mathbf{Z}' - (\mathbf{U}_{\mathrm{XR}}\mathbf{I}_k)^\top \mathbf{Y} \right\|_{\mathrm{F}}^2 . \tag{17}$$

The last term of (17) is minimized by setting $\mathbf{Z}' \leftarrow \mathbf{Z}'_* := \boldsymbol{\Sigma}_{\mathrm{XR}}^{-1}(\mathbf{U}_{\mathrm{XR}}\mathbf{I}_k)^\top \mathbf{Y}$. Since $\mathbf{Z}'_* = \mathbf{V}_{\mathrm{XR}}^\top \mathbf{Z}_* = \mathbf{V}_{\mathrm{XR}}^\top \grave{\mathbf{L}}^\top$, this yields the left low rank factor to be

$$\grave{\mathbf{L}}^\top = \mathbf{V}_{\mathrm{XR}}\boldsymbol{\Sigma}_{\mathrm{XR}}^{-1}(\mathbf{U}_{\mathrm{XR}}\mathbf{I}_k)^\top \mathbf{X}\mathbf{A}^\top \implies \grave{\mathbf{L}} = (\mathbf{A}\mathbf{X}^\top)(\mathbf{R}\mathbf{X}^\top)^\dagger \overset{(i)}{=} \mathbf{A}\mathbf{H}\mathbf{R}^\top (\mathbf{R}\mathbf{H}\mathbf{R}^\top)^{-1}, \tag{18}$$

where (i) follows from the explicit expression for the pseudoinverse of the wide matrix $\mathbf{R}\mathbf{X}^\top \in \mathbb{R}^{k \times d}$.

**Updating R with fixed L**: For a fixed $\mathbf{L} \in \mathbb{R}^{n \times k}$, the right low-rank factor is computed in lines 4 and 8 of Alg. 2 by solving

$$\min_{\mathbf{Z} \in \mathbb{R}^{d \times k}} \|\mathbf{X}\mathbf{Z}\mathbf{L}^\top - \mathbf{Y}\|_{\mathrm{F}}^2, \text{ where } \mathbf{Y} := \mathbf{X}\mathbf{A}^\top \in \mathbb{R}^{m \times n}. \tag{19}$$

Let $\mathbf{X} = \mathbf{U}_{\mathrm{X}} \widetilde{\boldsymbol{\Sigma}}_{\mathrm{X}} \mathbf{V}_{\mathrm{X}}^\top$ be the full SVD of $\mathbf{X} \in \mathbb{R}^{m \times d}$, where $\mathbf{U}_{\mathrm{X}} \in \mathbb{R}^{m \times m}$, $\widetilde{\boldsymbol{\Sigma}}_{\mathrm{X}} \in \mathbb{R}^{m \times d}$, and $\mathbf{V}_{\mathrm{X}} \in \mathbb{R}^{d \times d}$. Then, denoting $\mathbf{Z}' := \mathbf{Z}\mathbf{L}^\top$, the minimization (19) is equivalent to

$$\min_{\mathbf{Z}' \in \mathbb{R}^{d \times n}} \|\widetilde{\boldsymbol{\Sigma}}_{\mathrm{X}} \mathbf{V}_{\mathrm{X}}^\top \mathbf{Z}' - \mathbf{U}_{\mathrm{X}}^\top \mathbf{Y}\|_{\mathrm{F}}^2 \equiv \min_{\mathbf{Z}'' \in \mathbb{R}^{d \times n}} \|\widetilde{\boldsymbol{\Sigma}}_{\mathrm{X}} \mathbf{Z}'' - \mathbf{U}_{\mathrm{X}}^\top \mathbf{Y}\|_{\mathrm{F}}^2, \text{ where } \mathbf{Z}'' := \mathbf{V}_{\mathrm{X}}^\top \mathbf{Z}'. \tag{20}$$

Note that (20) is an undetermined linear system with multiple solutions. In particular, since $\widetilde{\boldsymbol{\Sigma}}_{\mathrm{X}} \in \mathbb{R}^{m \times d}$, the last $(d - m)$ columns of $\widetilde{\boldsymbol{\Sigma}}_{\mathrm{X}}$ consist of zeros, i.e., $\widetilde{\boldsymbol{\Sigma}}_{\mathrm{X}} \bar{\mathbf{I}}_{d-m} = \mathbf{0}$. Therefore, a solution of this system is given by

$$\mathbf{Z}''_* = \dot{\boldsymbol{\Sigma}}^\dagger \mathbf{U}_{\mathrm{X}}^\top \mathbf{Y} \quad \text{or,} \quad \mathbf{V}_{\mathrm{X}}^\top \mathbf{Z}'_* = \widetilde{\boldsymbol{\Sigma}}_{\mathrm{X}}^\dagger \mathbf{U}_{\mathrm{X}}^\top \mathbf{Y} \quad \text{or,} \quad \mathbf{Z}'_* = \mathbf{V}_{\mathrm{X}} \widetilde{\boldsymbol{\Sigma}}_{\mathrm{X}}^\dagger \mathbf{U}_{\mathrm{X}}^\top \mathbf{Y}$$

$$\text{or,} \quad \mathbf{Z}_* = \mathbf{V}_{\mathrm{X}} \widetilde{\boldsymbol{\Sigma}}_{\mathrm{X}}^\dagger \mathbf{U}_{\mathrm{X}}^\top \mathbf{Y} (\mathbf{L}^\top)^\dagger. \tag{21}$$

This implies $\dot{\mathbf{R}} = \mathbf{L}^\dagger \mathbf{A} \mathbf{X}^\top \mathbf{U}_{\mathrm{X}} (\widetilde{\boldsymbol{\Sigma}}_{\mathrm{X}}^\dagger)^\top \mathbf{V}_{\mathrm{X}}$.

The proof is completed by noting that $\mathbf{X}^\top \mathbf{U}_{\mathrm{X}} (\widetilde{\boldsymbol{\Sigma}}_{\mathrm{X}}^\dagger)^\top \mathbf{V}_{\mathrm{X}} = \mathbf{H}\mathbf{H}^\dagger$, because,

$$\mathbf{H}\mathbf{H}^\dagger = \mathbf{X}^\top \mathbf{X} \left(\mathbf{X}^\top \mathbf{X}\right)^\dagger = \mathbf{X}^\top \mathbf{U}_{\mathrm{X}} \widetilde{\boldsymbol{\Sigma}}_{\mathrm{X}} \mathbf{V}_{\mathrm{X}}^\top \left(\mathbf{V}_{\mathrm{X}} \widetilde{\boldsymbol{\Sigma}}_{\mathrm{X}}^\top \mathbf{U}_{\mathrm{X}}^\top \mathbf{U}_{\mathrm{X}} \widetilde{\boldsymbol{\Sigma}}_{\mathrm{X}} \mathbf{V}_{\mathrm{X}}^\top\right)^\dagger$$

$$= \mathbf{X}^\top \mathbf{U}_{\mathrm{X}} \widetilde{\boldsymbol{\Sigma}}_{\mathrm{X}} \mathbf{V}_{\mathrm{X}}^\top \left(\mathbf{V}_{\mathrm{X}} \widetilde{\boldsymbol{\Sigma}}_{\mathrm{X}}^\top \widetilde{\boldsymbol{\Sigma}}_{\mathrm{X}} \mathbf{V}_{\mathrm{X}}^\top\right)^\dagger$$

$$= \mathbf{X}^\top \mathbf{U}_{\mathrm{X}} \widetilde{\boldsymbol{\Sigma}}_{\mathrm{X}} \mathbf{V}_{\mathrm{X}}^\top \mathbf{V}_{\mathrm{X}} \left(\widetilde{\boldsymbol{\Sigma}}_{\mathrm{X}}^\top \widetilde{\boldsymbol{\Sigma}}_{\mathrm{X}}\right)^\dagger \mathbf{V}_{\mathrm{X}}^\top$$

$$= \mathbf{X}^\top \mathbf{U}_{\mathrm{X}} \widetilde{\boldsymbol{\Sigma}}_{\mathrm{X}} \left(\widetilde{\boldsymbol{\Sigma}}_{\mathrm{X}}^\top \widetilde{\boldsymbol{\Sigma}}_{\mathrm{X}}\right)^\dagger \mathbf{V}_{\mathrm{X}}^\top = \mathbf{U}_{\mathrm{X}} \left(\widetilde{\boldsymbol{\Sigma}}_{\mathrm{X}}^\dagger\right)^\top \mathbf{V}_{\mathrm{X}}^\top. \tag{22}$$

## C.2 Dynamic Ranges of Quantizers

Lemmas C.1 and C.2, stated in this section provide sufficient conditions to ensure that the quantizers $\mathrm{Q_L}$ and $\mathrm{Q_R}$, used to quantize the left and right low-rank factors in the LPLRFACTORIZE submodule, remain unsaturated.

**Lemma C.1. (Dynamic range of right quantizer)** *Given matrices $\mathbf{A} \in \mathbb{R}^{n \times d}$ and $\mathbf{X} \in \mathbb{R}^{m \times d}$, let $\sigma_{\max}$ denote the maximum singular value of $\mathbf{X}\mathbf{A}^\top$. Then, the quantizer $\mathrm{Q_R}$ remains unsaturated if the dynamic range is chosen to be $\mathrm{R_R} = \sigma_{\max}$.*

*Proof.* Note that the input to the right quantizer $\mathrm{Q_R}$, i.e., $\mathbf{I}_k^\top \dot{\boldsymbol{\Sigma}} \dot{\mathbf{V}}$ satisfies $\|\mathbf{I}_k^\top \dot{\boldsymbol{\Sigma}} \dot{\mathbf{V}}^\top\|_{\max} \leq \|\mathbf{I}_k^\top \dot{\boldsymbol{\Sigma}} \dot{\mathbf{V}}^\top\|_2 = \|\mathbf{I}_k^\top \dot{\boldsymbol{\Sigma}}\|_2 = \|\dot{\boldsymbol{\Sigma}}\|_2 = \|\mathbf{U}^\top \mathbf{X}\mathbf{A}^\top\|_2 = \|\mathbf{X}\mathbf{A}^\top\|_2$. This completes the proof. $\square$

Since the input to $\mathrm{Q_L}$ is dependent on the output of $\mathrm{Q_R}$, the following lemma C.2 provides an upper bound on the input to $\mathrm{Q_L}$, provided that $\mathrm{Q_R}$ was unsaturated.

**Lemma C.2. (Dynamic range of left quantizer)** *Given matrices $\mathbf{A} \in \mathbb{R}^{n \times d}$ and $\mathbf{X} \in \mathbb{R}^{m \times n}$, let $\lambda_{\min}$ denote the smallest eigenvalue of $\mathbf{H} = \frac{1}{m}\mathbf{X}^\top \mathbf{X}$, and let $\sigma_{\max}$ and $\sigma_k$ denote the largest and the $k^{\text{th}}$ singular values of $\mathbf{X}\mathbf{A}^\top$, respectively. For some small $\epsilon > 0$ and an absolute constant $C$, suppose the number of bits for the right quantizer $\mathrm{Q_R}$ satisfies*

$$\mathrm{B_R} \geq \log_2 \left( \frac{4C\sigma_{\max}}{\sigma_k \log 2} \left( \sqrt{d} + \sqrt{k} + \sqrt{\log\left(\frac{8\|\mathbf{X}\mathbf{A}^\top\|_{\mathrm{F}}^2}{\epsilon}\right)} \right) \right). \tag{23}$$

*Then, if the dynamic range of $\mathrm{Q_L}$ is chosen to be*

$$\mathrm{R_L} = \frac{2\sigma_{\max}}{\sigma_k \sqrt{m\lambda_{\min}}}, \tag{24}$$

*then $\mathrm{Q_L}$ remains unsaturated with probability exceeding $1 - 0.25\,\epsilon\,\|\mathbf{X}\mathbf{A}^\top\|_{\mathrm{F}}^{-2}$.*

*Proof.* In Line 5 of Alg. 2, the input to $Q_L$ is $\grave{\mathbf{L}}_0 = \left(\mathbf{W}\mathbf{X}^\top\right)\left(\mathbf{R}_0\mathbf{X}^\top\right)^\dagger$. In the rest of the proof, the subscript 0 in $\grave{\mathbf{L}}_0$ and $\grave{\mathbf{R}}_0$ is dropped for brevity. Recall the notation for the full SVD of $\mathbf{X}\mathbf{R}^\top$, i.e., $\mathbf{X}\mathbf{R}^\top = \mathbf{U}_{XR}\widetilde{\boldsymbol{\Sigma}}_{XR}\mathbf{V}_{XR}$, where $\mathbf{U}_{XR} \in \mathbb{R}^{m \times m}$, $\widetilde{\boldsymbol{\Sigma}}_{XR} \in \mathbb{R}^{m \times k}$, and $\mathbf{V}_{XR} \in \mathbb{R}^{k \times k}$. From Eq. (18), $\grave{\mathbf{L}}$ can be expressed as $\grave{\mathbf{L}} = \mathbf{V}_{XR}\boldsymbol{\Sigma}_{XR}^\dagger(\mathbf{U}_{XR}\mathbf{I}_k)^\top\mathbf{X}\mathbf{A}^\top$. Consequently, $\|\grave{\mathbf{L}}\|_{\max}$ is upper bounded as

$$\|\grave{\mathbf{L}}\|_{\max} \leq \|\grave{\mathbf{L}}\|_2 \stackrel{(i)}{=} \left\|\boldsymbol{\Sigma}_{XR}^\dagger(\mathbf{U}_{XR}\mathbf{I}_k)^\top\mathbf{X}\mathbf{A}^\top\right\|_2 \stackrel{(ii)}{\leq} \left\|\boldsymbol{\Sigma}_{XR}^\dagger\right\|_2\left\|(\mathbf{U}_{XR}\mathbf{I}_k)^\top\mathbf{X}\mathbf{A}^\top\right\|_2$$
$$\stackrel{(iii)}{\leq} \left\|\boldsymbol{\Sigma}_{XR}^\dagger\right\|_2\|\mathbf{X}\mathbf{A}^\top\|_2. \tag{25}$$

Here, (i) holds because $\mathbf{V}_{XR}$ is a unitary matrix, (ii) follows from submultiplicativity of spectral norm, and (iii) follows from the fact that $(\mathbf{U}_{XR}\mathbf{I}_k)(\mathbf{U}_{XR}\mathbf{I}_k)^\top \preccurlyeq \mathbf{I}$ and lemma G.4. Furthermore,

$$\left\|\boldsymbol{\Sigma}_{XR}^\dagger\right\|_2 = \sigma_{\min}^{-1}(\boldsymbol{\Sigma}_{XR}) \stackrel{(i)}{\leq} \left((m\lambda_{\min})^{1/2}\,\sigma_{\min}(\mathbf{R})\right)^{-1}$$
$$\stackrel{(ii)}{\leq} (m\lambda_{\min})^{-1/2}\left(\sigma_{\min}\left(\mathbf{I}_k^\top\grave{\boldsymbol{\Sigma}}\grave{\mathbf{V}}^\top\right) - \|\mathbf{E}_R\|_2\right)^{-1}, \tag{26}$$

where $\sigma_{\min}(\boldsymbol{\Sigma}_{XR})$ is the smallest non-zero singular value of $\mathbf{X}\mathbf{R}^\top$, (i) follows from lemma G.4 and (ii) follows from lemma G.5. Here, $\mathbf{E}_R \in \mathbb{R}^{k \times d}$ is the quantization error from quantizing right low-rank factor, and consist of unbiased random variables with bounded variance, as described in lemma G.6. Since $\grave{\boldsymbol{\Sigma}}$ contains the singular values of $\mathbf{U}^\top\mathbf{X}\mathbf{A}^\top$,

$$\sigma_{\min}\left(\mathbf{I}_k^\top\grave{\boldsymbol{\Sigma}}\grave{\mathbf{V}}^\top\right) \stackrel{(i)}{=} \sigma_{\min}(\mathbf{I}_k^\top\grave{\boldsymbol{\Sigma}}) = \sigma_k(\grave{\boldsymbol{\Sigma}}) = \sigma_k\left(\mathbf{U}^\top\mathbf{X}\mathbf{A}^\top\right) \stackrel{(ii)}{=} \sigma_k\left(\mathbf{X}\mathbf{A}^\top\right) \tag{27}$$

where (i) and (ii) follow because $\grave{\mathbf{V}}$ and $\mathbf{U}$ are unitary matrices. Substituting (27) in (26) and (26) in (25),

$$\|\grave{\mathbf{L}}\|_{\max} \leq \left\|\boldsymbol{\Sigma}_{XR}^\dagger\right\|_2\|\mathbf{X}\mathbf{A}^\top\|_2 \leq (m\lambda_{\min})^{-1/2}\left(\sigma_k\left(\mathbf{X}\mathbf{A}^\top\right) - \|\mathbf{E}_R\|_2\right)^{-1}\|\mathbf{X}\mathbf{A}^\top\|_2 \tag{28}$$

Lemma C.1 suggests choosing the dynamic range of $Q_R$ to be $\|\mathbf{X}\mathbf{A}^\top\|_2$ to ensure that $Q_R$ remains unsaturated. As a result,

$$\|\mathbf{E}_R\|_{\max} \leq \Delta_R := \frac{2\|\mathbf{X}\mathbf{A}^\top\|_2}{2^{B_R} - 1} \implies \|\mathbf{E}_R\|_{\psi_2} \leq \frac{2\|\mathbf{X}\mathbf{A}^\top\|_2}{(2^{B_R} - 1)\log 2}, \tag{29}$$

where $\|\cdot\|_{\psi_2}$ denotes the subgaussian norm (refer to lemma G.2). Subsequently, lemma G.2 yields that for some absolute constant $C$,

$$\|\mathbf{E}_R\|_2 \leq \frac{2C\|\mathbf{X}\mathbf{A}^\top\|_2}{(2^{B_R} - 1)\log 2}\left(\sqrt{d} + \sqrt{k} + t\right) \quad \text{with probability exceeding } 1 - 2e^{-t^2}. \tag{30}$$

Setting $t = \sqrt{\log\left(\frac{8\|\mathbf{X}\mathbf{A}^\top\|_F^2}{\epsilon}\right)}$ yields

$$\|\mathbf{E}_R\|_2 \leq \frac{2C\|\mathbf{X}\mathbf{A}^\top\|_2}{(2^{B_R} - 1)\log 2}\left(\sqrt{d} + \sqrt{k} + \sqrt{\log\left(\frac{8\|\mathbf{X}\mathbf{A}^\top\|_F^2}{\epsilon}\right)}\right), \tag{31}$$

with probability exceeding $1 - \frac{\epsilon}{4\|\mathbf{X}\mathbf{A}^\top\|_F^2}$. Note that if the bit budget $B_R$ is chosen to satisfy

$$B_R \geq \log_2\left(\frac{4C\sigma_1}{\sigma_k\log 2}\left(\sqrt{d} + \sqrt{k} + \sqrt{\log\left(\frac{8\|\mathbf{X}\mathbf{A}^\top\|_F^2}{\epsilon}\right)}\right)\right), \tag{32}$$

then $\|\mathbf{E}_R\|_2 \leq \sigma_k/2$. Consequently, from (28),

$$\|\grave{\mathbf{L}}\|_{\max} \leq \frac{2\sigma_1}{\sigma_k\sqrt{m\lambda_{\min}}}. \tag{33}$$

This completes the proof. $\qquad\square$

## C.3  Proof of Lemma C.3: Approximation Error Upper Bound for LPLRFACTORIZE

**Lemma C.3. (Approximation error of LPLRFACTORIZE)** *Given* $\mathbf{A} \in \mathbb{R}^{n \times d}$ *and* $\mathbf{X} \in \mathbb{R}^{m \times d}$ *with* $m \leq d$, *let* $\lambda_{\max}$ *and* $\lambda_{\min}$ *denote the maximum and minimum eigenvalues of* $\mathbf{H} = \frac{1}{m}\mathbf{X}^\top\mathbf{X}$, *and let*

$\sigma_1 \geq \ldots \geq \sigma_k$ denote the singular values of $\mathbf{X}\mathbf{A}^\top$. Suppose the target rank $k$ satisfies

$$\frac{\lambda_{\min}^{1/2}}{4m\sigma_1\lambda_{\max}^{3/2}} \sum_{i>k} \sigma_i^2 \leq k \leq m,$$

the dynamic ranges of quantizers $Q_L$ and $Q_R$ are respectively set as

$$R_L = \frac{2\sigma_1}{\sigma_k\sqrt{m\lambda_{\min}}} \quad and \quad R_R = \sigma_1,$$

and for some absolute constant $C$ and arbitrarily small $\epsilon$ that satisfies $0 < \epsilon \leq 4mk\lambda_{\max}^2\,\lambda_{\min}^{-1}\,\sigma_1$, suppose the bit-budgets of $Q_L$ and $Q_R$ satisfy

$$B_L \geq \log_2\left(4\frac{\sigma_1^2}{\sigma_k}\sqrt{\frac{nk}{\epsilon}\frac{\lambda_{\max}}{\lambda_{\min}}} + 1\right),$$

$$and \quad B_R \geq \max\{B_1, B_2\}, \quad where \ B_1 := \log_2\left(2\sigma_1\sqrt{\frac{kd}{\epsilon}\frac{\lambda_{\max}}{\lambda_{\min}}} + 1\right),$$

$$and \quad B_2 := \log_2\left(\frac{4C\sigma_1}{\sigma_k\log 2}\left(\sqrt{d} + \sqrt{k} + \sqrt{\log\left(\frac{8\sum_i \sigma_i^2}{\epsilon}\right)}\right)\right).$$

Then, the factors $\mathbf{L}, \mathbf{R}$ returned by Alg. 1 satisfy $\mathbb{E}\left\|(\mathbf{L}\mathbf{R} - \mathbf{A})\mathbf{X}^\top\right\|_F^2 \leq \sum_{i>k} \sigma_i^2 + \epsilon$, where the expectation is over the stochasticity of quantizers $Q_L$ and $Q_R$.

*Proof.* Firstly, note that upper bounding the error $\mathbb{E}\|(\mathbf{A} - \mathbf{L}_0\mathbf{R}_0)\mathbf{X}^\top\|_F^2$ suffices, since the lines 7 to 13 in Alg. 1 refine the estimates of $\mathbf{L}$ and $\mathbf{R}$ and can only yield a smaller Frobenius norm error. Consider the quantized low rank factorization $\mathbf{A} \approx Q_L\left(\dot{\mathbf{L}}_0\right) Q_R\left(\mathbf{I}_k^\top\dot{\boldsymbol{\Sigma}}\dot{\mathbf{V}}^\top\right)$. Let

$$\mathbf{E}_L := Q_L\left(\dot{\mathbf{L}}_0\right) - \dot{\mathbf{L}}_0, \quad and \quad \mathbf{E}_R := Q_R\left(\mathbf{I}_k^\top\dot{\boldsymbol{\Sigma}}\dot{\mathbf{V}}^\top\right) - \mathbf{I}_k^\top\dot{\boldsymbol{\Sigma}}\dot{\mathbf{V}}^\top \tag{34}$$

denote the quantization error matrices. Furthermore, let

$$\xi_* \triangleq \min_{\mathrm{rank}(\mathbf{Z})\leq k} \|(\mathbf{A} - \mathbf{Z})\mathbf{X}^\top\|_F^2 \tag{35}$$

denote the optimal value of the unquantized rank-constrained regression problem. Since the dynamic ranges of quantizers $Q_R$ and $Q_L$ are chosen according to lemmas C.1 and C.2 respectively, the entries of $\mathbf{E}_L$ and $\mathbf{E}_R$ are unbiased random variables with bounded variance as in lemma G.6. Let $\Delta_R := 2R_R/(2^{B_R}-1)$, and $\Delta_L := 2R_L/(2^{B_L}-1)$ denote the quantization resolutions. Conditioned on the event that the left quantizer $Q_L$ is unsaturated, the error matrix satisfies $\mathbb{E}\mathbf{E}_L = \mathbf{0}$. This yields,

$$\mathbb{E}\left\|\left(Q_L\left(\dot{\mathbf{L}}_0\right) Q_R\left(\mathbf{I}_k^\top\dot{\boldsymbol{\Sigma}}\dot{\mathbf{V}}^\top\right) - \mathbf{A}\right)\mathbf{X}^\top\right\|_F^2$$

$$= \mathbb{E}\left\|\left(\left(\dot{\mathbf{L}}_0 + \mathbf{E}_L\right) Q_R\left(\mathbf{I}_k^\top\dot{\boldsymbol{\Sigma}}\dot{\mathbf{V}}^\top\right) - \mathbf{A}\right)\mathbf{X}^\top\right\|_F^2$$

$$\stackrel{(i)}{=} \underbrace{\mathbb{E}\left\|\left(\dot{\mathbf{L}}_0 Q_R\left(\mathbf{I}_k^\top\dot{\boldsymbol{\Sigma}}\dot{\mathbf{V}}^\top\right) - \mathbf{A}\right)\mathbf{X}^\top\right\|_F^2}_{T_1} + \underbrace{\mathbb{E}\left\|\mathbf{E}_L Q_R\left(\mathbf{I}_k^\top\dot{\boldsymbol{\Sigma}}\dot{\mathbf{V}}^\top\right)\mathbf{X}^\top\right\|_F^2}_{T_2}, \tag{36}$$

where (i) follows from the fact that the error matrix is unbiased.

Since $\dot{\mathbf{L}}_0 = \arg\min_{\mathbf{Z}\in\mathbb{R}^{k\times d}} \left\|(\mathbf{Z}\mathbf{R}_0 - \mathbf{A})\mathbf{X}^\top\right\|_F^2$, term $T_1$ can be upper bounded as

$$\mathbb{E}\left\|\left(\dot{\mathbf{L}}_0 Q_R\left(\mathbf{I}_k^\top\dot{\boldsymbol{\Sigma}}\dot{\mathbf{V}}^\top\right) - \mathbf{A}\right)\mathbf{X}^\top\right\|_F^2$$

$$\stackrel{(i)}{\leq} \mathbb{E}\left\|\left(\mathbf{V}\boldsymbol{\Sigma}^{-1}\dot{\mathbf{U}}\mathbf{I}_k Q_R\left(\mathbf{I}_k^\top\dot{\boldsymbol{\Sigma}}\dot{\mathbf{V}}^\top\right) - \mathbf{A}\right)\mathbf{X}^\top\right\|_F^2$$

$$= \mathbb{E}\left\|\left(\mathbf{V}\boldsymbol{\Sigma}^{-1}\dot{\mathbf{U}}\mathbf{I}_k\left(\mathbf{I}_k^\top\dot{\boldsymbol{\Sigma}}\dot{\mathbf{V}}^\top + \mathbf{E}_R\right) - \mathbf{A}\right)\mathbf{X}^\top\right\|_F^2$$

$$= \underbrace{\left\|\left(\mathbf{V}\boldsymbol{\Sigma}^{-1}\dot{\mathbf{U}}\mathbf{I}_k\mathbf{I}_k^\top\dot{\boldsymbol{\Sigma}}\dot{\mathbf{V}}^\top - \mathbf{A}\right)\mathbf{X}^\top\right\|_F^2}_{=\xi_*} + \mathbb{E}\left\|\mathbf{V}\boldsymbol{\Sigma}^{-1}\dot{\mathbf{U}}\mathbf{I}_k\mathbf{E}_R\mathbf{X}^\top\right\|_F^2, \tag{37}$$

where inequality (i) is obtained by replacing the minimizer $\grave{\mathbf{L}}_0$ with a different, but appropriately chosen matrix. Here, $\boldsymbol{\Sigma}$ is the diagonal matrix containing the non-zero singular values of $\mathbf{X}$.

Since $\kappa^2 := \lambda_{\max}/\lambda_{\min}$, the second term in (37) is

$$\mathbb{E}\left\|\mathbf{V}\boldsymbol{\Sigma}^{-1}\grave{\mathbf{U}}\mathbf{I}_k\mathbf{E}_{\mathrm{R}}\mathbf{X}^\top\right\|_{\mathrm{F}}^2 \overset{(i)}{=} \mathbb{E}\left\|\boldsymbol{\Sigma}^{-1}\grave{\mathbf{U}}\mathbf{I}_k\mathbf{E}_{\mathrm{R}}\mathbf{X}^\top\right\|_{\mathrm{F}}^2$$

$$= \mathbb{E}\left[\mathrm{Tr}\left(\boldsymbol{\Sigma}^{-1}\grave{\mathbf{U}}\mathbf{I}_k\mathbf{E}_{\mathrm{R}}\mathbf{X}^\top\mathbf{X}\mathbf{E}_{\mathrm{R}}^\top\mathbf{I}_k^\top\grave{\mathbf{U}}^\top\boldsymbol{\Sigma}^{-1}\right)\right]$$

$$\overset{(ii)}{\leq} m\lambda_{\max}\,\mathbb{E}\left[\mathrm{Tr}\left(\boldsymbol{\Sigma}^{-1}\grave{\mathbf{U}}\mathbf{I}_k\mathbf{E}_{\mathrm{R}}\mathbf{E}_{\mathrm{R}}^\top\mathbf{I}_k^\top\grave{\mathbf{U}}^\top\boldsymbol{\Sigma}^{-1}\right)\right]$$

$$\overset{(iii)}{=} m\lambda_{\max}\,\mathbb{E}\left[\mathrm{Tr}\left(\boldsymbol{\Sigma}^{-2}\grave{\mathbf{U}}\mathbf{I}_k\mathbf{E}_{\mathrm{R}}\mathbf{E}_{\mathrm{R}}^\top\mathbf{I}_k^\top\grave{\mathbf{U}}^\top\right)\right]$$

$$\overset{(iv)}{\leq} \kappa^2\,\mathbb{E}\left[\mathrm{Tr}\left(\mathbf{I}_k\mathbf{E}_{\mathrm{R}}\mathbf{E}_{\mathrm{R}}^\top\mathbf{I}_k^\top\right)\right]$$

$$\overset{(v)}{=} \kappa^2\,\mathbb{E}\left[\mathrm{Tr}\left(\mathbf{E}_{\mathrm{R}}\mathbf{E}_{\mathrm{R}}^\top\right)\right] = \kappa^2\,\mathbb{E}\left\|\mathbf{E}_{\mathrm{R}}\right\|_{\mathrm{F}}^2 \leq kd\,\frac{\Delta_{\mathrm{R}}^2}{4}\,\kappa^2. \tag{38}$$

Here, (i) follows since $\mathbf{V}$ is a unitary matrix, (ii) follows from lemma G.4, (iii) follows from the cyclic property of trace, (iv) follows as $\grave{\mathbf{U}}$ is unitary, and (v) follows since $\mathbf{I}_k^\top\mathbf{I}_k = \mathbf{I}$ as $k \leq m$.

Moreover, since $\mathbb{E}\mathbf{E}_{\mathrm{R}} = \mathbf{0}$, term $\mathrm{T}_2$ can be upper bounded as

$$\mathbb{E}\left\|\mathbf{E}_{\mathrm{L}}\mathbf{Q}_{\mathrm{R}}\left(\mathbf{I}_k^\top\grave{\boldsymbol{\Sigma}}\grave{\mathbf{V}}^\top\right)\mathbf{X}^\top\right\|_{\mathrm{F}}^2 = \mathbb{E}\left\|\mathbf{E}_{\mathrm{L}}\left(\mathbf{I}_k^\top\grave{\boldsymbol{\Sigma}}\grave{\mathbf{V}}^\top + \mathbf{E}_{\mathrm{R}}\right)\mathbf{X}^\top\right\|_{\mathrm{F}}^2$$

$$= \mathbb{E}\left\|\mathbf{E}_{\mathrm{L}}\mathbf{I}_k^\top\grave{\boldsymbol{\Sigma}}\grave{\mathbf{V}}^\top\mathbf{X}^\top\right\|_{\mathrm{F}}^2 + \mathbb{E}\left\|\mathbf{E}_{\mathrm{L}}\mathbf{E}_{\mathrm{R}}\mathbf{X}^\top\right\|_{\mathrm{F}}^2 \tag{39}$$

The first term in (39) is

$$\mathbb{E}\left\|\mathbf{E}_{\mathrm{L}}\mathbf{I}_k^\top\grave{\boldsymbol{\Sigma}}\grave{\mathbf{V}}^\top\mathbf{X}^\top\right\|_{\mathrm{F}}^2 = \mathrm{Tr}\left(\mathbf{E}_{\mathrm{L}}\mathbf{I}_k^\top\grave{\boldsymbol{\Sigma}}\grave{\mathbf{V}}^\top\mathbf{X}^\top\mathbf{X}\grave{\mathbf{V}}\grave{\boldsymbol{\Sigma}}^\top\mathbf{I}_k\mathbf{E}_{\mathrm{L}}^\top\right)$$

$$\overset{(i)}{\leq} m\lambda_{\max}\,\mathrm{Tr}\left(\mathbf{E}_{\mathrm{L}}\mathbf{I}_k^\top\grave{\boldsymbol{\Sigma}}\grave{\boldsymbol{\Sigma}}^\top\mathbf{I}_k\mathbf{E}_{\mathrm{L}}^\top\right)$$

$$\overset{(ii)}{\leq} m\lambda_{\max}\,\sigma_1^2\,\mathrm{Tr}\left(\mathbf{E}_{\mathrm{L}}\mathbf{E}_{\mathrm{L}}^\top\right) \leq nk\,\frac{\Delta_{\mathrm{L}}^2}{4}\,m\lambda_{\max}\,\sigma_1^2, \tag{40}$$

where (i) and (ii) follow from lemma G.4 and the fact that $\grave{\mathbf{V}}$ is a unitary matrix. Similarly, the second term of (39) is

$$\mathbb{E}\left\|\mathbf{E}_{\mathrm{L}}\mathbf{E}_{\mathrm{R}}\mathbf{X}^\top\right\|_{\mathrm{F}}^2 = \mathbb{E}\left[\mathrm{Tr}\left(\mathbf{E}_{\mathrm{L}}\mathbf{E}_{\mathrm{R}}\mathbf{X}^\top\mathbf{X}\mathbf{E}_{\mathrm{R}}^\top\mathbf{E}_{\mathrm{L}}^\top\right)\right]$$

$$\leq m\lambda_{\max}\,\mathbb{E}\left[\mathrm{Tr}\left(\mathbf{E}_{\mathrm{R}}\mathbf{E}_{\mathrm{R}}^\top\mathbf{E}_{\mathrm{L}}^\top\mathbf{E}_{\mathrm{L}}\right)\right]$$

$$= m\lambda_{\max}\,\mathbb{E}\left[\sum_{i=1}^k\left(\mathbf{E}_{\mathrm{R}}\mathbf{E}_{\mathrm{R}}^\top\right)_{ii}\left(\mathbb{E}\left[\mathbf{E}_{\mathrm{L}}\mathbf{E}_{\mathrm{L}}^\top\right]\right)_{ii}\right]$$

$$\overset{(i)}{\leq} m\lambda_{\max}\,\frac{n\Delta_{\mathrm{L}}^2}{4}\,\mathbb{E}\left[\sum_{i=1}^k\left(\mathbf{E}_{\mathrm{R}}\mathbf{E}_{\mathrm{R}}^\top\right)_{ii}\right]$$

$$= m\lambda_{\max}\,\frac{n\Delta_{\mathrm{L}}^2}{4}\,\mathbb{E}\left\|\mathbf{E}_{\mathrm{R}}\right\|_{\mathrm{F}}^2 \leq k\,\frac{n\Delta_{\mathrm{L}}^2}{4}\,\frac{d\Delta_{\mathrm{R}}^2}{4}\,m\lambda_{\max}. \tag{41}$$

Here, (i) follows because $\mathbb{E}\left[\mathbf{E}_{\mathrm{L}}^\top\mathbf{E}_{\mathrm{L}}\right]$ is a diagonal matrix since

$$\left(\mathbb{E}\left[\mathbf{E}_{\mathrm{L}}^\top\mathbf{E}_{\mathrm{L}}\right]\right)_{ij} = \sum_{l=1}^n\mathbb{E}\left[E_{li}E_{lj}\right] = \begin{cases} n\,\mathrm{Var}\left(E_{li}^2\right) \leq \frac{n\Delta_{\mathrm{L}}^2}{4} & \text{for } i = j, \\ \sum_{l=1}^n\mathbb{E}\left[E_{li}\right]\mathbb{E}\left[E_{lj}\right] = 0 & \text{for } i \neq j. \end{cases} \tag{42}$$

As a consequence of (37) to (41),

$$\mathbb{E}\left\|\left(\mathbf{Q}_{\mathrm{L}}\left(\grave{\mathbf{L}}_0\right)\mathbf{Q}_{\mathrm{R}}\left(\mathbf{I}_k^\top\grave{\boldsymbol{\Sigma}}\grave{\mathbf{V}}^\top\right) - \mathbf{A}\right)\mathbf{X}^\top\right\|_{\mathrm{F}}^2$$

$$\leq \xi_* + kd\,\frac{\Delta_{\mathrm{R}}^2}{4}\kappa^2 + nk\,\frac{\Delta_{\mathrm{L}}^2}{4}\,m\lambda_{\max}\,\sigma_1^2 + k\,\frac{n\Delta_{\mathrm{L}}^2}{4}\,\frac{d\Delta_{\mathrm{R}}^2}{4}\,m\lambda_{\max}. \tag{43}$$

Since $R_R = \sigma_1$, the second term of (43) does not exceed $0.25\epsilon$ if

$$B_R \geq \log_2\left(2\sigma_1\kappa\sqrt{\frac{kd}{\epsilon}} + 1\right). \tag{44}$$

This needs to be considered in conjunction with the lower bound on $B_R$ in lemma C.1, which yields the maximum of two quantities in the theorem statement. Since $R_L = \frac{2\sigma_1}{\sigma_k\sqrt{m\lambda_{\min}}}$, the third term of (43) does not exceed $0.25\epsilon$ if

$$B_L \geq \log_2\left(4\frac{\sigma_1^2}{\sigma_k}\sqrt{\frac{nk}{\epsilon}\frac{\lambda_{\max}}{\lambda_{\min}}} + 1\right), \tag{45}$$

provided $Q_L$ stays unsaturated. Furthermore, if $\epsilon \leq 4km\lambda_{\max}^2\,\lambda_{\min}^{-1}\,\sigma_1$, choosing $B_R$ and $B_L$ as above ensures that the third term of (43) is also upper bounded by $0.25\epsilon$.

The approximation error upper bound in (43) holds conditioned on the event that $Q_L$ was unsaturated, which according to lemma C.2, is ensured with probability exceeding $1 - 0.25\epsilon\,\|\mathbf{X}\mathbf{A}^\top\|_F^{-2}$. For analysis purposes, it is assumed that when $Q_L$ gets saturated, Alg. 1 returns $\mathbf{L} = \mathbf{0}$ and $\mathbf{R} = \mathbf{0}$, as a result of which, the approximation error is upper bounded by $\|\mathbf{X}\mathbf{A}^\top\|_F^2$.[4] Then, since $\Pr(Q_L \text{ is unsat.}) \geq 1 - 0.25\,\epsilon\,\|\mathbf{X}\mathbf{A}^\top\|_F^2$, using Cauchy-Schwarz inequality for expectations,

$$\mathbb{E}\left\|(\mathbf{L}\mathbf{R} - \mathbf{A})\mathbf{X}^\top\right\|_F^2 = \mathbb{E}\left[\left\|(\mathbf{L}\mathbf{R} - \mathbf{A})\mathbf{X}^\top\right\|_F^2 \;\middle|\; Q_L \text{ is unsat.}\right] + \mathbb{E}\left[\left\|(\mathbf{0} - \mathbf{A})\mathbf{X}^\top\right\|_F^2 \;\middle|\; Q_L \text{ is sat.}\right]$$

$$\leq \xi_* + \frac{3\epsilon}{4} + \Pr(Q_L \text{ is sat.})\sum_i \sigma_i^2 \;\leq\; \xi_* + \epsilon. \tag{46}$$

This completes the proof. $\qquad\qquad\square$

## C.4 Proof of Thm. 4.1: Approximation Error Upper Bound for CALDERA

**Theorem C.4. Approximation error of CALDERA (Formal)** *Given* $\mathbf{W} \in \mathbb{R}^{n\times d}$ *and* $\mathbf{X} \in \mathbb{R}^{m\times d}$ *with* $m \leq d$, *let* $\mathbf{D}$ *be obtained from the LDL decomposition* $\mathbf{X}^\top\mathbf{X} = m\mathbf{H} = (\mathbf{M} + \mathbf{I})\mathbf{D}(\mathbf{M} + \mathbf{I})^\top$, *and* $\lambda_{\max}, \lambda_{\min}$ *denote the max and min eigenvalues of* $\mathbf{H}$. *Additionally, let* $\mathbf{Q} \triangleq \text{LDLQ}(\mathbf{W}, Q_Q)$, *where* $Q_Q$ *has dynamic range* $R$ *and bit-budget* $B_Q$, *the quantization error be* $\boldsymbol{\eta} \triangleq Q_Q(\mathbf{Q} + (\mathbf{W} - \mathbf{Q})\mathbf{M}) - (\mathbf{Q} + (\mathbf{W} - \mathbf{Q})\mathbf{M})$, *and* $\sigma_1 \geq \ldots \geq \sigma_k \ldots$ *be the singular values of* $\mathbf{X}(\mathbf{W} - \mathbf{Q})^\top$. *If the target rank* $k$ *satisfies*

$$\frac{\lambda_{\min}^{1/2}}{4m\sigma_1\lambda_{\max}^{3/2}}\sum_{i>k}\sigma_i^2 \leq k \leq m,$$

*the dynamic ranges of* $Q_L$ *and* $Q_R$ *are set as*

$$R_L = \frac{2\sigma_1}{\sigma_k\sqrt{m\lambda_{\min}}} \quad \text{and,} \quad R_R = \sigma_1,$$

*and for some absolute constant* $C$ *and arbitrarily small* $\epsilon$ *that satisfies* $0 < \epsilon \leq 4mk\lambda_{\max}^2\,\lambda_{\min}^{-1}\,\sigma_1$, *suppose the bit-budgets of* $Q_L$ *and* $Q_R$ *are set so that they satisfy*

$$B_L \geq \log_2\left(4\frac{\sigma_1^2}{\sigma_k}\sqrt{\frac{nk}{\epsilon}\frac{\lambda_{\max}}{\lambda_{\min}}} + 1\right),$$

$$\text{and} \quad B_R \geq \max\{B_1, B_2\}, \quad \text{where } B_1 := \log_2\left(2\sigma_1\sqrt{\frac{kd}{\epsilon}\frac{\lambda_{\max}}{\lambda_{\min}}} + 1\right),$$

$$\text{and} \quad B_2 := \log_2\left(\frac{4C\sigma_1}{\sigma_k\log 2}\left(\sqrt{d} + \sqrt{k} + \sqrt{\log\left(\frac{8\sum_i\sigma_i^2}{\epsilon}\right)}\right)\right),$$

*then* $\mathbf{Q}$, $\mathbf{L}$ *and* $\mathbf{R}$ *returned by CALDERA (Alg. 1) satisfy*

$$\mathbb{E}\left\|(\mathbf{Q} + \mathbf{L}\mathbf{R} - \mathbf{W})\mathbf{X}^\top\right\|_F^2 \leq m\sum_{i>k}\mathbb{E}\lambda_i(\boldsymbol{\eta}\mathbf{D}\boldsymbol{\eta}^\top) + \epsilon \lesssim \frac{4md\lambda_{\max}R^2}{\pi(2^{B_Q} - 1)^2}\left(1 - \frac{k}{n}\right)\left(n - \frac{k}{2}\right) + \epsilon,$$

---

[4]In practice, it can be easily checked if $Q_L$ was saturated, and if so, Alg. 1 is repeated with a fresh realization of the stochastic quantizer $Q_R$. Since lemma C.2 guarantees that the saturation probability is low, it implies that "few" realizations suffice.

*where the expectation is over the stochasticity of the quantizers* $Q_Q$, $Q_L$ *and* $Q_R$.

*Proof.* As in §C.3, an upper bound is obtained on $\mathbb{E}\left\|(\mathbf{Q}_0 + \mathbf{L}_0\mathbf{R}_0 - \mathbf{W})\mathbf{X}^\top\right\|_F^2$, since after the first iteration, the alternating steps 3 and 4 in Alg. 2 simply refine the estimates of $\hat{\mathbf{Q}}$, $\mathbf{L}$ and $\mathbf{R}$, and can only yield a smaller error. For convenience of notation, in what follows, the subscript 0 is omitted.

Since $\mathbf{L}$, $\mathbf{R}$ is obtained after applying LPLRFACTORIZE submodule to $(\mathbf{W} - \mathbf{Q})$, lemma C.3 suggests that if $R_L$, $R_R$, $B_L$, and $B_R$ are chosen appropriately,

$$\mathbb{E}\|(\mathbf{Q} + \mathbf{LR} - \mathbf{W})\mathbf{X}^\top\|_F^2 = \mathbb{E}\|(\mathbf{LR} - (\mathbf{W} - \mathbf{Q}))\mathbf{X}^\top\|_F^2 \leq \sum_{i>k} \mathbb{E}\sigma_i^2\left((\mathbf{Q} - \mathbf{W})\mathbf{X}^\top\right) + \epsilon, \quad (47)$$

where the expectation in the upper bound on the right hand side is over the randomness in $\mathbf{Q}$ due to the stochasticity in quantizer $Q_Q$. Since $\mathbf{Q} = \text{LDLQ}(\mathbf{W}, Q_Q)$ is the LDLQ quantizer of Chee et al. [3] (or, BLOCKLDLQ quantizer of Tseng et al. [36], which is a successor of LDLQ proposed by Chee et al. [3]), it is shown that

$$\mathbf{Q} = Q_Q(\mathbf{Q} + (\mathbf{W} - \mathbf{Q})\mathbf{M}),$$

where $\mathbf{M}$ is a strictly upper triangular matrix. This is a consequence of the fact that LDLQ quantizes one column at a time, while simultaneously incorporating a linear feedback from the already quantized columns. Moreover, if $\boldsymbol{\eta} = Q_Q(\mathbf{W} + (\mathbf{W} - \mathbf{Q})\mathbf{M}) - (\mathbf{W} + (\mathbf{W} - \mathbf{Q})\mathbf{M})$ denotes the quantization error, then it can be seen that

$$\mathbf{Q} - \mathbf{W} = \boldsymbol{\eta}(\mathbf{M} + \mathbf{I})^{-1} \quad (48)$$

Moreover, since $m\mathbf{H} = \mathbf{X}^\top\mathbf{X} = (\mathbf{M} + \mathbf{I})\mathbf{D}(\mathbf{M} + \mathbf{I})^\top$,

$$\mathbb{E}\sigma_i^2\left((\mathbf{Q} - \mathbf{W})\mathbf{X}^\top\right) = \mathbb{E}\lambda_i\left((\mathbf{Q} - \mathbf{W})\mathbf{H}(\mathbf{Q} - \mathbf{W})^\top\right)$$

$$\overset{(i)}{=} \mathbb{E}\lambda_i\left(\boldsymbol{\eta}(\mathbf{M} + \mathbf{I})^{-1}(\mathbf{M} + \mathbf{I})\mathbf{D}(\mathbf{M} + \mathbf{I})^\top(\mathbf{M} + \mathbf{I})^{-\top}\boldsymbol{\eta}^\top\right)$$

$$= \mathbb{E}\lambda_i(\boldsymbol{\eta}\mathbf{D}\boldsymbol{\eta}^\top), \quad (49)$$

where (i) follows from (48). From (47),

$$\mathbb{E}\|(\mathbf{Q} + \mathbf{LR} - \mathbf{W})\mathbf{X}^\top\|_F^2 \leq \sum_{i>k} \mathbb{E}\lambda_i(\boldsymbol{\eta}\mathbf{D}\boldsymbol{\eta}^\top) + \epsilon. \quad (50)$$

Since $\boldsymbol{\eta}\mathbf{D}\boldsymbol{\eta}^\top = \sum_{j=1}^d D_{jj}\boldsymbol{\eta}_j\boldsymbol{\eta}_j^\top$, where $D_{jj}$ denotes the $j^{\text{th}}$ diagonal entry of $\mathbf{D}$ and $\eta_j$ is the $j^{\text{th}}$ column of $\boldsymbol{\eta} \in \mathbb{R}^{n \times d}$,

$$\lambda_i\left(\boldsymbol{\eta}\mathbf{D}\boldsymbol{\eta}^\top\right) = \lambda_i\left(\sum_{j=1}^d D_{jj}\boldsymbol{\eta}_j\boldsymbol{\eta}_j^\top\right) \leq d\ (\max_j D_{jj})\ \lambda_i\left(\frac{1}{d}\sum_{j=1}^d \boldsymbol{\eta}_j\boldsymbol{\eta}_j^\top\right). \quad (51)$$

Note that $\boldsymbol{\eta}$ consists of quantization errors, which, for uniformly dithered scalar quantization, are zero mean random variables with bounded variance upper bounded by $\frac{\Delta^2}{4} = \frac{R^2}{(2^{B_Q}-1)^2}$. As the exact error distribution for non-subtractively dithered quantization (which is commonly used in practice), is not fully understood, a simplification is made for easier analysis, assuming the quantizer is subtractively dithered. For subtractively dithered quantizers, if Schuchman's conditions are met [11, 32], this results in a uniform error distribution in the interval $\left[-\frac{\Delta}{2}, \frac{\Delta}{2}\right]$. Therefore, assuming quantization errors are independent and identically distributed, for large dimensions $d$, the eigenvalues of $\frac{1}{d}\sum_{j=1}^d \boldsymbol{\eta}_j\boldsymbol{\eta}_j^\top$ are distributed according to the Marchenko-Pastur distribution, given by $f_{\text{mp}}(x) := \frac{2d}{\pi n \Delta^2 x}\sqrt{(\lambda_+ - x)(x - \lambda_-)}$, where $\lambda_{\pm} := \frac{\Delta^2}{4}\left(1 \pm \sqrt{\frac{n}{d}}\right)^2$. Furthermore, denoting $q := F_{\text{mp}}^{-1}\left(1 - \frac{k}{n}\right)$, where $F_{mp}^{-1}$ is the inverse cumulative distribution function (CDF) of $f_{\text{mp}}(x)$, it can be seen that

$$\sum_{i>k} \mathbb{E}\lambda_i\left(\frac{1}{d}\sum_j \boldsymbol{\eta}_j\boldsymbol{\eta}_j^\top\right) \approx (n-k)\int_{\lambda_-}^q \frac{x}{2\pi(n/d)(\Delta^2/4)}\frac{\sqrt{(\lambda_+ - x)(x - \lambda_-)}}{x}dx$$

$$\overset{(i)}{\leq} \frac{\sqrt{nd}}{\pi}\left(1 - \frac{k}{n}\right)\left(F_{\text{mp}}^{-1}\left(1 - \frac{k}{n}\right) - \lambda_-\right)$$

$$\overset{(ii)}{\leq} \frac{\Delta^2}{\pi}\left(1 - \frac{k}{n}\right)\left(n - \frac{k}{2}\right). \quad (52)$$

Here, $\approx$ denotes that the equality holds asymptotically, (i) follows from upper bounding the integral using the fact that $(\lambda_+ - x)(x - \lambda_-)$ is maximized at $x = \frac{\lambda_+ + \lambda_-}{2}$. Furthermore, (ii) follows after substituting the values of $\lambda_\pm$ and using the following linear upper bound on the inverse CDF of Marchenko-Pastur distribution,

$$F_{\mathrm{mp}}^{-1}(x) \le \lambda_+ - \left(\frac{\lambda_+ - \lambda_-}{2}\right)(1 - x). \tag{53}$$

It can be verified by inspection that $F_{\mathrm{mp}}(x)$ is lower bounded by $\left(\frac{\lambda_+ - \lambda_-}{2}\right)x - \left(\frac{\lambda_+^2 - \lambda_-^2}{4}\right)$, and inequality (53) is a consequence of this fact. Furthermore, from lemma G.3, $\max_j D_{jj} \le \lambda_{\max}$. Substituting (52) in (51), and substituting $\Delta^2 = \frac{4R^2}{(2^{B_Q} - 1)^2}$, completes the proof. $\qquad\square$

**Informal version of Thm. C.4**: Consider $n \approx d$. From Thm. C.4, a (simplified) asymptotic dependence on the bit-budgets $B_L$ and $B_R$ can be obtained. In what follows, constant factors inside the $\log_2(\cdot)$ have been ignored. Comparing the expressions for $B_1$ and $B_2$, it can be seen that the desired bit-budgets are

$$B_L \ge \log_2\left(4\frac{\sigma_1^2}{\sigma_k}\sqrt{\frac{nk}{\epsilon}\frac{\lambda_{\max}}{\lambda_{\min}}}\right) \quad \text{and,} \quad B_R \ge \log_2\left(2\sigma_2\sqrt{\frac{kd}{\epsilon}\frac{\lambda_{\max}}{\lambda_{\min}}}\right).$$

For $n \approx d$, this yields an average bit-budget of

$$\frac{1}{2}(B_L + B_R) = \frac{1}{2}\log_2\left(\frac{8k}{\epsilon}\frac{\sigma_1^3}{\sigma_k}\frac{\lambda_{\max}}{\lambda_{\min}}\sqrt{nd}\right) \quad \text{bits per parameter.}$$

## C.5 A simplification for positive definite Hessians

The expressions in (8) are a little involved, but they can be simplified if $\mathbf{H}$ is positive definite, i.e., all eigenvalues are strictly greater than 0. Let $\mathbf{H} = \mathbf{U}\boldsymbol{\Lambda}\mathbf{U}^\top$ be the eigenvalue decomposition of $\mathbf{H}$, and let $\mathbf{H}^{\frac{1}{2}} = \mathbf{U}\boldsymbol{\Lambda}^{\frac{1}{2}}\mathbf{U}^\top$ be its symmetric square root. Furthermore assume that $\mathbf{H}$ is positive definite, i.e., all diagonal entries of $\boldsymbol{\Lambda}$ are strictly positive. In practice, this can be ensured by regularizing $\mathbf{H}$ through the addition of an identity matrix, scaled by a small amount.

Then, consider the following equivalent optimization problems:

$$\min_{\mathrm{rank}(\mathbf{Z}) \le k} \left\|(\mathbf{A} - \mathbf{Z})\mathbf{X}^\top\right\|_F^2 \equiv \min_{\mathrm{rank}(\mathbf{Z}) \le k} \mathrm{Tr}\left((\mathbf{A} - \mathbf{Z})\mathbf{H}(\mathbf{A} - \mathbf{Z})^\top\right)$$

$$\equiv \min_{\mathrm{rank}(\mathbf{Z}) \le k} \left\|(\mathbf{A} - \mathbf{Z})\mathbf{H}^{\frac{1}{2}}\right\|_F^2$$

$$\equiv \min_{\mathrm{rank}(\mathbf{Z}) \le k} \left\|(\mathbf{A} - \mathbf{Z})\mathbf{U}\boldsymbol{\Lambda}^{\frac{1}{2}}\right\|_F^2$$

$$\equiv \min_{\mathrm{rank}(\mathbf{Z}) \le k} \left\|\mathbf{Y} - \mathbf{Z}\mathbf{U}\boldsymbol{\Lambda}^{\frac{1}{2}}\right\|_F^2 \quad (\text{where } \mathbf{Y} \triangleq \mathbf{A}\mathbf{U}\boldsymbol{\Lambda}^{\frac{1}{2}})$$

$$\equiv \min_{\mathrm{rank}(\mathbf{Z}') \le k} \left\|\mathbf{Y} - \mathbf{Z}'\right\|_F^2 \tag{54}$$

Here, $\mathbf{Z}' \triangleq \mathbf{Z}\mathbf{U}\boldsymbol{\Lambda}^{\frac{1}{2}}$, and the constraint $\mathrm{rank}(\mathbf{Z}') \le k$ follow from the fact that multiplying $\mathbf{Z}$ by an invertible matrix, $\mathbf{U}\boldsymbol{\Lambda}^{\frac{1}{2}}$, keeps its rank unchanged. The final optimization problem can be solved optimally by considering the SVD of $\mathbf{Y}$ as $\mathbf{Y} = \mathbf{U}\boldsymbol{\Sigma}\mathbf{V}^\top$, and the optimal solution is

$$\mathbf{Z}'_* = (\mathbf{U}\mathbf{I}_k)\left(\mathbf{I}_k^\top\boldsymbol{\Sigma}\mathbf{V}^\top\right), \tag{55}$$

where $\mathbf{I}_k$ denotes the first $k$ columns of the identity matrix. So the final solution is given by,

$$\mathbf{Z}_* = (\mathbf{U}\mathbf{I}_k)\left(\mathbf{I}_k^\top\boldsymbol{\Sigma}\mathbf{V}^\top\boldsymbol{\Lambda}^{-\frac{1}{2}}\mathbf{U}^\top\right). \tag{56}$$

This yields a closed form expression for the optimal solution of the rank-constrained regression problem for the case of positive definite Hessians.

# D    Computational Complexity of CALDERA

**Complexity of pre-preprocessing**: For a given calibration dataset/activation $\mathbf{X} \in \mathbb{R}^{m \times d}$, the Hessian is computed as $\mathbf{H} = \frac{1}{m}\mathbf{X}^\top\mathbf{X}$, which requires $\mathrm{O}(md^2)$ compute. Subsequently, the LDL decomposition of $\mathbf{H}$ is computed, which entails $\mathrm{O}(d^3)$ complexity [20]. Moreover, $\mathbf{HH}^\dagger$, which is used in the update equation of $\grave{\mathbf{R}}$ is also computed beforehand, with a complexity of $\mathrm{O}(d^3)$. therefore, the total complexity of pre-processing is $\mathrm{O}(md^2 + 2d^3)$.

**Complexity of LDLQ**: Each iteration of CALDERA invokes a single call to LDLQ. An LDLQ call involves updating each column of an $n \times d$ iteratively. From (2), updating the $k^{\text{th}}$ column requires multiplying a matrix (of already quantized columns) in $\mathbb{R}^{n \times (k-1)}$ with a vector in $\mathbb{R}^{k-1}$, implying a complexity of $\mathrm{O}(n(k - 1))$. Subsequently, if $\mathrm{Q}_\mathrm{Q}$ is uniform scalar quantization, quantizing the (feedback incorporated) $k^{\text{th}}$ column entails $\mathrm{O}(n)$ compute, implying a total of $\mathrm{O}(nk)$ compute for the $k^{\text{th}}$ column. Hence, quantizing all columns from $k = 1, \ldots, n$ would require $\mathrm{O}\left(n \sum_{k=1}^{n} k\right) = \mathrm{O}\left(\frac{n^2}{2} + \frac{3n}{2}\right) = \mathrm{O}(n^2)$ compute.

**Complexity of LPLRFACTORIZE**: Every iteration of CALDERA invokes a call to LPLRFACTORIZE, which itself consists of $\mathrm{T_{in}}$ inner iterations. Each inner iteration of LPLRFACTORIZE, consists of initializing the left and right low rank factors using rank-constrained regression. As seen in §B, it has a computational complexity of $\mathrm{O}\left(m^2(n + d)\right)$. Subsequently, for any matrix $\mathbf{A}$, the left low rank factor as $\grave{\mathbf{L}} = \mathbf{AHR}^\top(\mathbf{RHR}^\top)^{-1}$ can be found using successive matrix multiplications. While a gradient descent based algorithm can possibly speed up this computation, in implementation, closed form expressions are computed directly as they are computationally affordable for the hidden dimensions of the LLMs considered. Computing $\mathbf{A}\left(\mathbf{HR}^\top\right)$ requires $\mathrm{O}\left(dk(n + d)\right)$, computing $(\mathbf{RHR}^\top)^{-1}$ entails $\mathrm{O}\left(dk(k + d) + k^3\right)$, and multiplying them together requires $\mathrm{O}\left(nk^2\right)$. Keeping the dominant term, the aggregated computational complexity for computing the left low-rank factor in each inner iteration is $\mathrm{O}\left(ndk\right)$.

Additionally, the left low-rank factor can be computed as $\grave{\mathbf{R}} = \mathbf{L}^\dagger\mathbf{A}(\mathbf{HH}^\dagger)$, where computing $\mathbf{L}^\dagger\mathbf{A}$ is $\mathrm{O}\left(ndk\right)$. Since $\mathbf{HH}^\dagger$ is already computed beforehand, multiplying $\mathbf{L}^\dagger\mathbf{A}$ and $\mathbf{HH}^\dagger$ entails $\mathrm{O}\left(kd^2\right)$. Once again, keeping the dominant term, the complexity is $\mathrm{O}\left(ndk\right)$. Hence, the total complexity of each inner iteration of LPLRFACTORIZE is $\mathrm{O}\left(ndk\right)$.

**Remark**: Compared to the two-stage LPLR algorithm proposed in [31], the left and right low-rank factors are iteratively refined in Alg. 2 since this additional compute can be afforded for quantizing LLM weight matrices. Moreover, SVD computations can be sped up with randomized SVD [14].

# E    Additional Experimental Details

## E.1    System Details

The code for this paper is available at https://github.com/pilancilab/caldera. The framework is built in PyTorch on top of the QuIP# [36] and LoftQ [22] repositories, and utilizes HuggingFace implementations of all datasets and LLMs. Experiments were performed on either NVIDIA RTX A6000, NVIDIA A10G, or NVIDIA H100 GPUs. Hessian computation for all but LLaMa-2 70B was performed on four NVIDIA A10 GPUs (provisioned as Amazon AWS EC2 instances), parallelized by distributing different layers to different GPUs. Hessian computation for LLaMa-2 70B was performed on a single H100 GPU. Model quantization was performed on four NVIDIA RTX A6000 GPUs, parallelized in the same manner as Hessian computation. All zero-shot and fine-tuning experiments were also run on A6000 GPUs, except for LLaMa-2 70B zero-shot experiments, which were run on an H100 GPU. Low-rank adaptation experiments were parallelized across four GPUs via HuggingFace Accelerate [12], with DeepSpeed ZeRO Level 2 [28, 30]. The use of Llama family of LLMs for research is guided by a community license that can be founded at https://llama.meta.com/license/ and https://llama.meta.com/llama3/license/.

## E.2  Parameters of CALDERA Quantization

For all CALDERA decompositions, the number of alternating iterations between updating $\mathbf{Q}$ and $\mathbf{L}$, $\mathbf{R}$ (i.e., $T_{\text{out}}$ in Alg. 1) is 15. For decompositions with quantized low-rank factors, except LLaMa-2 7B and LLaMa-3 8B, the number of LPLR iterations (i.e., $T_{\text{in}}$ in Alg. 2) is 10. For LLaMa-2 7B and LLaMa-3 8B, the number of LPLR iterations is 50.

For the LLaMa-2 7B and LLaMa-3 8B models (for both CALDERA and QuIP# decompositions), the calibration dataset consists of 256 random samples from the RedPajama dataset. For all other models, the calibration dataset provided by the QuIP# [36] authors was used. This calibration dataset consists of 6144 random samples from RedPajama.

The number of tokens in each calibration data point is equal to the context length of the corresponding model, i.e., 4096 for LLaMa-2 and 8192 for LLaMa-3 8B.

An additional heuristic applied during the CALDERA decomposition is an update to the Hessian matrix based on the subspace spanned by $\mathbf{LR}$:

$$\widetilde{\mathbf{H}} \triangleq \mathbf{H} - \mathbf{MVV}^\top \mathbf{M}^\top, \tag{57}$$

Where $\mathbf{H} = \mathbf{MM}^\top$ is the LDL decomposition of the Hessian and $\mathbf{LRM} = \mathbf{U\Sigma V}^\top$ is the singular value decomposition of the product $\mathbf{LRM}$. The updated $\widetilde{\mathbf{H}}$ is used in all LPLR updates, and the original Hessian is used for updating the low-rank factors via Alg. 2. This heuristic was found in the QuIP# codebase, and empirically speeds up convergence of CALDERA, as is discussed in App. F.1.

Although there is randomness in the generation of the randomized Hadamard matrices, we found that it has negligible effect on the results, due to the iterative nature of the CALDERA algorithm. As such, the effect of random seeds is minimal.

## E.3  Details about Goodness-of-Fit Metrics

In addition to perplexity on the WikiText2 and C4 datasets, CALDERA was evaluated using zero-shot accuracy on the following sequence classification tasks:

1. **Winogrande** [19]: a collection of 44k problems problems inspired by the Winograd Schema Challenge [21], specifically designed to be robust to learned biases in model training datasets. Specifically, Winogrande is a two-choice fill-in-the-blank task that requires significant commonsense reasoning.

2. **RTE** [1]: Recognizing Text Entailment is a task in the General Language Understanding Evaluation (GLUE) benchmark [40]. Given two statements, the model must classify whether or not the second follows from the first.

3. **PiQA** [2]: Physical Interaction: Question Answering is a question-answering task that requires understanding about physical relationships between objects.

4. **ArcE** [4]: In general, the AI2 Reasoning Challenge is a set of multiple-choice questions that require a grade-school level of knowledge. ArcE is the subset that is labeled Easy.

5. **ArcC** [4]: The ARC-Challenge problems are in the same format as ARC-Easy, but the dataset only includes problems that were answered incorrectly by selected algorithms.

The unquantized (16-bit) numbers mentioned in Tabs. 1 and 2 are either copied from [36] for the tasks available reported therein. For some ones that could not be easily found, inference was performed directly on the 16-bit model downloaded from HuggingFace.

## E.4  Fine-tuning Parameter Details

Fine-tuning of the diagonal Rademacher matrices in the RHT was performed over a calibration dataset sampled from the training split of RedPajama. Due to computational constraints, each sample contains 512 tokens, rather than the full context length of the model. The calibration dataset is 256 samples in total, with 192 data points in the training split and 64 in the evaluation split. RHT fine-tuning was performed for 5 epochs with a learning rate of $10^{-3}$. The epoch with the lowest evaluation loss was used for further zero-shot and fine-tuning experiments.

Parameters for low-rank adaptation experiments can be found in Table 8. Overall, the number of epochs and therefore number of training steps, was determined empirically based on when the evaluation accuracy plateaued.

Table 8: Hyperparameter settings for low-rank adaptation[*]. Batch size refers to the per-device batch size. All fine-tuning experiments are parallelized across four GPUs.

| Dataset | Block Size | Batch Size[*] | Grad. Acc. | Epochs | LR | Weight Decay | LR Sched. | Warmup | Steps |
|---|---|---|---|---|---|---|---|---|---|
| Wikitext2 | 512 | 2 | 1 step | 3 | 3e-6 | 0.001 | Linear | 42 steps | 2091 |
| RTE | 128 | 20 | 2 steps | 10 | 3e-5 | 0.01 | Linear | 100 steps | 160 |
| Winogrande | 256 | 10 | 1 step | 3 | 1e-5 | 0.01 | Linear | 100 steps | 3030 |

For the comparison of CALDERA to LQ-LoRA in Winogrande accuracy, the LQ-LoRA result reported in [13] did not involve fine-tuning directly on the Winogrande dataset, but rather on a subset of C4 and Wikitext2.

### E.5 Computation of Average Bit Budget Per-Parameter

Assume that the dimensions of the seven weight matrices in each transformer layer have dimensions $n_i \times d_i$, where $i \in \{1, \ldots, 7\}$. Also, for the models we consider, each transformer layer has matrices of the same dimensions. In general, if the full-rank matrix $\mathbf{Q}$ has a bit budget of $B_\mathrm{Q}$, the factors $\mathbf{L}$ and $\mathbf{R}$ both have a bit budget of $B_\mathrm{LR}$, and the rank of the factors is $k$, the average number of bits per parameter is

$$\frac{\sum_{i=1}^{7} \left( B_\mathrm{Q} n_i d_i + k B_\mathrm{LR}(n_i + d_i) \right)}{\sum_{i=1}^{7} n_i d_i}.$$

For LLaMa-2 7B, all self-attention matrices are $4096 \times 4096$, the `gate` and `up` projections are $11008 \times 4096$, and the `down` projection is $4096 \times 11008$. So, $(n_i, d_i) = (4096, 4096)$ for $i \in \{1, \ldots, 4\}$, $(n_5, d_5) = (n_6, d_6) = (11008, 4096)$, and $(n_7, d_7) = (4096, 11008)$.

For LLaMa-3 8B, $(n_1, d_1) = (n_4, d_4) = (4096, 4096)$, $(n_2, d_2) = (n_3, d_3) = (4096, 1024)$, $(n_5, d_5) = (n_6, d_6) = (14336, 4096)$, and $(n_7, d_7) = (4096, 14336)$.

In §5.3, $r = 64$ of the $k$ factors are fine-tuned in 16-bit precision. This results in the following number of bits per parameter:

$$\frac{\sum_{i=1}^{7} \left( B_\mathrm{Q} n_i d_i + (k - r) B_\mathrm{LR}(n_i + d_i) + 16r(n_i + d_i) \right)}{\sum_{i=1}^{7} n_i d_i}.$$

### E.6 Additional Notes on Numerical Simulations

For LLaMa-2 70B, the zero-shot perplexities and accuracies reported in the QuIP# paper [36] could not be replicated. As such, the QuIP# performance reported in this paper are based on recreations of those experiments.

## F Additional Numerical Simulations

### F.1 Ablation Study of CALDERA Parameters with respect to Frobenius Norm Error

The per-iteration relative Frobenius norm error of CALDERA, for a few selected weight matrices of LLaMa-2 7B, is plotted in Figure 3. The data-aware relative Frobenius norm error is defined as

$$\frac{\left\| (\mathbf{Q} + \mathbf{LR} - \mathbf{W})\mathbf{X}^\top \right\|_\mathrm{F}}{\|\mathbf{W}\mathbf{X}^\top\|_\mathrm{F}},$$

where $\mathbf{W}$ is the unquantized weight matrix, $\mathbf{X}$ is the calibration data, and $\mathbf{Q} + \mathbf{LR}$ is the CALDERA decomposition. The following ablations of CALDERA decomposition parameters are considered:

1. **16-Bit Factors**: $\mathbf{Q}$ is quantized to 2 bits via LDLQ. The low-rank factors are kept in half precision, so no LPLR iterations are required after rank-constrained regression is performed.

The randomized Hadamard transform is performed on the $\mathbf{W}$ and $\mathbf{H}$ matrices are prior to quantization.

2. **4-Bit Factors**: The low-rank factors are quantized to 4 bits of precision using an E8 lattice quantizer, and 10 iterations of LPLRFACTORIZE (Alg. 2) are run for each update of the factors (i.e., $T_{\text{in}} = 10$ in Alg. 1). Otherwise, the decomposition parameters are the same as in **16-Bit Factors**.

3. **4-Bit Factors (No RHT)**: The parameters of CALDERA are the same as in **4-Bit Factors**, except the randomized Hadamard transform step is omitted.

4. **4-Bit Factors (No LDLQ)**: The parameters of CALDERA are the same as in **4-Bit Factors**, except $\mathbf{Q}$ is quantized to 2-bit precision using direct E8 lattice quantization instead of the LDLQ algorithm.

5. **16-Bit Factors (Hessian Update)**: The parameters of CALDERA are the same as in **16-Bit Factors**, except the Hessian update step described in App. E.2 is performed prior to LDLQ.

For each ablation, the rank of $\mathbf{L}$ and $\mathbf{R}$ varies between $k \in \{64, 128, 256\}$. For comparison, QuIP# with the same-size low-rank factors is performed, using the same calibration data and the low-rank decomposition from the QuIP# codebase.

Overall, CALDERA with 16-bit factors consistently achieves a lower error than QuIP#. Additionally, the Hessian update heuristic improves convergence and often, but not always, reduces the final error achieved after 50 iterations of Alg. 1. CALDERA with 4-bit factors has higher Frobenius-norm error than with 16-bit factors, but the degradation is minor. On the other hand, replacing LDLQ with a lattice quantizer significantly degrades the Frobenius norm error, and omitting the randomized Hadamard transform worsens the error for some of the weight matrices considered.

To demonstrate the convergence of LPLRFACTORIZE (Alg. 2), the per-iteration relative data-aware error of LPLRFACTORIZE for the factorization $\mathbf{W} \approx \mathbf{LR}$ is plotted in Figure 4. For all curves plotted, the randomized Hadamard transform is performed on $\mathbf{W}$ and $\mathbf{H}$ before the factorization is computed, and both factors are quantized to 4 bits of precision via an E8 lattice quantizer.

In both cases, the alternating minimization iterations reduce the error. For the Up projection matrix, this reduction is nominal, whereas, for the Key projection matrix, the alternating iterations result in a significant improvement.

## F.2 Experiments on Mistral-7B

Perplexity and language modeling benchmark accuracy results for quantizing Mistral 7B via CALDERA and QuIP# are in Table 9. Results are consistent with those in Section 5.1.

Table 9: Evaluations of Wikitext2 and C4 perplexities, as well as percent accuracies on some common language modeling benchmarks, on CALDERA-compressed Mistral 7B. All quantizations use calibration datasets released on Huggingface by the authors of QuIP#. $B_Q = 2$ bits throughout, and $B_L = B_R = 4$ bits where low-rank factors are present. For fairness of comparison, QuIP# numbers reported do not include RHT finetuning.

| Model | Method | Rank | Avg Bits | Wiki2 ↓ | C4 ↓ | Wino ↑ (acc) | PiQA ↑ (acc_norm) | ArcE ↑ (acc_norm) | ArcC ↑ (acc_norm) |
|-------|--------|------|----------|---------|------|------|------|------|------|
| Mistral 7B | CALDERA | 128 | 2.19 | 6.01 | 8.92 | **71.51** | 78.02 | **75.21** | 46.59 |
| Mistral 7B | CALDERA | 256 | 2.38 | **5.71** | **8.44** | 70.24 | **79.92** | 74.92 | **46.93** |
| Mistral 7B | QuIP# | 0 | 2 | 6.19 | 9.09 | 69.30 | 78.45 | 72.90 | 44.80 |

# G    Auxiliary Results

## G.1    Useful Results from Linear Algebra and Probability

This section states some useful linear algebra results as lemmas. Some of them are stated without proof, which can be easily proved or found in a linear algebra textbook such as [10].

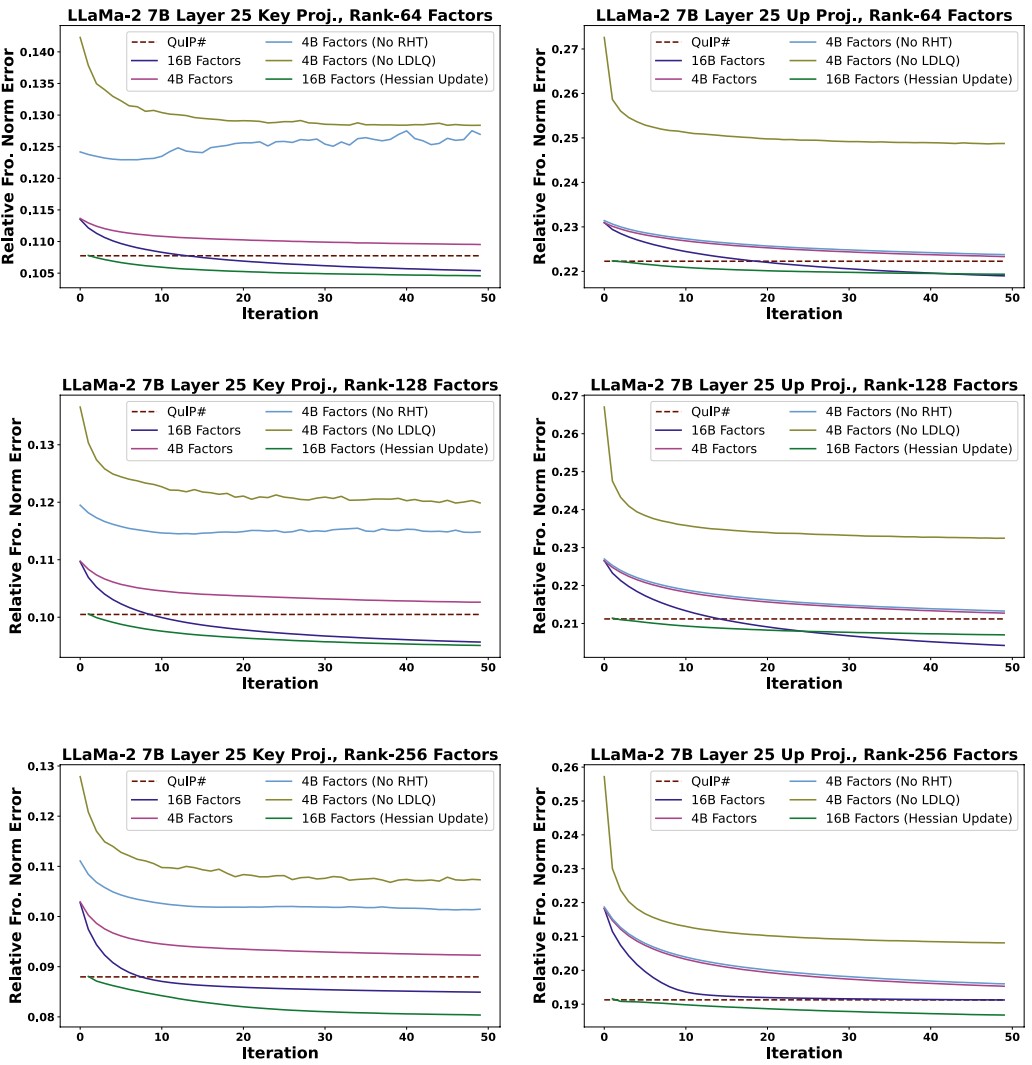

Figure 3: Relative data-aware Frobenius norm error per iteration of CALDERA for selected matrices of LLaMa-2 7B layer 25. For all experiments, the bit precision of **Q** is 2, and the calibration dataset is the same as used in §5. The first iteration of CALDERA with the Hessian update is omitted, as it has a large error, inhibiting plot readability.

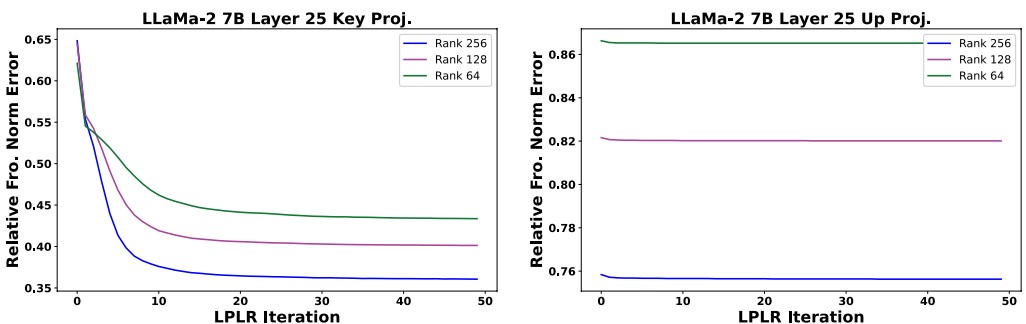

Figure 4: Relative data-aware Frobenius norm error per iteration of LPLRFACTORIZE, for the decomposition $\mathbf{W} \approx \mathbf{L}\mathbf{R}$, for two matrices in LLaMa-2 7B layer 25.

**Lemma G.1. (Eckart-Young-Mirsky theorem)** *For any matrix $\mathbf{A}$ with SVD given by $\mathbf{A} = \mathbf{U}\boldsymbol{\Sigma}\mathbf{V}^\top$, the solution of $\mathbf{A}_k := \arg\min_{\mathrm{rank}(\widehat{\mathbf{A}}) \leq k} \|\mathbf{A} - \widehat{\mathbf{A}}\|_{\mathrm{F}}^2$ is given by $\mathbf{A}_k = \left(\mathbf{U}\boldsymbol{\Sigma}^{1/2}\mathbf{I}_k\right)\left(\mathbf{I}_k^\top \boldsymbol{\Sigma}^{1/2}\mathbf{V}^\top\right)$, and $\|\mathbf{A} - \mathbf{A}_k\|_{\mathrm{F}}^2 = \sum_{i>k}\sigma_i^2(\mathbf{A})$.*

**Lemma G.2.** *(Vershynin [38, Thm 4.4.5]) Let $\mathbf{X}$ be a $d \times m$ random matrix whose entries $X_{ij}$ are independent, zero-mean, subgaussian random variables. Then, for any $t > 0$, $\|\mathbf{X}\|_2 \leq CK\left(\sqrt{d} + \sqrt{m} + t\right)$ with probability exceeding $1 - 2e^{-t^2}$, where $K = \max_{i,j}\|A_{ij}\|_{\psi_2}$, and $\|\cdot\|_{\psi_2}$ denotes the subgaussian norm, and $C$ is an absolute constant.*

**Remark**: Lemma G.2 states a high probability upper bound on the spectral norm of a random matrix in terms of the subgaussian norm of the entries of the matrix. The subgaussian norm of a subgaussian random variable $X$ is defined as $\|X\|_{\psi_2} \triangleq \inf\{t \geq 0 \mid \mathbb{E}[e^{X^2/t^2}] \leq 2\}$. It can be shown that any bounded random variable $X$ is subgaussian, and satisfies $\|X\|_{\psi_2} \leq \frac{\|X\|_\infty}{\log 2}$.

**Lemma G.3.** *Suppose the LDL decomposition of a matrix $\mathbf{H} \in \mathbb{R}^{d \times d}$ is given by $\mathbf{H} = (\mathbf{M} + \mathbf{I})\mathbf{D}(\mathbf{M} + \mathbf{I})^\top$, where $\mathbf{M}$ is strictly upper triangular and $\mathbf{D}$ is diagonal. Then, $\max_j D_{jj} \leq \lambda_{\max}$, where $\lambda_{\max}$ denotes the largest eigenvalue of $\mathbf{H}$.*

*Proof.* Let $\mathbf{e}_j$ denote the $j^{\mathrm{th}}$ canonical basis vector. Then,

$$\mathbf{e}_j^\top \mathbf{H}\mathbf{e}_j = \mathbf{e}_j^\top(\mathbf{M} + \mathbf{I})\mathbf{D}(\mathbf{M} + \mathbf{I})^\top \mathbf{e}_j \overset{(\mathrm{i})}{=} D_{jj} + \sum_{i>j}M_{ji}^2. \tag{58}$$

Moreover, from properties of eigenvalues, $\mathbf{e}_j^\top \mathbf{H}\mathbf{e}_j \leq \lambda_{\max}$. Therefore, for any $j$, $D_{jj} \leq D_{jj} + \sum_{i>j}M_{ji}^2 \leq \lambda_{\max}$. Since it holds true for all $j$, this completes the proof. $\qquad\square$

**Lemma G.4. (Loewner ordering for matrix products)** *For any matrix $\mathbf{A}$ and $\mathbf{B}$,*
$$\sigma_{\min}^2(\mathbf{A})\,\mathbf{B}^\top\mathbf{B} \preccurlyeq \mathbf{B}^\top\mathbf{A}^\top\mathbf{A}\mathbf{B} \preccurlyeq \sigma_{\max}^2(\mathbf{A})\,\mathbf{B}^\top\mathbf{B}.$$

**Lemma G.5. (Minimum singular value)** *For matrices $\mathbf{A}$ and $\mathbf{B}$, $\sigma_{\min}(\mathbf{A} + \mathbf{B}) \geq \sigma_{\min}(\mathbf{A}) - \|\mathbf{B}\|_2$.*

### G.2 Uniformly dithered scalar quantizer

Let us consider quantizing a scalar $x$ with $|x| \leq \mathrm{R}$. Given B bits, the scalar quantizer with *dynamic range* R is described by first specifying the $M = 2^{\mathrm{B}}$ quantization points
$$q_1 = -\mathrm{R}, q_2 = -\mathrm{R} + \Delta, q_3 = -\mathrm{R} + 2\Delta, \ldots, q_M = -\mathrm{R} + (M-1)\Delta,$$
where $\Delta = \frac{2\mathrm{R}}{M-1}$ is the resolution. The quantizer operation is defined as:
$$\mathrm{Q}_{\mathrm{R},\mathrm{B}}(x) = \begin{cases} q_{k+1} & \text{with probability } r, \\ q_k & \text{with probability } 1 - r, \end{cases} \tag{59}$$
where $k = \arg\max_j\{q_j \leq x\}$, i.e., $x \in [q_k, q_{k+1})$, and $r = \frac{x - q_k}{\Delta}$. As shown in the following lemma G.6, such a quantizer satisfies
$$\mathbb{E}\left[\mathrm{Q}_{\mathrm{R},\mathrm{B}}(x)\right] = x \quad \text{and} \quad \mathbb{E}\left(\mathrm{Q}_{\mathrm{R},\mathrm{B}}(x) - x\right)^2 \leq \frac{\Delta^2}{4} = \frac{\mathrm{R}^2}{\left(2^{\mathrm{B}} - 1\right)^2}, \tag{60}$$
i.e., it is unbiased and the error variance depends on R and B. Here, the $\mathbb{E}(\cdot)$ is over the randomness from dithering in (59). If the input $x$ to the quantizer falls outside this range, i.e., $x > \mathrm{R}$ or $x < -\mathrm{R}$, the quantizer is said to be *saturated*. To quantize any matrix $\mathbf{X}$, $\mathrm{Q}_{\mathrm{R},\mathrm{B}}(\mathbf{X})$ is obtained by quantizing each entry independently, i.e., $[\mathrm{Q}_{\mathrm{R},\mathrm{B}}(\mathbf{X})]_{ij} \triangleq \mathrm{Q}_{\mathrm{R},\mathrm{B}}(X_{ij})$.

**Lemma G.6.** *For scalar $x$ with $|x| \leq \mathrm{R}$, denote the quantization error of uniformly dithered B–bit scalar quantizer as $\epsilon = \mathrm{Q}_{\mathrm{R},\mathrm{B}}(x) - x$. Then, $\mathbb{E}[\epsilon] = 0$ and $\mathrm{Var}\left(\epsilon\right) \leq \frac{\mathrm{R}^2}{(2^{\mathrm{B}}-1)^2}$, where the expectation $\mathbb{E}$ is over the randomness due to dithering in the quantizer operation.*

*Proof.* Suppose $x \in [q_k, q_{k+1})$ and $q_{k+1} = q_k + \Delta$, where $\Delta = \frac{2\mathrm{R}}{2^{\mathrm{B}}-1}$. Then,
$$\mathbb{E}\,\mathrm{Q}_{\mathrm{R},\mathrm{B}}(x) = q_{k+1}\frac{x - q_k}{\Delta} + q_k\left(1 - \frac{x - q_k}{\Delta}\right) = \frac{(q_k + \Delta)(x - q_k) + q_k(\Delta - x + q_k)}{\Delta} = x.$$

Furthermore, the variance can be upper bounded as

$$\text{Var}\left(\text{Q}_{\text{R,B}}(x) - x\right)^2 = (q_{k+1} - x)^2 \frac{(x - q_k)}{\Delta} + (q_k - x)^2 \left(1 - \frac{x - q_k}{\Delta}\right)$$
$$\leq (q_{k+1} - x)(x - q_k)$$
$$\leq \sup_{x \in [q_k, q_{k+1})} (q_{k+1} - x)(x - q_k)$$
$$= \left(q_{k+1} - \frac{q_k + q_{k+1}}{2}\right)\left(\frac{q_k + q_{k+1}}{2} - q_k\right) = \frac{\Delta^2}{4} = \frac{\text{R}^2}{(2^{\text{B}} - 1)^2}.$$

$\square$

When the input $x$ to the quantizer exceeds R, the quantizer is said to be **saturated**, causing the quantization error to deviate from zero mean and bounded variance. Thus, it is crucial to ensure that the quantizer operates within the **unsaturated** regime with high probability.

## H   Limitations and Further Discussions

Our proposed algorithm CALDERA exploits low-rank structure in weight matrices of LLMs, and is seen to boost the performance of existing methods in the literature of LLM compression. It does so by providing an additional degree of freedom to control the compression ratio – namely, the target rank ($k$). Theoretical guarantees show improved performance over existing benchmarks, when the matrix being compressed is inherently low-rank to begin with – something that is observed in LLMs. This also implies that CALDERA can be used for any other application scenarios that require matrix compression, as discussed in [31]. Its effectiveness largely depends on the presence of an inherent approximate low-rank structure, which is seen in many real-world matrices [37].

Despite attaining a lower loss for LLM compression, since CALDERA is designed to tackle an optimization problem through an iterative process, it requires slightly more computational resources. For instance, compressing Llama-2 7B or Mistral-7B models (with rank-256) took approximately 34 GPU-hours (on NVIDIA A10G GPUs provisioned from an AWS G5 instance), and compressing Llama-2 13B took 59 GPU-hours (on a locally hosted NVIDIA A6000 GPU). Moreover, LLaMa-2 70B can be quantized via CALDERA with rank-256 factors in approximately 90 GPU hours (on an H100 from Lambda labs), which is on par with QuIP# (with RHT finetuning), which reports 100 GPU hours (on A100). It should be noted that these wallclock times can be reasonably afforded, and are *not prohibitively large*. Furthermore, since the compression of an LLM is a one-time computational expense, this cost becomes highly amortized considering the frequent use of LLMs for inference following the deployment of the compressed model.

Although CALDERA often achieves perplexities and accuracies similar to unquantized models, a gap remains, as shown in Tables 1 and 2. This indicates there is potential for enhancing quantization strategies. For example, the target rank ($k$) can be treated as a hyper-parameter, and adjusted across different layers. Sharma et al. [33] demonstrate that such a layer-specific rank reduction can improve generalization capabilities. Additionally, improved fine-tuning strategies such as those proposed by Liu et al. [23], which reduce the gap between LoRA and full fine-tuning, can be incorporated with the low-rank adaptation step of CALDERA. Detailed investigations are left for future work.

**Broader Impacts**: Compressing models enables their deployment in resource-constrained settings, facilitating educational and technological advancements with limited infrastructure. Deploying on edge devices for inference also enhances privacy by reducing the need for data to be sent to centralized servers for inference, thereby enhancing user privacy and data security. Additionally, lower computational requirements of inference using compressed models is a step towards adoption of environment-friendly AI strategies.

On the other hand, as the compression is often not always lossless, any technique may result in reduced accuracy or loss of nuanced understanding, potentially impacting the reliability of the models in critical applications. Consequently, due diligence should be exercised when deploying these models. From a broader perspective, LLMs are quite powerful, and their easier deployment can possibly lead to misuse, such as the spread of misinformation or automated generation of harmful content. Consequently, robust regulatory frameworks are necessary, as LLMs continue becoming more accessible to the general audience.

