# OpenReview forum: "Compressing Large Language Models using Low Rank and Low Precision Decomposition"
_NeurIPS.cc/2024/Conference — NeurIPS 2024 poster_

### Official Review · Reviewer_23tr · 2024-06-21

**Soundness:** 3
**Presentation:** 3
**Contribution:** 3
**Rating:** 7
**Confidence:** 3

**Summary:**

This paper proposes a framework that combines quantization and low-rank approximation. Given a neural net weight matrix $W\in \mathbb{R}^{n\times d}$, it considers the following decomposition: $W=Q+LR$, where $Q$ is a quantized sketch of $W$ with very few bits (2 or 4 in their experiments), and $L\in \mathbb{R}^{n\times k}, R\in \mathbb{R}^{k\times d}$ are low-rank factors that serve as a "correction" to the quantization sketch $Q$. They are usually quantized with more bits (4 or 16). To compute respective $Q, L, R$, the authors propose an alternating minimization framework: starts with $L, R=0$, update $Q$ as $Q_t=Q_Q(W-L_{t-1}R_{t-1})$, then solve for $L_t, R_t$ by solving the rank-constrained regression against $W-Q_t$ and $X\in \mathbb{R}^{m\times d}$, where $X$ is the calibration dataset. They further show their algorithm could be used in conjunction with LoRA and fine-tuning the randomized Hadamard transform. Extensive experiments show that their proposed algorithm gives low perplexities and high accuracies when compared against earlier algorithms.

**Strengths:**

This paper proposes a method that is a good mixture of quantization and low-rank approximation, as simply using only one of quantization sketch or low-rank approximation intuitively "misses" an important part. The algorithm proposed in this paper is a natural and simple alternating minimization framework. This paper also contains a theoretical analysis for approximation error under assumptions on the target rank. Empirically, it is also shown the proposed algorithm has good performance under small rank and bit precision.

The paper is also well-written.

**Weaknesses:**

Overall, I don't see big weaknesses of this paper. One potential direction for improvement is theoretically, the alternating minimization framework could possibly be sped up given $k\ll n, d$, especially the rank-constrained regression, see [1]. On the other hand, it is not clear how to integrate these possible algorithmic pieces into post-processing a model weight, so I think the approach and runtime analysis of this paper are fair.

[1] Gu, Song, Yin and Zhang. Low Rank Matrix Completion via Robust Alternating Minimization in Nearly Linear Time. ICLR'24.

**Questions:**

If we allow $L$ and $R$ to be computed to full bit-precision, could the statement / result of Theorem 4.1 be simplified?

**Limitations:**

Yes.

---

> ### Author Rebuttal · Authors · 2024-08-07
>
> > If we allow $\mathbf{L}$ and $\mathbf{R}$ to be computed to full bit-precision, could the statement/result of Theorem 4.1 be simplified?
>
> Yes, in Thm. $4.1$, the error from quantizing the low-rank factors is captured in the additive $\epsilon$ term. When $\mathrm{B_L} \to \infty, \mathrm{B_R} \to \infty$, this error $\epsilon \to 0$, and we get $\frac{1}{nm}\mathbb{E}\left\lVert\mathbf{(Q + LR - W)X}\right\rVert_F^2 \lesssim \frac{4d\lambda_{max}\mathrm{R}^2}{\pi(2^{\mathrm{B_Q}}-1)^2}\left(1 - \frac{k}{2n}\right)^2$. Here, the only source of error is the finite precision of the $\mathbf{Q}$ matrix, determined by $\mathrm{B_Q}$. Furthermore, if the low-rank factors $\mathbf{L, R}$ are not required to be quantized, CALDERA simplifies to LoftQ -- the only difference being CALDERA uses LDLQ quantizer, whereas LoftQ uses NF4 quantizer.
>
> > One potential direction for improvement is theoretically, the alternating minimization framework could possibly be sped up given, especially the rank-constrained regression
>
> Thank you for pointing out this interesting work. We agree that solving rank-constrained regression can be sped up using sketching techniques from Gu et. al. [1]. A careful analysis is required to ensure that the error from sketching does not accumulate across the alternating iterations. A study of the efficacy of sketching in solving the non-convex optimization problem (1) is worth investigating.
>
> *[1] Gu et. al., Low Rank Matrix Completion via Robust Alternating Minimization in Nearly Linear Time (ICLR, 2024).*

---

> > ### Comment · Reviewer_23tr · 2024-08-07
> >
> > I thank the authors for answering my questions and including additional experiments. I'll keep my score as is.

---

### Official Review · Reviewer_GVVM · 2024-07-03

**Soundness:** 4
**Presentation:** 3
**Contribution:** 3
**Rating:** 7
**Confidence:** 2

**Summary:**

This paper introduces CALDERA, a new post-training compression algorithm for large language models (LLMs). CALDERA uses the inherent low-rank structure of LLM weight matrices by approximating them via a low-rank, low-precision decomposition $W \approx Q + LR$, where $L$ and $R$ are low-rank factors with quantized entries. CALDERA provides an effective way to compress large language models by exploiting their low-rank structure, enabling efficient distribution and deployment of LLMs on resource-constrained hardware while maintaining strong performance.

**Strengths:**

1. CALDERA introduces a new post-training compression technique for LLMs that exploits their inherent low-rank structure, setting it apart from existing methods. Also, the paper provides theoretical upper bounds on the approximation error of CALDERA using a rank-constrained regression framework, lending credibility to the approach.

2. Experimental results demonstrate that CALDERA outperforms existing post-training LLM compression techniques in the low-bit regime (less than 2.5 bits per parameter), highlighting its effectiveness. By enabling efficient distribution and deployment of LLMs on resource-constrained hardware, CALDERA can make LLMs more accessible and promote their broader adoption.

**Weaknesses:**

This paper is generally well-written. Here are some minor weaknesses:

1. The performance of CALDERA depends on the choice of target rank and quantization bit budget. The paper does not provide a systematic way to determine the optimal values for these hyperparameters. Is it possible to apply meta-learning here to decide the optimal values?

2. The experiments focus on compressing LLaMa models. It would be valuable to see how CALDERA performs on a wider range of LLMs to assess its generalizability.

3. As mentioned by the authors, the iterative nature of CALDERA's optimization process may require more computational resources compared to simpler compression methods, although this is a one-time cost.

**Questions:**

How does CALDERA perform compared to other compression approaches? Is it possible to combine CALDERA with these techniques?

**Limitations:**

Please see weaknesses.

---

> ### Author Rebuttal · Authors · 2024-08-07
>
> > wider range of LLMs
>
> We have performed quantization experiments on the Mistral 7B model, which can be found in Table 2 of the global response PDF. The PPLs obtained using CALDERA are consistently lower than QuIP# (no RHT fine-tuning) for comparable average bits. For future work, with additional compute, we intend to continue quantizing more such models.
>
> > other compression approaches ... possible to combine CALDERA
>
> Thank you for another interesting question. It is indeed possible to combine CALDERA with other existing model compression techniques.  A primary motivation of CALDERA of is to jointly solve two popular approaches in LLM compression, namely, low-rank decomposition and quantization -- both of which are usually treated as independently in existing works. The same motivation can indeed be used if one wants to adapt CALDERA to incorporate other compression approaches, such as sparsity, quantization-aware training (QAT), knowledge distillation, etc. For instance, the quantized $\mathbf{Q}$ matrix can be additionally pruned, making it sparse in each CALDERA iteration, further compressing the model. Moreover, provided additional compute, the quantized entries of our CALDERA decomposition can also be subjected to QAT using a straight-through estimator. QAT can either be done for fine-tuning on downstream tasks. Alternatively, QAT can also be done by performing knowledge distillation with an uncompressed teacher model, similar to what is done in LLM-QAT (ref. [1] below).
>
> *[1] Liu et. al., LLM-QAT: Data-Free Quantization Aware Training for Large Language Models (arXiV 2023).*
>
> > systematic way to determine the optimal values for these hyperparameters ... apply meta-learning here
>
> Thank you for this very interesting question. Meta-learning can indeed be useful. For instance, each matrix decomposition problem along with a given calibration set, can be treated as a separate task. The objective would be to learn how to choose the target rank and bit budget for the decomposition. The input to the meta-learning algorithm could be input characteristics of the matrix, such as dimensions, dynamic range of the entries, spectrum of the matrix, etc. Furthermore, the measure of performance could be the Frobenius norm error, possibly regularized with some metrics for the computation time required for obtaining the decomposition, or inference time with the decomposed representation. This is a direction worth exploring!
>
> > iterative nature of CALDERA's optimization process may require more computational resources
>
> Yes, we acknowledge this additional computational cost of compressing models iteratively. We will release open-source our compressed models, so that this one-time compression cost need not be incurred for someone who wishes to align or fine-tune our compressed models for downstream evaluations.
>
> Compressing Llama-2 7B or Mistral-7B models (with rank-256) took approximately 34 GPU-hours, when done on NVIDIA A10G GPUs provisioned from a G5 instance on AWS. Additionally, compressing Llama-2 13B took 59 GPU-hours, when done on NVIDIA A6000 GPUs. As noted in the paper, CALDERA is expected to take more time that QuIP# (without RHT finetuning) as it is an iterative algorithm. However, as seen above, the wallclock times are not prohibitively large. Furthermore, LLaMa-2 70B can be quantized via CALDERA with rank-256 factors in about 90 GPU hours on an H100 cluster, which is on par with QuIP# (with RHT finetuning), which reports 100 GPU hours. CALDERA is also faster than some other state-of-the-art methods such as AQLM, which reports a range between 192 to 384 GPU hours, and LLM-QAT, which requires 960 GPU hours. It should be kept in mind that the cost of compressing a model is a one-time cost, and it can be reasonably afforded as long as it is not prohibitively huge.

---

> > ### Comment · Reviewer_GVVM · 2024-08-08
> >
> > Thank you very much for your detailed response. I will keep my original score.

---

### Official Review · Reviewer_32tN · 2024-07-12

**Soundness:** 3
**Presentation:** 1
**Contribution:** 1
**Rating:** 2
**Confidence:** 5

**Summary:**

This paper introduces CALDERA, a novel post-training method that combines quantization and low-rank decomposition techniques for compressing large language models. The primary contribution lies in the design of a combined pipeline and the application of low-rank decomposition. Experimental results demonstrate that integrating the designed low-rank decomposition technique enhances compression performance compared to using quantization alone.

**Strengths:**

1. This paper addresses a highly significant and widely studied research topic: compressing large language models.

2. The author not only conducts experiments but also provides numerical evidence to demonstrate the effectiveness of the designed compression method.

3. Experimental results indicate that the combined methods designed in this study effectively enhance the performance of large language models compressed using quantization-only methods.

**Weaknesses:**

1. **Missing Experimental Results:** The experiments conducted in this paper are inadequate. Many critical experiments conducted in previous works have not been included. For instance:

- Evaluation of throughput for LLaMA models compressed by CALDERA and QuIP# under the same average bits with the same batch size and sequence length. It's crucial to assess whether combining low-rank decomposition with quantization compromises efficiency compared to uncompressed LLMs.
- Evaluation results on LLMs with different architectures from LLaMA and on generation datasets.
- Evaluation of time consumption between CALDERA and QuIP# for compressing LLMs.
- Evaluation of compression performance using different calibration sets and varying numbers of calibration data.
2. **Lack of Novelty:** The novelty of this paper is limited. The primary contributions are the design of a combination framework and a low-rank decomposition algorithm, both of which appear simplistic and resemble iterative weight compression methods with minor losses. The use of the existing quantization method LDLQ for combination and the effectiveness of the designed low-rank decomposition method in achieving minimal compression loss are questionable. Replacing it with established work like data whitening method in SVD-LLM[1] might yield better performance, given SVD-LLM's theoretical proof of achieving global optimality through data whitening.

*[1] Wang, X., Zheng, Y., Wan, Z., & Zhang, M. (2024). SVD-LLM: Truncation-aware Singular Value Decomposition for Large Language Model Compression. https://arxiv.org/abs/2403.07378*

3. **Intuitive Experimental Results:** While it's intuitive that combining low-rank decomposition with quantization can improve performance over quantization-only methods, existing works such as SVD-LLM already demonstrate this. To add value to the proposed methods, it's crucial to include experimental results where the average bits are equal to or less than 1-bit, demonstrating capabilities that quantization-only methods cannot achieve.

**Questions:**

See above.

**Limitations:**

N.A.

---

> ### Author Rebuttal · Authors · 2024-08-07
>
> > Novelty
>
> We propose an idea which is simple in its execution; that does not mean it lacks novelty. CALDERA is the first work that combines quantization with a low-rank decomposition -- both of which are usually treated independently in existing works on LLM compression. Additionally, a significant contribution of this work lies in the novel theoretical guarantees bounding the expected approximation error with respect to the bit budget. The quantization of the low-rank factors, as well as the complexity of the LDLQ quantization method, make these bounds challenging to obtain. Thank you for pointing us out to the nice work, SVD-LLM. Specifically, our work is different in the following aspects:
>
> 1. For a given weight matrix $\bf A$, SVD-LLM considers solving the rank-constrained regression problem, ${\rm min}_{\bf L,R}|| {\bf (LR-Z)X} ||_F^2$ to global optimality. This is a known result (ref. [1] below). We use an equivalent whitening process in Lemma 4.2 of our paper. Additionally, our theoretical guarantees upper bounds the perturbation error to this optimal solution due to the **joint quantization constraints** on $\bf L$ and $\bf R$ -- something not considered in the SVD-LLM paper. Quantization of low-rank factors in SVD-LLM is done independent to the low-rank decomposition, (i.e., one after the other) in Sec. 4.4, which is suboptimal compared to the joint decomposition of our LPLRFactorize module. This is reflected in the numerical results, where SVD-LLM reports a perplexity of 13.29 (Table 8 for SVD-LLM + GPTQ-4bit), which is significantly higher that our reported PPLs in Table 1 (for example, 6.19 for CALDERA Rank-256, $B_L = B_R = 4$).
>
> 2. Moreover, CALDERA considers a $\bf Q + LR$ decomposition. This additive matrix $\bf Q$ (not considered in SVD-LLM), coarsely captures the effect of trailing singular values, which are entirely truncated in SVD-LLM. The $\bf Q$ matrix enables CALDERA to achieve PPLs close to the unquantized model in the extreme compression regime considered.
>
> *[1] Xiang, et. al., Optimal exact least squares rank minimization (ACM SIGKDD, 2012)*
>
> > Intuitive Experimental Results
>
> One of the core motivations of post-training quantization of LLMs is to ensure minimal loss in PPLs or accuracies compared to the uncompressed model. Current methods that compress an LLM in the regime of 2 to 2.5 bits per parameter, fail to close the gap to uncompressed performance. While contemporary works like BiLLM (Huang et. al, 2024) do consider the lower ~1 bit per parameter regime, the associated PPLs are relatively high compared to the uncompressed model. In contrast, our work is motivated with **closing the gap with respect to uncompressed models** in the 2 to 2.5 bits per parameter regime.
>
> While it is indeed intuitive that low-rank + quantization is expected to perform better than quantization only methods, our work rigorously formalizes this intuition with theoretical backing and experimental results. Although both SVD-LLM and our work explore the benefits of combining quantization with low-rank decomposition, SVD-LLM directly applies LLM quantization methods to the low-rank factors in an orthogonal fashion (which is suboptimal), whereas CALDERA additively combines a full-rank, aggressively-quantized matrix with a **quantized low-rank correction** (which is obtained by treating the quantization and low-rank constraints **jointly**).
>
> > Evaluation of throughput
>
> We agree that it is important to ensure CALDERA does not degrade generation throughput. We have performed some throughput experiments, which can be found in Table 1 of the global response PDF. CALDERA achieves a slightly lower throughput than QuIP#, as it requires dequantization of 3 matrices -- $\bf Q, L$ and $\bf R$ (instead of just one in the case of QuIP#), and perform additional matrix multiplications during inference. Despite this, it is worthwhile to note that the throughput is quite higher than the uncompressed model. Furthermore, it should be kept in mind that designing specialized CUDA kernels that are aware of the $\bf Q + LR$ decomposition can improve the throughput even further -- for example, by computing $\bf Qx$ and $\bf LRx$ in parallel. Writing custom kernels is left for future work.
>
> > Evaluation ... different calibration sets
>
> Our choice of sampling from RedPajama to get our calibration dataset is motivated from the fact that OpenLlama uses it to pre-train the model from scratch, and it is important to ensure that the calibration data is from the same distribution as the pre-training data. While ablations over calibration datasets is worthwhile to explore, we have decided to prioritize other experiments (cf. global response PDF) within our current computational budget.
>
> > different architectures
>
> We have performed additional experiments (specifically, we compressed Llama-2 13B and Mistral-7B), which can be found in Table 2 of the global response PDF.
>
> > Evaluation ... time consumption
>
> While a comprehensive evaluations would require more time, we report some numbers here. Compressing Llama-2 7B or Mistral-7B models (with rank-256) took approximately 34 GPU-hours, when done on NVIDIA A10G GPUs provisioned from a G5 instance on AWS. Additionally, compressing Llama-2 13B took 59 GPU-hours, when done on NVIDIA A6000 GPUs.
>
> CALDERA is expected to take more time that QuIP# (no RHT finetuning) as it is an iterative algorithm. However, as seen above, the wallclock times are not prohibitively large. Also, LLaMa-2 70B can be quantized via CALDERA with rank-256 factors in ~90 GPU hours on an H100 cluster, which is on par with QuIP# (with RHT finetuning), which reports 100 GPU hours. CALDERA is also faster than some other state-of-the-art methods such as AQLM, which reports a range between 192 to 384 GPU hours, and LLM-QAT, which requires 960 GPU hours. It should be kept in mind that the cost of compressing a model is a one-time cost, and it can be reasonably afforded as long as it is not prohibitively huge.

---

> > ### Author Response · Authors · 2024-08-10
> > **Further clarifications**
> >
> > Dear Reviewer 32tN,
> >
> > Please let us know if your queries have been addressed satisfactorily. As mentioned in our response, we've thoroughly incorporated your feedback, along with suggestions from the other reviewers. We hope that our response has positively influenced your perception of our work.
> >
> > If you require further clarifications to potentially reconsider your score, we are enthusiastic about engaging in further discussion. Please do not hesitate to contact us. We highly value the generous contribution of your time to review our paper.

---

> ### Comment · Reviewer_32tN · 2024-08-10
> **More concerns**
>
> Thank the authors for the detailed response and clarification. After reading the global response given by the authors. I have two more concerns.
> 1. **The compression time cost of CALDERA is much higher than QuIP#.** In Appendix F.7 of QuIP#'s paper, the authors claim that "All experiments were run on NVIDIA A100 GPUs ... We find that we can quantize Llama2 70B without fine-tuning in under 10 GPU-hours and with fine-tuning in around 100 GPU-hours." Therefore, the comparison of time costs in the global response is **NOT fair**. Based on the GPU hours for running CALDERA and QuIP#, I made the following table. The time cost of CALDERA without fine-tuning is already higher than QuIP# with fine-tuning. However, the perplexity (PPL) of the LLM compressed by CALDERA without fine-tuning is not as good as that compressed by QuIP# with fine-tuning, as reported in [1]. Therefore, given the same compute budget for compression, CALDERA is **less competitive** than QuIP#. Also, since QuIP# requires less time than CALDERA for compression, the explanation in the footnote of page 8 in the submission for using the LLM compressed by QuIP# but **WITHOUT** being fully fine-tuned is **NOT convincing**, and the all results reported in Section 5.2 are also **NOT convincing**. Therefore, it is still **questionable** whether the designed algorithm CALDERA is better than the baseline method QuIP#.
>
> |         | w/o Fine-tuning    | w/ fine-tuning    |
> |---------|-------------------|--------------------|
> | CALDERA | 90 H100 GPU-hours | Unknown            |
> | QuIP#   | 10 A100 GPU-hours | 100 A100 GPU-hours |
>
> 2. **The reported generation throughput is poor.** In Table 1 shown in the global response, the throughput of LLM compressed by CALDERA with its best configuration (Rank=256, $B_L = B_R = 4$) is 45 tok/sec, which is much worse than that of QuIP#, which is 76 tok/sec. Therefore, if throughput is the primary target, CALDERA is still **less competitive** than QuIP#.
>
> Based on the concerns above, I have decided to temporarily adjust my score to reject and I hope more clarifications on these two concerns.
>
> Thank you.
>
> [1] Albert Tseng et al., QuIP#: Even Better LLM Quantization with Hadamard Incoherence and Lattice Codebooks. ICML, 2024

---

> > ### Comment · Reviewer_32tN · 2024-08-13
> > **Abnormal Results in Table 5**
> >
> > When comparing the accuracy of uncompressed LLaMA 2-7B and LLaMA 2-70B in Table 1 in the submission with LLaMA 2-7B compressed by CALDERA with LoRA fine-tuning in Table 5, I found that the accuracy of compressed LLaMA 2-7B is much higher than that of the uncompressed LLaMA 2-7B  and even uncompressed LLaMA 2-70B, as shown below.
> > |                                          | Bits | Wino  | RTE   |
> > |------------------------------------------|------|-------|-------|
> > | Uncompressed LLaMA 2-7B                  | 16   | 67.3  | 63.2  |
> > | Uncompressed LLaMA 2-70B                 | 16   | 77.0  | 67.9  |
> > | LLaMA 2-7B compressed by CALDERA with FT | 2.4  | 84.93 | 86.28 |
> >
> > This is **abnormal**, as the compressed LLM shows a significant improvement in accuracy despite an extensive compression ratio (the weight memory of LLaMA 2-70B 16bit is about 80 times larger than that of LLaMA 2-7B 2bit). The reasonable explanations I can think of are either the failure to use the optimal parameter configuration during inference, or the overfitting of the fine-tuned LLM on these three datasets. Therefore, the results in table 5 are **NOT convincing**.

---

> ### Author Response · Authors · 2024-08-13
> **Response to more concerns and abnormal results**
>
> Dear Reviewer 32tN, we believe you have made two major misunderstandings. Please allow us to clarify them.
>
> Firstly, we believe there has been a misunderstanding as to what is referred to as fine-tuning in our paper. In LoRA fine-tuning, the model is adapted or fine-tuned using a **smaller, task-specific dataset**. Please note that Randomized Hadamard Transform fine-tuning (RHT-FT) and LoRA fine-tuning are two separate components, and independent of each other. RHT-FT is a step of the quantization process in QuIP# -- it is done on the calibration set, and is meant to decrease the PPL (or increase the accuracy) across all tasks generally. In contrast, LoRA fine-tuning is task-specific, and is done on a *specific dataset* like Wino or RTE. For instance, an accuracy of 84.93\% for Wino and 86.28\% for RTE, is a consequence of fine-tuning the LLM using LoRA on Wino and RTE, respectively. The uncompressed models have **NOT** been fine-tuned to these task specific datasets, which is why CALDERA + Fine-tuning performs better than the uncompressed model. This is very natural to expect as a consequence of fine-tuning, and not abnormal at all. It is observed in other prior works as well -- for example, see Fig. 2 of LQ-LoRA [1]. Hence, it is **not a fair comparison** to compare *CALDERA-quantized models fine-tuned for a specific downstream task* with *non fine-tuned uncompressed models*.
>
> [1] Guo et. al, LQ-LoRA: Low-rank Plus Quantized Matrix Decomposition for Efficient Language Model Finetuning, ICLR 2024.
>
> Secondly, we believe there has been a misunderstanding on your end regarding performance of our compressed models and prior works like QuIP#. We agree that if the metric of performance is purely quantization time or generation throughput, QuIP# achieves higher performance *with respect to those metrics*. However, that has never been the primary thesis of our work, i.e., we do not claim that compressing LLMs with CALDERA is faster than QuIP#. A central thesis of our work is that CALDERA **does** perform better than QuIP\# in the sense that that it consistently achieves lower perplexities than QuIP# -- when comparing *QuIP# without RHT-FT to CALDERA without RHT-FT*, or *QuIP# with RHT-FT to CALDERA with RHT-FT*. As RHT-FT is an optimization that applies to both QuIP# and CALDERA, it is unfair to compare accuracies and PPLs achieved by QuIP# with RHT-FT to CALDERA without RHT-FT. For a fair comparison, please refer to Table 3 where we have compared CALDERA with RHT-FT and QuIP# with RHT-FT, for 7B models. It is evident from Table 5 that CALDERA consistently has lower PPL than QuIP#. Furthermore, if you look at Table 4 of QuIP#, they report PPLs of 6.19 (Wiki) and 8.16 (C4), which are still higher than 5.84 (Wiki) and 7.75 (C4) (highlighted numbers on Table 5).
>
> We have not done RHT-FT on the 70B model because of limited rebuttal duration. But our original submission **does report** extensive comparisons with RHT-FT for 7B models. Doing such extensive comparisons with the 70B model within the rebuttal window is infeasible. We report wallclock time for quantization, just to show that it *not prohibitively high* when compared to QuIP#. Additionally, doing RHT-FT on the CALDERA quantized 70B would take the same amount of time as the RHT-FT step in QuIP# does, which is once again, *not prohibitively slow*, given this is a one-time cost.
>
> Regarding throughput, our global response simply shows that it is not degraded much compared to QuIP# due to the low-rank component. It should be noted that it is still significantly higher than the unquantized model (even without custom CUDA kernels for the low-rank component).
>
> We reiterate that our work is motivated with closing the gap with respect to uncompressed models in the 2 to 2.5 bits per parameter regime. And, it is clear that we are able to get lower perplexities than QuIP# when the comparison is fair, i.e., either with or without RHT-FT in both cases.

---

> > ### Comment · Reviewer_32tN · 2024-08-14
> > **Keep my rating**
> >
> > Thanks for the authors' response. However, the response does not address my concerns:
> > 1. **The author is confusing the fine-tuning of LLMs with the fine-tuning of BERT.** LLMs are designed to handle a wide range of tasks, including text generation, translation, summarization, and more. Fine-tuning an LLM often involves adapting the model to perform well on diverse tasks rather than focusing on a single, narrowly defined task like classification. In contrast, BERT is often fine-tuned for specific tasks such as classification, question answering, and named entity recognition.  I highly recommend that the authors add evaluations on several generation tasks. As mentioned before, I deeply suspect that the LLM is overfitting on fine-tuned classification datasets such as RTE and will show much poorer generation ability.
> > 2. **The author does not follow the literature to fine-tune the LLM.** To the best of our knowledge, none of the existing LLM compression works fine-tune their compressed LLM on classification datasets. Common datasets for fine-tuning include language modeling datasets such as WikiText-2 and C4, or instruction tuning datasets such as Alpaca. The paper [1] mentioned by the authors in their response also follows this trend. For instance, the authors fine-tune RoBERTa-Large on classification datasets but fine-tune models like LLaMA-2 on C4 and OpenAssistant datasets. We recommend the authors follow the literature to design the LoRA fine-tuning experiments.
> > 3. **The author does not address the poor time cost for compression and the poor throughput for inference.** Even though CALDERA achieves slightly better PPL than QuIP#, its poor time cost for compression and inferior throughput for inference make it less attractive. For example, when the goal is to compress the LLM to reduce its inference latency on a server where memory is not the main concern, CALDERA is even less appealing than some 3-bit quantization algorithms, as these algorithms guarantee lower latency and similar or even better accuracy.
> > 4. **The author should use the default configuration to fully RHT-FT QuIP# and update the evaluation results on Table 3 and Table 4.** It is insufficient to only use PPL to compare the accuracy of the compressed LLMs, especially when the PPL values from two compressed LLMs are so close. The authors should update the evaluation results of LLMs compressed by QuIP# with fully RHT-FT in both Table 3 and Table 4 in the submission.
> >
> > Therefore, I would still like to keep my rating.

---

### Official Review · Reviewer_xRtR · 2024-07-13

**Soundness:** 2
**Presentation:** 3
**Contribution:** 2
**Rating:** 4
**Confidence:** 2

**Summary:**

The article introduces CALDERA, a post-training compression algorithm for large language models (LLMs) that leverages the low-rank structure of weight matrices to achieve significant compression. CALDERA approximates a weight matrix $W$ using a low-rank, low-precision decomposition $W \approx Q + LR$, where $Q$, $L$ and $R$ are quantized. The method aims to reduce the memory and computational footprint of LLMs, facilitating their deployment on memory-constrained edge devices.

**Strengths:**

Innovative Compression Technique: CALDERA combines low-rank approximation with low-precision quantization, addressing both the redundancy and precision issues in LLM weight matrices.

Theoretical Foundations: The algorithm is backed by rigorous theoretical guarantees on the approximation error, which enhances its reliability.

Performance: Empirical results show that CALDERA outperforms existing post-training compression techniques in terms of zero-shot performance, particularly when compressed to less than 2.5 bits per parameter.

Adaptability: The method supports low-rank adaptation, which can further enhance the performance of the compressed models on specific tasks.

**Weaknesses:**

Complexity: The algorithm involves a nested optimization process, which may introduce significant computational overhead during the compression phase.

Dependency on Calibration Data: The performance of CALDERA relies on the availability and quality of calibration data, which may not always be accessible or representative.

Limited Empirical Comparisons: While the article claims superiority over existing methods, the empirical comparisons might be limited in scope, potentially omitting some relevant state-of-the-art techniques.

Potential Issues

Quantization Artifacts: Aggressive quantization, especially at very low bit budgets, might introduce artifacts that could degrade model performance in specific scenarios.

Generalization: The effectiveness of the method across a diverse range of LLMs and tasks needs thorough validation, as it might not generalize well beyond the tested models and datasets.

Optimization Stability: The iterative nature of the optimization process might lead to stability issues, particularly for large models with highly non-convex loss landscapes.

**Questions:**

See Weaknesses

**Limitations:**

Yes

---

> ### Author Rebuttal · Authors · 2024-08-07
>
> > Quantization Artifacts
>
> We agree that compression introduces artifacts. However, PPLs on language modeling datasets like Wikitext and C4 are generally considered to be reasonably good indicators of the performance of an LLM, as can be seen in the existing literature. A comprehensive evaluation over the complete spectrum of LLM evaluation tasks is beyond the scope of our computational budget. In addition, we have performed additional model quantization experiments on different sizes of LLaMa-2 model, as well as Mistral 7B, for which CALDERA generalizes well. These results can be found in Table 2 in the global response PDF.
>
> > Limited Empirical Comparisons
>
> Please refer to additional experiments in the global response PDF.
>
> > Optimization Stability
>
> While it is possible that the iterative nature of our algorithm may oscillate, it is not a cause of concern as Algs. 1 and 2 keep track of (and return) the best matrices $\mathbf{Q}$, $\mathbf{L}$ and $\mathbf{R}$ seen across all the iterations. Moreover, as can be seen from our Frobenius norm vs. iteration plots in Appendix G, the convergence of our algorithm is pretty well-behaved when decomposing LLM weight matrices. Furthermore, our theoretical guarantee in Thm 4.1 serves as an assurance that stability is not a cause of concern.
>
> > Complexity
>
> We have discussed the computational complexity of CALDERA in Appendix D, and acknowledge it as a price paid for the improved error guarantees (as stated in Thm 4.1, as well as our numerical evaluations).
>
> Additionally, we report (and compare) some numbers here. Compressing Llama-2 7B or Mistral-7B models (with rank-256) took approximately 34 GPU-hours, when done on NVIDIA A10G GPUs provisioned from a G5 instance on AWS. Additionally, compressing Llama-2 13B took 59 GPU-hours, when done on NVIDIA A6000 GPUs.
>
> As noted in the paper, CALDERA is expected to take more time that QuIP\# (without RHT finetuning) as it is an iterative algorithm. However, as seen above, the wallclock times are not prohibitively large. Furthermore, LLaMa-2 70B can be quantized via CALDERA with rank-256 factors in about 90 GPU hours on an H100 cluster, which is on par with QuIP# (with RHT finetuning), which reports 100 GPU hours. CALDERA is also faster than some other state-of-the-art methods such as AQLM, which reports a range between 192 to 384 GPU hours, and LLM-QAT, which requires 960 GPU hours. It should be kept in mind that the cost of compressing a model is a one-time cost, and it can be reasonably afforded as long as it is not prohibitively huge.
>
> > Dependency on Calibration Data
>
> While this is true, the availability of good quality data is a central premise for the success of any data-centric algorithm. It is possible to derive an entirely data-agnostic variant of CALDERA, by simply replacing the calibration data matrix $\mathbf{X}$ by an identity matrix, $\mathbf{I}$. However, using a data-aware version yields better results as validated by ours, as well as several prior works, including QuIP# and LQ-LORA.

---

> > ### Author Response · Authors · 2024-08-12
> > **Further clarifications**
> >
> > Dear Reviewer xRtR,
> >
> > Please let us know if your queries have been addressed satisfactorily. As mentioned in our response, we've thoroughly incorporated your feedback, along with suggestions from the other reviewers. We hope that our response has positively influenced your perception of our work.
> >
> > If you require further clarifications to potentially reconsider your score, we are enthusiastic about engaging in further discussion. Please do not hesitate to contact us. We highly value the generous contribution of your time to review our paper.

---

### Official Review · Reviewer_ZG6L · 2024-07-14

**Soundness:** 3
**Presentation:** 3
**Contribution:** 3
**Rating:** 7
**Confidence:** 4

**Summary:**

This work proposes a new decomposition procedure to achieve low-precision and low-rank compression of large language model (LLM) weight matrices. Such a new method could capture high singular components accurately while compressing less significant ones. An efficient algorithm is proposed to optimize the quantized backbone and low-rank factors, which can be fine-tuned to enhance the model performance. The work also provides a tighter approximation error bound and is validated by compressing the LlaMA LLMs to below 2.5 bits per parameter.

**Strengths:**

1. The proposed CALDERA method can effectively compress LLMs without degrading the performance too much. And it is demonstrated through a series of experiments that the method is capable of model compression and low-rank adaptive fine-tuning at the same time.
2. The paper also provides a rigorous approximation error analysis of the proposed CALDERA method.

**Weaknesses:**

I think the presented experiment study is limited. I mainly have the following suggestions.
1. It would be better to supplement the experiments with LoftQ and LQ-LoRA, and add a set of experiments with full parameter fine-tuning as a baseline.
2. Please provide an experimental comparison of model size.
3. It would be better to add some experiments on evaluating the computation cost of the proposed CALDERA.

**Questions:**

1. Since the CALDERA method requires many time-consuming operations like SVD and matrix inversion, can this be improved in future work?
2. Are there any efficient ways to improve the selection strategy of the hyperparameter rank?

**Limitations:**

In my opinion, how to efficiently tune the  hyperparameter rank is the main limitation of the proposed method.

---

> ### Author Rebuttal · Authors · 2024-08-07
>
> > experimental comparison of model size
>
> We have performed ablations over multiple sizes of the Llama-2 family. Please refer to Table 2 of the global response. The PPLs obtained using CALDERA are consistently better than QuIP# (without fine-tuning) for comparable average bits.
>
> > evaluating the computation cost
>
> Thank you for this suggestion. While a comprehensive evaluations would require more time, we report (and compare) some numbers here. Compressing Llama-2 7B or Mistral-7B models (with rank-256) took approximately 34 GPU-hours, when done on NVIDIA A10G GPUs provisioned from a G5 instance on AWS. Additionally, compressing Llama-2 13B took 59 GPU-hours, when done on NVIDIA A6000 GPUs.
>
> As noted in the paper, CALDERA is expected to take more time that QuIP# (without RHT finetuning) as it is an iterative algorithm. However, as seen above, the wallclock times are not prohibitively large. Furthermore, LLaMa-2 70B can be quantized via CALDERA with rank-256 factors in about 90 GPU hours on an H100 cluster, which is on par with QuIP# (with RHT finetuning), which reports 100 GPU hours. CALDERA is also faster than some other state-of-the-art methods such as AQLM, which reports 192 to 384 GPU hours, and LLM-QAT, which requires 960 GPU hours. Additionally, each layer is quantized independent of other layers -- hence, the wallclock time for CALDERA can be reduced by processing the layers in parallel (i.e., scaling horizontally by using more GPUs). Moreover, it should be kept in mind that the cost of compressing a model is a one-time cost, and it can be reasonably afforded as long as it is not prohibitively huge.
>
> > time-consuming operations like SVD and matrix inversion, can this be improved in future work?
>
> Thank you for raising this question. It is indeed possible to reduce the computational complexity of CALDERA. For instance, the SVD computation in the LPLRFactorize submodule can be replaced with randomized LPLR submodule from (ref [30] in the paper), which leverages Gaussian sketching matrices to reduce the complexity from $O(nd^2)$ to $O(ndm)$, where $m \ll \mathrm{min}\\{n,d\\}$ is the sketch size.
>
> Furthermore, we would like to clarify that matrix inversion is not necessary to obtain the left and right low-rank factors. These factors can be obtained by directly solving the corresponding least-squares minimization problem (in lines 8 and 9 of Alg. 2) using a conjugate gradient descent based solver, which will be significantly faster. The closed form expressions in our paper are used to facilitate analysis and derive Thm 4.1.
>
> More generally, the constrained optimization problem (1) is NP-hard, and designing efficient algorithms to solve it is an interesting research avenue, with applications much broader than LLM compression.
>
> > selection strategy of the hyperparameter rank?
>
> Thank you for this very interesting question! At a high level, it is possible to adaptively select the rank of each layer during fine-tuning using a strategy similar to AdaLoRa [1]. The initial rank can be chosen to be high, for instance 256. Subsequently, during fine-tuning for a downstream task, the singular values of the low-rank component can be analyzed such that smaller singular values can be truncated to adaptively reduce the rank of each layer. Finding efficient and optimal ways to do this warrants a deeper investigation.
>
> [1] Zhang et. al., Adaptive Budget Allocation for Parameter-Efficient Fine-Tuning (ICLR, 2023)
>
> > experiments with LoftQ and LQ-LoRA ... full parameter fine-tuning as a baseline.
>
> The results of LoftQ and LQ-LoRa are restated in Table 5 (copied from respective papers). Fine-tuning the low-rank factors obtained from CALDERA shows that we get lower PPLs for fewer average number of bits when compared to LoftQ or LQ-LoRA.
>
> LoftQ and LQ-LoRA report PPLs that are at par or better than vanilla LoRA on WikiText2 PPLs. Moreover, it is know that for several tasks, LoRA performs at par with full fine-tuning. Therefore, we have prioritized utilizing our limited computational budget for other ablation experiments on CALDERA (please refer to the global response).

---

> > ### Author Response · Authors · 2024-08-10
> > **Further clarifications**
> >
> > Dear Reviewer ZG6L,
> >
> > Please let us know if your queries have been addressed satisfactorily. As mentioned in our response, we've thoroughly incorporated your feedback, along with suggestions from the other reviewers. We hope that our response has positively influenced your perception of our work.
> >
> > If you require further clarifications to potentially reconsider your score, we are enthusiastic about engaging in further discussion. Please do not hesitate to contact us. We highly value the generous contribution of your time to review our paper.

---

> > > ### Comment · Reviewer_ZG6L · 2024-08-13
> > >
> > > I think the authors have addressed my comments. I shall change my rating to 7.

---

### Author Rebuttal · Authors · 2024-08-07

Dear Reviewers,

We are very grateful for the valuable time you spent in reading our paper and sharing your concerns, and greatly appreciate the voluntary nature of the review process. In this global response, we have have summarized the major points from our individual responses.

**Additional experiments compressing other models**: We used CALDERA to compress some additional popular LLMs -- namely, **Llama-2 13B, 70B**, and **Mistral 7B**, which can be found in Table 2 of the global response PDF. We also recompressed Llama-2 70B using the Hessians provided by QuIP# for a fairer comparison. The perplexities obtained using CALDERA are consistently lower than QuIP# (without RHT finetuning). Similar trend can be seen for the zero-shot task accuracies as well.

**Additional experiments to evaluate throughput**: We also did additional evaluations on the generation throughput of our compressed models, which can be found in Table 1 of the global response PDF. It is noteworthy that the throughput of our CALDERA-compressed model is quite higher than the uncompressed model. CALDERA does achieve a slightly lower throughput than QuIP# -- however, this is expected, as it requires dequantization of three matrices -- $\bf Q, L$ and $\bf R$.  It should be kept in mind that designing specialized CUDA kernels that are aware of the $\bf Q + LR$ decomposition can improve the throughput even further.

**Computation cost of compressing using CALDERA**: We conducted some more experiments to compare the wall-clock times of using CALDERA. Compressing Llama-2 7B or Mistral-7B models (with rank-256) took approximately 34 GPU-hours, when done on NVIDIA A10G GPUs provisioned from a G5 instance on AWS. Additionally, compressing Llama-2 13B took 59 GPU-hours, when done on NVIDIA A6000 GPUs.

CALDERA is expected to take more time that QuIP\# (no RHT finetuning) as it is an iterative algorithm. However, as seen above, the wallclock times are not prohibitively large. Also, LLaMa-2 70B can be quantized via CALDERA with rank-256 factors in ~90 GPU hours on an H100 cluster, which is on par with QuIP\# (with RHT finetuning), which reports 100 GPU hours. CALDERA is also faster than some other state-of-the-art methods such as AQLM, which reports a range between 192 to 384 GPU hours, and LLM-QAT, which requires 960 GPU hours. It should be kept in mind that the cost of compressing a model is a one-time cost, and it can be reasonably afforded as long as it is not prohibitively huge.

In summary, we reiterate that our work proposes and rigorously analyzes a simple optimization-centric point of view to the problem of obtaining quantized and low-rank decompositions of the weight matrices of an LLM. We evaluate the success of our $\bf Q + LR$ style of matrix decomposition algorithm for post-training LLM quantization in the challenging regime of 2 to 2.5 bits per parameter. Our LLM compression scheme reduces the model distribution and deployment costs, and the low-rank component $\bf LR$ provides good initializations for further fine-tuning using popular low-rank adaptation methods. We hope our responses clarify the reviewers' concerns.

---

### Decision · Program_Chairs · 2024-09-25

**Decision:**

Accept (poster)

**Comment:**

This paper introduces a new method for compressing LLM's by taking advantage of both quantization and low-rank approximation of weight matrices. In contrast to prior work, they take advantage of recent approaches for *jointly* optimizing a quantized, low-rank approximation, which surprisingly performs quite a bit better than treating the problems separately from each other (e.g., finding a low-rank approximation first, then quantizing the factors). There were some concerns with the paper's experimental setup, but my feeling is that the concerns were sufficiently addressed during the response period. Without a clear benchmark for LLM compression, decisions have to be made in terms of exactly how to conduct the experiments and which ones to conduct, and the paper's choices seem reasonable enough. For the experiments they do run, the results show a clear improvement over prior approaches that treat low-rank approximation and quantization as separate steps in the compression process. For this reason, we recommend accepting the paper.